# Partition between Supercooled Liquid Droplets and Ice Crystals in Mixed-phase Clouds based on Airborne In-situ Observations

Flor Vanessa Maciel[1,2], Minghui Diao[1], Ching An Yang[1]

[1]Department of Meteorology and Climate Science, San Jose State University, San Jose, 95192, USA

[2]*Current affiliation:* Department of Atmospheric and Oceanic Sciences, University of California, Los Angeles, 90095, USA

*Correspondence to*: Minghui Diao (minghui.diao@sjsu.edu)

**Abstract.** The on-set of ice nucleation in mixed-phase clouds determines ice cloud lifetime and their microphysical properties. In this work, we develop a novel method that differentiates various phases of mixed-phase clouds, such as clouds dominated by pure liquid or pure ice segments, compared with those having ice crystals surrounded by supercooled liquid water droplets

or vice versa. Using this method, we examine the relationship between the macrophysical and microphysical properties of Southern Ocean mixed-phase clouds at -40 to 0°C (e.g., stratiform and cumuliform clouds) based on the in-situ aircraft-based observations during the US National Science Foundation Southern Ocean Clouds, Radiation, Aerosol Transport Experimental Study (SOCRATES) flight campaign. The results show that the exchange between supercooled liquid water and ice crystals in a macrophysical perspective, represented by the increasing spatial ratio of regions containing ice crystals relative to the total

in-cloud region (defined as ice spatial ratio), is positively correlated with the phase exchange in a microphysical perspective, represented by the increasing ice water content (IWC), decreasing liquid water content (LWC), increasing ice mass fraction, and increasing ice particle number fraction (IPNF). The mass exchange between liquid and ice becomes more significant during phase 3 when pure ice cloud regions (ICRs) start to appear. Occurrence frequencies of cloud thermodynamic phases show significant phase change from liquid to ice at a similar temperature (i.e., -17.5°C) among three types of definitions of

mixed-phase clouds based on ice spatial ratio, ice mass fraction, or IPNF. Aerosol indirect effects are quantified for different phases using number concentrations of aerosols greater than 100 nm or 500 nm ($N_{>100}$ and $N_{>500}$, respectively). $N_{>500}$ shows stronger positive correlations with ice spatial ratios compared with $N_{>100}$. This result indicates that larger aerosols potentially contain ice nucleating particles (INPs), which facilitate the formation of ice crystals in mixed-phase clouds. The impact of $N_{>500}$ is also more significant in phase 2 when ice crystals just start to appear in mixed phase compared with phase 3 when

pure ICRs have formed, possibly due to the competing aerosol indirect effects on primary and secondary ice production in phase 3. The thermodynamic and dynamic conditions are quantified for each phase. The results show stronger in-cloud turbulence and higher drafts in phases 2 and 3 when liquid and ice coexist than pure liquid or ice (phases 1 and 4, respectively). Highest updrafts and turbulence are seen in phase 3 when supercooled liquid droplets are surrounded by ice crystals. These results indicate both updrafts and turbulence support the maintenance of supercooled liquid water amongst ice crystals. Overall,

these results illustrate the varying effects of aerosols, thermodynamics, and dynamics through various stages of mixed-phase cloud evolution based on this new method that categorizes cloud phases.

## 1 Introduction

Clouds with different thermodynamic phases can have contrasting influences on the net radiation at the top of the atmosphere, depending on their microphysical properties, spatial extent and the distributions of hydrometeors (Matus and L'Ecuyer, 2017).

Among three types of cloud phases (i.e., ice, liquid, and mixed), mixed-phase clouds contain both supercooled liquid water and ice crystals. Radiative forcing of mixed-phase clouds over the Southern Ocean has large impacts on Earth's climate based on global climate model simulations (e.g., Tan et al., 2016; Hyder et al., 2018). Evaluating and improving the model parameterizations of mixed-phase clouds requires an improved understanding of their macrophysical and microphysical properties, as well as the factors controlling their formation and evolution.

Previous observations of mixed-phase clouds in the high latitudes have identified complex structures both vertically and horizontally. Using aircraft-based observations over the Southern Ocean, a high frequency of supercooled liquid water was found within low-level clouds in this region, and mixed-phase cloud segments were found to be more spatially heterogeneous compared with the pure liquid and pure ice segments (D'Alessandro et al., 2021). When calculating cloud top phase frequencies as a function of cloud top temperature by using aircraft-based lidar and radar observations over the Southern Ocean, liquid

phase was seen as the dominant phase for 74.9% of the cloud top cases with subfreezing temperatures, and supercooled liquid water was found in cloud tops at temperatures as low as -30°C (Zaremba et al., 2020). Using a large dataset collected by the Convair 580 aircraft of the National Research Council (NRC) of Canada it was found that several microphysical properties are dependent upon temperature, including supercooled liquid droplet number concentration (Nliq), IWC, and LWC (Korolev et al., 2003).

Ice nucleation within mixed-phase clouds and the factors behind the sustainability of mixed-phase clouds are still topics of contention within the field. The persistent existence of mixed-phase cloud systems has been shown to be affected by local processes such as the formation and growth of cloud droplets and ice crystals (Morrison et al., 2012). The thermodynamics and dynamics of the atmosphere also play a large role in affecting the formation and development of mixed-phase clouds. Using observations of vertical motion within Arctic mixed-phase stratiform, Shupe et al. (2008) showed that an in-cloud

updraft sustains the clouds, which also supports growth of ice and liquid mass concentrations. Their results also suggest that ice crystal number concentrations (Nice) are often limited in order to support the persistent supercooled liquid water. The connection between ice formation and vertical air velocity at cloud base was examined for mixed-phase clouds with less than 380-m depth by using ground-based Doppler lidar and cloud radar, and the mass flux of IWC was found to increase by two orders of magnitude when the vertical velocity fluctuation increases (Bühl et al., 2019). A study analyzed generating cells of

ice crystals inside mixed-phase cloud layers over the Southern Ocean and found that these generating cells have small horizontal widths and contain supercooled liquid water with higher LWC and Nliq than that of the areas between the generating cells, which also held true for ice particles whose dispersions, number concentration, and sizes are larger within the generating cells (Wang et al., 2020). With seven years of ground-based observations at an Alaskan site, it was found that Arctic mixed-

phase clouds occur less often in the early fall when the winds are southerly as the atmosphere is more stable, drier, colder and has lower relative humidity. Conversely, during northerly winds they have wider particle distributions (Qiu et al., 2018).

Aerosols have been documented to influence the microphysical properties of mixed-phase clouds around the globe. Field study observations over a fourteen-year time period and from various locations around the Earth were combined to show that both temperature and the number concentration of aerosols larger than 0.5 μm in diameter can impact the concentrations of INPs in mixed-phase clouds (DeMott et al., 2010). From aircraft observations over the Arctic, it was found that entrainment above mixed-phase clouds could enhance Nice and aerosol thermodynamic indirect effect likely occurs (Jackson et al., 2012). Using a nine-year long aerosol dataset, Norgren et al. (2018) found that clean mixed-phase clouds with a lower aerosol loading have higher IWC at their base compared with clouds with a higher aerosol loading. Other studies over the Southern Ocean, e.g., McFarquhar et al. (2021), showed that those environments are primarily pristine, suggesting limited long-range continental aerosol transport and potentially more aerosols newly formed over the high southern latitudes. Observations and simulations of INPs showed that sea spray aerosol may play a major role in initiating primary ice nucleation in low-level mixed-phase clouds over the Southern Ocean (McCluskey et al., 2018). Besides primary ice production, secondary ice production has also been shown to be a critical process enhancing Nice in mixed-phase clouds based on both in-situ airborne observations (Huang et al., 2017; Järvinen et al., 2022) and global climate simulations (Zhao and Liu, 2021; Zhao et al., 2023) over the Southern Ocean. Secondary ice production can also be affected by aerosol loading, e.g., higher concentrations of cloud condensation nuclei can lead to higher supercooled liquid droplet concentrations, and therefore reducing the efficiency of the rime-splintering process.

These aforementioned studies demonstrated that the coexistence and interaction between supercooled liquid droplets and ice crystals hold a key for understanding the persistence of mixed-phase clouds despite of ice–liquid mixtures being unstable. An examination of aerosol indirect effects on liquid and ice hydrometeors separately is also a critical step towards a better understanding of the net aerosol indirect effects on the entire cloud (Korolev et al., 2017; Storelvmo, 2017). Targeting these topics, in this work, we develop a method to identify several phases of mixed-phase clouds, by using the spatial relationships among segments containing pure ice or liquid, as well as those containing both ice and liquid. In section 2, a description of the observation dataset and instruments is given. In section 3, the details of the identification of four phases, their occurrence frequencies, and comparisons with previously established mixed-phase cloud definitions are provided. A contrast of thermodynamic and dynamic conditions among these phases is shown. In addition, the relationships between macrophysical and microphysical properties of mixed-phase clouds during various phases are examined. Aerosol indirect effects from larger and smaller aerosols are quantified for individual phases. Lastly, in section 4, we discuss the applications of this method for contrasting different definitions of mixed-phase clouds, and the implications of model parameterizations.

## 2 Observational Dataset

### 2.1 SOCRATES In-situ Observations and Instrumentation

The U.S. National Science Foundation (NSF) Southern Ocean Clouds, Radiation, Aerosol Transport Experimental Study (SOCRATES) flight campaign was conducted from January 15[th] to February 24[th] in 2018 (McFarquhar et al., 2021). This NSF-funded campaign utilized the NSF/National Center for Atmospheric Research (NCAR) Gulfstream V (G-V) research aircraft which flew over the Southern Ocean region of 62°S–42°S and 133°E–164°E as shown in Figure 1. A total of fifteen research flights (RFs) in this campaign were performed with a combined total of 111 flight hours flown. In this work, we applied a temperature restriction of -40°C to 0°C, commonly known as the mixed-phase cloud regime as this temperature range allows for the occurrence of both ice particles and supercooled liquid water, for all our analyses.

The NSF G-V research aircraft during the SOCRATES campaign was equipped with scientific instruments to measure the various characteristics of the atmosphere, such as aerosol number concentrations (Na), cloud microphysical properties and common meteorological components – temperature, pressure, wind speed and humidity. The temperature was measured by the Rosemount temperature probe. To measure the water vapor molecule number density at 25-Hz resolution the Vertical Cavity Surface Emitting Laser (VCSEL) hygrometer was used. The final data reported the water vapor mixing ratio in 1-Hz resolution and a corrected version of water vapor data based on a post-campaign calibration in summer 2018 is used in this study (Diao, 2021). The water vapor and temperature data are used to calculate relative humidity with respect to liquid and ice ($RH_{liq}$ and $RH_i$), by using the equations for saturation vapor pressure with respect to liquid and ice from Murphy and Koop (2005), respectively. The uncertainties associated with $RH_{liq}$ and $RH_i$ originate from both water vapor and temperature measurements, which sum up to 6%–7% for the mixed-phase cloud regime. We placed a ceiling on RH values by restricting all $RH_{liq}$ greater than 101% to 101%. For $RH_{liq}$ lower than 100%, an adjustment to 100% is applied if two criteria are satisfied for a 1-Hz sample: 1) it contains supercooled liquid water and 2) either CDP or the King probe measures LWC greater than 0.001 g m$^{-3}$. The hydrometeor measurements used in this study were obtained from the Two-Dimensional Stereo Probe (2DS) and the cloud droplet probe (CDP), which have size ranges at 40 – 5000 μm and 2 – 50 μm, respectively. IWC and LWC are derived from 2DS and CDP probes following the method described in Yang et al. (2021). That is, a mass-Dimension relationship based on a spherical shape is used to calculate LWC for liquid droplets in both CDP and 2DS measurements. A mass-Dimension relationship based on Wu and McFarquhar (2016) is used to calculate IWC for ice particles in 2DS measurements. In-cloud conditions are defined as the 1-Hz measurements with total water content (TWC = IWC+LWC) greater than 0.001 g m$^{-3}$. Lower IWC and LWC values have also been reported by the two probes, but the threshold of 0.001 g m$^{-3}$ is chosen here due to the larger uncertainties of these cloud probes reporting lower mass concentrations of hydrometeors (e.g., Baumgardner et al., 2017). To provide a more focused analysis of cloud layers instead of precipitation below the clouds, we use two remote sensing instruments onboard the G-V aircraft – NSF/NCAR High-performance Instrumented Airborne Platform for Environmental Research (HIAPER) Cloud Radar (HCR) and High Spectral Resolution Lidar (HSRL) to identify potential precipitating samples. The particle identification (PID) product is used, which includes identifications of 11

categories – rain, supercooled rain, drizzle, supercooled drizzle, cloud liquid, supercooled cloud liquid, melting, large frozen, small frozen, precipitation and cloud (Romatschke and Vivekanandan, 2022). By manually inspecting hourly time series of this product, we remove segments that are identified as precipitation, supercooled drizzle, drizzle, supercooled rain, and rain.

In addition, we further examined the NSF SOCRATES campaign field catalogue for each flight to ensure that we do not miss any precipitation segments that have been identified in the field catalogue. The time stamps of the beginning and end of these segments are stored in supplemental Table S1. For most flights, we identified on average about 5 – 20 minutes of samples of precipitating regions, except RF15 which has about an hour of precipitating samples. It is worth noting that most of these precipitating segments occur at temperatures above 0°C, while this study only focuses on -40°C to 0°C.

Aerosol number concentration and size distribution are measured by the Ultra-High Sensitivity Aerosol Spectrometer (UHSAS) which has a size range of 60 – 1000 nanometers (nm). The vertical velocity measurements are derived from several instruments, including radome pressure, static pressure, Honeywell LASEREF IV Inertial Reference Unit, pitot tubes, temperature probe, and differential Global Positioning System, providing an accuracy of ~±0.15–0.30 m/s and precision ~0.01 m/s (Diao et al., 2015). When examining the in-cloud and clear-sky conditions in the SOCRATES campaign, we noticed a low

bias of the original vertical velocity measurements, and therefore applied a correction of +0.125 m/s for the vertical velocity values. After this correction, the peak of the frequency distributions of vertical velocity is centered at 0 m/s for both in-cloud and clear-sky conditions. To minimize the impacts of ascent and descent and the possible associated biases of vertical velocity measurements, we restrict the analysis of vertical velocity fluctuations (i.e., standard deviations of vertical velocity calculated for every 40 seconds) to segments where the maximum pressure change difference (dP) within 40 seconds is less than 10 hPa.

**2.2 Two Previous Datasets for Cloud and Hydrometeor Thermodynamic Phase Classifications**

For this work, two previously published datasets regarding thermodynamic phase classifications for the SOCRATES observations are used. Both datasets cover all research flights in the SOCRATES campaign with the exception of research flight 15 due to the malfunction of 2DS probe. The first dataset reports cloud phase (ice, liquid or mixed) at 1-Hz resolution, which was mainly derived from the 2DS and CDP cloud probes (Yang et al., 2021). That method used in Yang et al. (2021)

was built upon the study of D'Alessandro et al. (2019) and their figure 1. The cloud phase identification was also verified by other cloud probes, such as the King probe for detecting LWC, and the Rosemount Icing Detector for detecting the existence of supercooled liquid droplets by freezing them when they collide with the detector, which subsequently changes the vibration frequency of the detector. Two modifications are applied to the previous cloud phase identification method of D'Alessandro et al. (2019) and Yang et al. (2021). The first modification is that only when CDP measurements are categorized as liquid

droplets, these samples are used in the analysis. Measurements categorized by CDP as ice particles are excluded since previous work has shown that these measurements related to counting ice are most likely artifacts (e.g., Korolev et al., 2013). The second modification is about the treatment of large particles identified as liquid droplets. The previous method restricts particles with maximum dimensions ($D_{max}$) > 312.5 μm as ice particles, while those with $D_{max}$ between 112.5 and 312.5 μm can be either liquid or ice depending on the standard deviation of particle sizes measured by 2DS in that second. In this work,

we further restrict particles with $D_{max} > 212.5$ µm to be ice particles, reducing the number of large particles being categorized as liquid droplets.

The second dataset that detects individual hydrometeor's thermodynamic phase (either ice or liquid) is also used, which was produced by the University of Washington with the Ice-Liquid Discriminator (UWILD) through a machine learning approach (Atlas et al., 2021; Mohrmann et al., 2021). Each particle imaged by the 2DS probe is classified particle-by-particle into ice, liquid or unclassified, as 0, 1 and NaN, respectively. In this dataset, the group also provides 1-Hz aggregated data for each research flight that include a quantification of phase-separated particle size distributions (PSDs). We use the hydrometeor count defined by the maximum diameter in the UWILD dataset to calculate Nliq and Nice detected by the 2DS probe within each second. Then we further add Nliq detected by CDP to those detected by 2DS to derive the total Nliq. Finally, we define ice particle number fraction (IPNF), which equals Nice / (Nice + Nliq) in one second.

## 3 Results

### 3.1 A Method to Classify Four Phases of Mixed-Phase Clouds

A method to classify four phases of mixed-phase clouds is developed for 1-Hz aircraft-based observations, which mainly involves two steps. In the first step, each second of observations are categorized into four conditions, including a second of clear-sky condition, liquid cloud region (LCR), ice cloud region (ICR), or mixed-phase cloud region (MCR). LCR is defined as a one-second sample where only supercooled liquid droplets were observed, while ICR is defined as a one-second sample with only ice crystals. MCR is a one-second sample with occurrence of both ice and liquid. Here the identification of liquid and ice within each second of observations is based on the 1-Hz cloud phase identification method modified from D'Alessandro et al. (2019) and Yang et al. (2021) as described in Section 2.2.

In the second step, a total cloud region (TCR) that can potentially contain multiple seconds with a combination of LCR, ICR and MCR is identified, which basically is a consecutive in-cloud segment surrounded by clear-sky conditions. In other words, LCR, ICR and MCR are defined at the scale of each second, while TCR is defined at the scale of a consecutive in-cloud segment which can contain more than one second. If a TCR sample is surrounded by two adjacent seconds of NaN, then this sample is deleted, because one cannot determine if the NaN points are the edge of the cloud or if they are still part of the cloud. But if a TCR sample is surrounded by two adjacent seconds of clear-sky samples, then this in-cloud sample is valid, and its measurement can last from one second to many seconds. An illustration of the identification of TCR is shown in Figure 2. In that example, 1 second of LCR, 2 seconds of MCR, and 4 seconds of ICR are adjacent to each other. Then the 1 LCR sample, 2 MCR samples, and 4 ICR samples all belong to the same TCR, which produces a total of 7 seconds of samples. All the 1-Hz samples within the TCR are used in the analysis in Sections 3.3 – 3.8 (i.e., Figure 4 a and b, Figures 5 – 10). The length of each second of sample within an TCR is calculated based on the aircraft true air speed at that specific second. The length of each TCR is calculated as the sum of all in-cloud samples within that TCR. The mean true air speed of the G-V research aircraft between -40°C and 0°C during the SOCRATES campaign is ~172 m/s.

Within each TCR, the spatial ratios of LCR, MCR, and ICR relative to TCR are calculated. The definitions of each phase are based on these spatial ratios as described in Table 1. The number of one-second samples and the number of cloud segments for four phases are summarized. Following the calculation of these spatial ratios, four phases are defined as follows: (1) only LCR appears in the TCR, (2) MCR exists by itself or coexists with LCR, but no ICR exists, (3) ICR appears and it either resides with LCR, MCR, or both, (4) only ICR appears in the TCR. In other words, phases 1 and 4 stand for pure liquid and pure ice cloud segments, respectively. Phase 2 represents those ice crystals embedded in MCR and surrounded by supercooled liquid droplets. Phase 3 represents the stage when pockets of pure ice segments start to appear. The four phases are depicted in a conceptual diagram in Figure 3.

The spatial distributions of LCR, MCR and ICR are also related to the two types of mixed-phase clouds – genuinely versus conditionally mixed, separated by the level of mixing between supercooled liquid water and ice crystals (e.g., Korolev et al., 2017, their Fig. 5-1; Korolev and Milbrandt, 2022, their Fig. 1). The scenario of "LCR+ICR" indicates a sequence of spatially adjacent cloud segments …-ice-liquid-ice-liquid-…., which is considered a conditionally mixed-phase cloud as a sub-category of phase 3. Such clouds may be thermodynamically stable, and their lifetime would be determined by processes other than the interaction between ice and liquid (e.g., WBF and riming). This special scenario when only "LCR+ICR" exist in the TCR without the existence of MCR has 840 seconds of samples, which is a small fraction of the total 11988 seconds of phase 3 samples. This result suggests that most of the clouds with coexisting supercooled liquid water and ice particles at least contain some partial segments of genuinely mixed phase, i.e., MCR.

To investigate the possibility of misclassifying MCR as LCR due to the relatively lower number concentrations of ice particles compared with supercooled liquid droplets in a one-second sampling volume, distributions of mass and number concentrations of ice crystals are examined against those of supercooled liquid droplets (not shown). When liquid and ice coexist, the majority of the 1-second samples have both IWC > 0.01 g m$^{-3}$ and LWC > 0.01 g m$^{-3}$. In addition, the mass concentrations and number concentrations of ice and liquid are positively correlated with each other. This indicates that when ice and liquid coexist, most likely both types of hydrometeors have significant mass and number concentrations. Thus, it is less likely that the smaller sampling volume for ice crystals would lead to a misclassification of MCR as LCR. It is possible though, that some pure ICR pockets with very low number concentrations of ice crystals may be missing.

### 3.2 Relationships of Four Phases to Potential Evolution Pathways of Mixed-Phase Clouds

Several potential evolution pathways have been documented and discussed in previous literature, which can be linked with the separation of the four phases described above. A "classical" type of evolution pathway follows phases (1)=>(2)=>(3)=>(4), which was observed and documented over 35 years ago (e.g., Hobbs and Rangno, 1985). This type of evolution describes the situation that a cloud is initiated as liquid phase under supercooled conditions; then it experiences ice nucleation and turns into mixed-phase; after that some section of the mixed-phase cloud glaciates and turns into ice; and in the final stage, the entire cloud is glaciated. Besides the classical progression of mixed-phase, there are two other routes of evolution of mixed-phase clouds. The first "non-classical" pathway is when, after nucleation of INPs and turning liquid clouds into mixed-phase, all ice

particles precipitate out of the clouds, turning the mixed-phase back into liquid. In other words, the thermodynamic phase evolution of such clouds can be described as liquid => mixed-phase => liquid, i.e., phases (1)=>(2)=>(1). The imbalance between the water vapor supply and the bulk ice mass crystal growth, required for the maintenance of mixed-phase clouds, was discussed in Rauber and Tokay (1991), Pinto (1998), and Westbrook and Illingworth (2011). There is a fair amount of modelling attempts to find an explanation of maintenance of mixed-phase clouds through the balance of INPs and dynamic

forcing (e.g., Avramov et al., 2011; Fan et al., 2009, 2011; Smith et al., 2009). The second "non-classical" pathway of mixed-phase evolution is related to the generation of mixed-phase clouds in a pre-existing ice cloud due to dynamic forcing, which can be presented as ice=>mixed-phase, i.e., phases (4)=>(2), or (4)=>(3)=>(2). Note that the numerical order of phases 1 – 4 does not necessarily represent the evolution direction as indicated by arrows in Figure 3. For example, phase 4 may either be the final stage in the classical pathway, whereas in the second non-classical pathway, phase 4 is an initial stage. The theoretical

basis explaining such process was developed in several previous studies (e.g., Korolev and Mazin, 2003; Korolev and Field, 2008, Field et al., 2014; Hill et al., 2014). These studies were supported by earlier observations of mixed-phase clouds embedded in pre-existing, deep ice clouds (e.g., Hogan et al., 2002; Field et al., 2004). We caution that a mixed-phase cloud may or may not follow these exact pathways in the real atmosphere, as certain phases may be skipped, the evolution direction could be reversed, and multiple phases can appear in the same cloud in a 3-D view. Nevertheless, this method provides a

statistical separation of the cloud phases and allows a more focused analysis of the coexistence of supercooled liquid water and ice crystals that cannot be achieved if a one-second sample is analyzed without the context of its surrounding conditions, for instance, if a one-second LCR is part of a pure liquid cloud segment or is surround by MCR or ICR.

After defining the four phases, the following sections will examine the thermodynamic (i.e., temperature and relative humidity) and dynamic conditions of four phases (Figures 4 and 5), the macro- and microphysical properties, as well as their correlations

with each other. For macrophysical properties of mixed-phase clouds, we focus on investigating the lengths of cloud segments (Figure 4 c and d) and the spatial fraction of a cloud segment containing ice (Figures 7 – 10). For microphysical properties of mixed-phase clouds, we focus on investigating particle size distributions (Figure 6), fraction of number concentrations containing ice (Figure 7) and mass concentrations of supercooled liquid droplets and ice crystals (Figure 8).

### 3.3 Distributions of Four Phases at Various Temperatures

Figure 4 a and b show the number of 1-second samples for each phase as well as their probability among all phases within 5-degree temperature bins. The results show that phases 1 and 4 are more dominant at higher and lower temperatures, respectively, which follows the basic thermodynamic process that the phase change from liquid to ice phase occurs more frequently at lower temperatures. At temperatures between -20°C and -5°C, phase 2 is the most dominant phase and contributes to 40% of the total samples, while phase 3 contributes to 20% – 40% of the total samples. The fact that the pure ice or liquid

phase only contribute to 5% – 35% of the total samples between -20°C and -5°C demonstrates that the cloud segments sampled in the SOCRATES campaign are spatially heterogeneous, consistent with the results in the previous study of D'Alessandro et al. (2021). Figure 4 c and d show the distributions of the length of TCRs in four phases. The distribution of TCR lengths is

consistent with the previously observed power-law distribution of cloud horizontal sizes shown in Wood and Field (2011). The lengths of TCR segments vary from ~0.2 – 180 km in various temperature ranges, with low sampling statistics (i.e., less than 100) of continuous in-cloud segments longer than 60 km, which indicates a patchy horizontal structure with clear-sky gaps inside the clouds. Since the 1-D aircraft sampling can be at any vertical level relative to a cloud layer, we further examine the impacts of restricting the analysis to different ranges of LWC, IWC, and $RH_i$ values (supplementary Figures S1 and S2). Previous studies such as Wang et al. (2012) and D'Alessandro et al. (2023) have shown that cloud top usually contains higher LWC than cloud base, while IWC increases from the cloud top to cloud base. By using different ranges of LWC and IWC as proxies for vertical levels within cloud layers, we found that the number of samples of the four phases are relatively similar unless very high LWC or IWC are used (> 0.1 g m$^{-3}$).

The impact of length scales of TCR on the phase distributions is examined in supplemental Figure S3. TCR samples are separated into four scales – 0.1 – 1 km, 1 – 10 km, 10 – 100 km, and > 100 km. The dependence on temperature for the distributions of four phases is consistently seen for various scales, e.g., phase 1 has more samples at higher temperatures, while phase 4 has more samples at lower temperatures. Comparing the shorter (panels a and b) and longer (c and d) TCR samples, the shorter ones have more samples in phase 1 (i.e., pure liquid phase), while the longer ones have more phases 2 and 3. For the length scales of phase 4, more 1-second samples in phase 4 were found in shorter segments (0.1 – 1 km) at -20 to 0°C, while more 1-second samples in phase 4 were found in longer segments (10 – 100 km) at -40 to -20°C. This result indicates that the coexistence of ice and liquid occurs more frequently in clouds with larger spatial extent, such as stratocumulus and stratus clouds.

### 3.4 Thermodynamic and Dynamic Effects on the Evolution of Mixed-phase Clouds

Thermodynamic and dynamic conditions of each phase are examined at various temperatures in Figure 5, which shows the distributions of $RH_i$, vertical velocity, and standard deviation of vertical velocity ($\sigma_w$, calculated for every 40 seconds). For phases 2 and 3, LCR represents seconds without ice particles, while MCR and ICR represent seconds with ice particles. These two conditions (i.e., without or without ice) are separately examined in Figure 5 e – l. For 1-Hz samples dominated by supercooled liquid water (i.e., the entire phases 1 and 2, and phase 3 samples without ice), RH values are distributed closely to the liquid saturation line. This is consistent with previous theoretical and observational studies (Korolev and Mazin, 2003; Korolev and Isaac, 2006), which showed that $RH_{liq}$ in mixed-phase clouds is close to 100%, due to evaporating droplets rapidly via the Wegener-Bergeron-Findeisen (WBF) process, bringing the system of "droplets-water vapor" to quasi-equilibrium and therefore saturating the environment. As liquid droplets glaciate into ice particles, the peak of RH frequency would also shift towards ice saturation (e.g., D'Alessandro et al., 2019), as shown by the wider range of $RH_i$ in 1-Hz samples containing ice in phase 3 (Figure 5 g). The in-cloud samples used in this study contain a small amount of sub-saturated conditions that deviate from liquid saturation in phases 1 – 3, with phase 1 showing the least amount of liquid sub-saturation compared with other phases. These liquid sub-saturated conditions may be attributed to a combination of reasons, such as 6%–7% uncertainties in RH values originated from water vapor and temperature measurement uncertainties, heterogeneous distributions of LCR, MCR

and ICR that lead to an uneven distribution of supercooled liquid water, as well as non-equilibrated states between vapor/liquid or vapor/ice phase due to a larger volume being sampled by fast aircraft measurements (~172 m horizontal resolution for 1-Hz measurements used here). For all four phases, $RH_i$ values above ice saturation and closer to liquid saturation have been seen, providing observational evidence that new formation of supercooled liquid water droplets and ice crystals may occur in any of the four phases, following either of the three evolution pathways mentioned in Section 3.2.

Probability density functions (PDFs) of $RH_i$, $RH_{liq}$, vertical velocity, and $\sigma_w$ are further examined in Figure 5 i – l. The peak frequencies of $RH_{liq}$ are seen at liquid saturation for phases 1 – 3, consistent with the findings in Figure 5 a – d. The PDFs of vertical velocity show higher frequencies of updrafts for phases 2 and 3 compared with phases 1 and 4. In addition, PDFs of $\sigma_w$ show higher frequencies of large $\sigma_w$ values in phases 2 and 3 than phases 1 and 4. The number of 1-Hz $\sigma_w$ samples at various ranges (i.e., $\geq 0.5$ m/s, $\geq 1$ m/s, and $\geq 1.25$ m/s) and their percentages relative to the total samples in each phase are shown in supplemental Table S2. That analysis also shows higher percentages of larger $\sigma_w$ values in phases 2 and 3 compared with phases 1 and 4. Similarly, the distributions of $\sigma_w$ as a function of temperature in supplemental Figure S4 show more samples above 1 m/s across a wide range of temperatures from -36°C to 0°C in phases 2 and 3 than phases 1 and 4. These results indicate that the segments containing both supercooled liquid droplets and ice particles are subject to relatively stronger updrafts and more turbulent conditions, compared with the segments containing only liquid droplets or only ice crystals. This finding is consistent with Shupe et al. (2008) which pointed out the importance of updrafts for sustaining mixed-phase clouds. Differing from the previous studies, our method can further specify that the highest updrafts and the highest vertical velocity fluctuations are both found in phase 3 when pure ice segments start to appear (~4.5 m/s in Figure 5 k and ~2.3 m/s in Figure 5 l), consistent with the fact that $RH_{liq}$ deviates more from liquid saturation in phase 3 (Figure 5 c), and therefore higher updrafts would be required to maintain supercooled liquid droplets.

**3.5 Particle Size Distributions in Four Phases of Mixed-Phase Clouds**

The PSD for four phases is shown in Figure 6, separately plotted for the 2DS and CDP probes. Phases 1, 2 and 3 have similar concentrations of small liquid droplets between 2 – 10 µm. Phase 2 has the highest concentrations of hydrometeors at 10 – 60 µm, while phase 3 has the highest concentrations at 60 – 3000 µm. Phase 4 also has relatively high concentrations of ice crystals at 200 – 3000 µm, but they are lower than the values from phase 3 by a factor of 5 – 10. The decreasing ice crystal concentrations per size bin from phase 3 to phase 4 may be caused by stronger aggregation, sublimation, and/or sedimentation of ice crystals in phase 4, as well as by stronger glaciation and/or secondary ice production in phase 3. The significant decrease (1 to 4 orders of magnitude) of hydrometeor concentrations per size bin at 20 – 100 µm in phase 4 compared with the other three phases suggests that most supercooled liquid water may have evaporated and transitioned into ice phase through WBF process or riming, instead of the freezing of individual droplets, while the small ice crystals may have sublimated. It is possible that some of the phase 4 samples may represent the trails of generating cells, where the growth is aloft, and sublimation is at the lower part of the cloud layer. In addition, smaller supercooled liquid droplets require lower temperatures to freeze into ice crystals. This feature is also shown in supplemental Figure S5 d from -10°C to -40°C, as small ice crystals at 20 – 200 µm size

range show increasing concentrations with decreasing temperatures in Phase 4. On the other hand, phase 3, which still has supercooled liquid water coexisting with ice particles, does not show such trend, probably because ice crystal formation and growth may occur via various processes in phase 3, such as secondary ice production, WBF process, glaciation, and/or riming. Phases 3 and 4 in Figure S5 c and d show a trend of decreasing frequency of large ice particles (e.g., $D_{max} > 1000$ µm) with decreasing temperature. This could be due to an increasing probability of droplet freezing with decreasing temperature given the same dimension that reduces the available amount of large supercooled liquid droplets for glaciation or riming at lower temperatures.

### 3.6 Relationship between Microphysical and Macrophysical Properties of Mixed-phase Clouds

One unique contribution of this work is to quantify how cloud microphysical and macrophysical properties are correlated with each other. The relationship between cloud macrophysical properties (represented by mixed or ice spatial ratio) and several microphysical properties are further examined, including IPNF (Figure 7), as well as LWC, IWC and ice mass fraction (Figure 8). Specifically, the mixed spatial ratio represents the fraction of MCR as part of an individual, consecutive TCR, calculated as length of MCR / length of TCR. Ice spatial ratio represents the fraction of ice-containing segments as part of an individual consecutive TCR, calculated as (length of ICR + length of MCR × IWC/TWC) / length of TCR. The contribution of MCR to ice spatial ratio in phase 3 is weighted by the ice mass fraction, giving the MCR a smaller weighting function compared with ICR since MCR contains higher fractions of supercooled liquid droplets than ICR. Note that the definitions of mixed spatial ratio and ice spatial ratio differ from the spatial ratio previously used for characterization of mixed-phase clouds in Korolev et al. (2017, Fig.5-13a). In that previous method, the spatial ratio of a thermodynamic phase (i.e., liquid, mixed or ice) is calculated as the number of samples of that thermodynamic phase divided by the total cloud samples in a certain temperature bin. In this work, the mixed spatial ratio and ice spatial ratio are calculated for individual TCR segments, and therefore each TCR would produce one value for mixed spatial ratio and one value of ice spatial ratio. These values of mixed spatial ratio or ice spatial ratio are applied to every 1-second sample within this TCR.

In Figures 7 and 8, linear regressions of the mixed spatial ratio and ice spatial ratio against each microphysical property are shown for phases 2 and 3, respectively. The analysis is separated by whether the one-second sample is an LCR, MCR or ICR. The slope value (b) of the linear regression is provided in the text legend. Since phase 2 does not contain ICR, no data points are shown in those sub-panels in Figures 7, 8 and 10. The linear regression analysis is applied to the average values of microphysical properties in each spatial ratio bin, in order to assign an equal weight to each bin of mixed or ice spatial ratio. When directly applying the linear regressions analysis to individual seconds of IPNF (as shown in supplemental Figure S6), similar slope values are seen compared with Figure 7, but the bins of mixed spatial ratio and ice spatial ratio have uneven distributions of samples.

Note that additional quality control procedures are applied to the IPNF data, because the machine-learning based particle identifications of 2DS data may misidentify small ice fragments as supercooled liquid droplets, especially at lower temperatures. To minimize such misidentifications, the following two quality control procedures are applied, which are

developed after inspecting the Particle Habit Imaging and Polar Scattering (PHIPS) airborne cloud probe: (1) for 1-Hz samples of ICR in phase 3 and 4, when temperatures are below -20°C and 0 < IPNF < 1, IPNF is reset to 1 to be pure ice; (2) for 1-Hz samples of ICR in phase 3, when temperatures are between -20 and -10°C and 0.4 < IPNF < 1, these IPNF values are reset to

1. After the corrections, out of 2866 seconds of samples analyzed in Figure 7 b, 172 seconds (i.e., 6.00%) show IPNF > 0.1. All regions (i.e., LCR, MCR and ICR) in Figure 7 show positive correlations between IPNF and mixed or ice spatial ratio in phases 2 and 3. This means that while ice crystals gradually dominate the total particle population (i.e., IPNF increases) in cloud segments, the spatial fraction containing ice particles (i.e., MCR+ICR) also approaches 1 from a macroscopic perspective. Comparing phase 2 and 3, phase 2 (without ICRs) shows smaller positive correlation (b values of 0.009 and 0.013)

compared with phase 3 (b values of 0.561, 0.026, and 0.469). This is because when ice particles are surrounded by supercooled liquid droplets, the latter has a much higher number concentration than ice crystals and therefore IPNFs are relatively low on average in phase 2. On the other hand, in phase 3, ice crystals start to become the dominant particles by number concentration and supercooled liquid droplets become less dominant. Note even after quality control is applied to IPNF, a small amount of high IPNF values is still seen (e.g., 0.4 ≤ IPNF < 1) in Figure 7 b and f. A sensitivity test is conducted by removing all 0.4 ≤

IPNF < 1 (not shown), which shows consistent conclusions as Figure 7, that is, all phases show positive correlations between IPNF and the spatial expansion of ice-containing regions. In addition, phase 3 still shows higher slopes of linear regressions compared with phase 2, indicating faster increases of IPNF in phase 3 when pure ice segments start to appear.

Previously, Wang et al. (2020) used airborne remote sensing measurements from the SOCRATES campaign to identify generating cells of ice crystals. Based on the definition from American Meteorological Society (2024), generating cells are

defined as cloud-top regions with high radar reflectivity, which often produce fall streaks of falling hydrometeors. Out of the 16 cases of generating cells detected by Wang et al. (2020), all 16 cases contain supercooled liquid droplets. The average LWC and Nliq inside generating cells were found to be greater than those outside the generating cells. In addition, larger ice particles and higher Nice were seen in the generating cells, associated with the updrafts inside the cells. These reported generating cells are also analyzed in Figure 7, with the average IPNF values shown in each mixed and ice spatial ratio bin. The generating cells

associated with LCR and MCR contain lower IPNF (Figure 7 a – d). This is because when generating cells are associated with high concentrations of supercooled liquid droplets, Nice may be lower than Nliq, which leads to the lower IPNF. But when the generating cells are associated with ICR, significantly higher IPNF (close to 1) are seen for most ice spatial ratio bins (Figure 7 f). This result suggests that not all regions within the generating cells experience significant phase change from liquid to ice, unless the ice-containing regions become dominated by ice.

Figure 8 shows the correlations of LWC and IWC with respect to mixed spatial ratio or ice spatial ratio. A clear negative slope is seen in Figure 8 a–d, indicating that as the mixed spatial ratio or ice spatial ratio increases, the LWC decreases. On the contrary, a positive trend is seen in Figure 8 e, f, and h, indicating increasing IWC with increasing mixed or ice spatial ratio. These results are consistent with the analysis of IPNF, showing that the increasing dominance of ice crystals in both mass and number concentrations is correlated with the increasing spatial ratio of ice-containing regions in cloud segments. Slope values

in Figure 8 illustrate that LWC decreases more significantly in LCR (b = -0.460) than MCR (b = -0.055) in phase 2. Phase 3

shows a more significant decrease of LWC by a factor of 3 compared with phase 2, with slope values b = -1.694 and -0.692 in LCR and MCR, respectively. For the changes of IWC, the slope values are similar between MCR and ICR in phase 3 (b = 1.358 and 1.261, respectively), while the slope value of IWC is slightly lower for MCR in phase 2 (b = 0.969).

Figure 8 i and j show the positive correlations of ice mass fraction with respect to mixed spatial ratio or ice spatial ratio for the entire phase 2 and phase 3, respectively. Ice mass fraction increases more rapidly with increasing spatial fraction of ice-containing regions in phase 3 than phase 2, with slope values of 1.013 and 0.238, respectively. This result indicates that when ice crystals first appear in MCR, the mass partitioning is still dominated by the liquid phase. As ice crystals grow into pure ice segments (i.e., ICR), liquid phase starts to rapidly evolve into ice phase. This result indicates that even though ice and supercooled liquid water coexist throughout the lifetime of mixed-phase clouds, the partition between them has different rates of phase change during different phases.

**3.7 Comparisons of Three Methods to Define Cloud Thermodynamic Phases**

Thermodynamic phases of an in-cloud sample can be defined based on the relative dominance of ice crystals and supercooled liquid water. Three parameters are evaluated here – the ice spatial ratio that represents the macrophysical property of a TCR, ice mass fraction per second, and IPNF per second. The frequency distributions of these three metrics are shown for four phases (Figure 9 a – c) as well as for all in-cloud conditions (Figure 9 d – f). The results show all three parameters have a bi-modal distribution that peaks at 0 and 1, indicating that most of the cloud segments are either dominated by liquid or ice, and few of them have similar amounts of liquid and ice, regardless in a macrophysical or microphysical perspective. The number of samples associated with each parameter and cloud phases is shown in supplemental Figure S7.

Using these three parameters, the distributions of three cloud thermodynamic phases (i.e., ice, liquid, and mixed) are compared among three types of definitions, including (i) cloud phases defined by the ice spatial ratio within each TCR using the method developed in this work. Liquid, mixed, and ice phases are defined as where the ice spatial ratio of an entire TCR is < 0.1, 0.1 – 0.9 and > 0.9, respectively; (ii) the 1-Hz cloud phase distribution defined by the ice mass fraction (i.e., IWC/TWC) derived for 1-second observations, i.e., liquid, mixed and ice phases defined as ice mass fraction < 0.1, 0.1 – 0.9 and > 0.9, respectively; This method of using ice mass fraction to define mixed-phase clouds has been used in the cloud physics community for approximately thirty years (e.g., Korolev, 1998; Korolev et al., 2017, their equation 5-1 and references therein); and (iii) cloud phase distribution defined by the majority of the hydrometeors by particle number concentrations using the combined CDP and 2DS data, i.e., liquid (ice) phase defined as a second of data with more than 90% (less than 10%) of hydrometeor particle number concentrations being liquid droplets, and mixed phase defined as a second of data with 10% – 90% of particle number concentrations being liquid droplets. To summarize, each of these three types of methods relies on a certain type of fraction of ice crystals relative to the total hydrometeors, either in terms of the spatial fraction relative to the entire cloud segment, or in terms of 1-Hz mass fraction or 1-Hz particle number fraction. This concept of using various ice fractions to define cloud thermodynamic phases has been summarized in the previous review article by Korolev et al. (2017).

Figure 9 g – i shows the occurrence frequencies of cloud thermodynamic phases in relation to temperature compared among three types of definitions. The results show that all three methods have similar distributions of three cloud thermodynamic phases at temperatures from 0 to -40°C, with the two definitions using ice mass fraction per second and ice number particle fraction per second being even closer to each other. For temperatures between -20°C and 0°C, the ice spatial ratio method has slightly higher mixed phase frequency (0.1 – 0.2) than the ice mass fraction and IPNF methods (~0.05 – 0.1). Overall, all three methods show a significant transition from liquid to ice phase at a similar temperature around -17.5°C. This indicates that the major transition from liquid to ice is reflected in both cloud microphysical (i.e., mass partition and number partition) and macrophysical properties (spatial extent partition). The rapid increase of occurrence of ice clouds in the temperature range of -15°C to -20°C was also observed by previous studies (e.g., Wallace and Hobbs, 1977; Moss and Johnson, 1994).

## 3.8 Aerosol Indirect Effects on the Evolution of Mixed-phase Clouds

The relationship between aerosol number concentration and mixed spatial ratio or ice spatial ratio is examined in Figure 10. Due to the possible complication of in-cloud measurements of aerosol number concentrations, we applied a moving average to calculate logarithmic scales of clear-sky aerosol concentrations at every 50 seconds in Figure 10. Furthermore, the average aerosol concentration is only analysed if more than half of the entire 50 seconds satisfy the criteria of in-cloud conditions. A coarser spatial averaging using the 100-second moving average of clear-sky conditions of every 100 seconds is also shown in supplementary Figure S8.

Number concentrations of larger aerosols (diameters > 500 nm, namely $N_{>500}$) and smaller aerosols (diameters > 100 nm, namely $N_{>100}$) are analyzed in Figure 10 a – h and i – p, respectively. The slope values of the linear regressions show strong positive correlations between $N_{>500}$ and ice spatial ratio in phase 2 (Figure 10 g, b = 1.534), when ice crystals just start to appear and are surrounded by supercooled liquid droplets. Such positive correlation becomes weaker in phase 3 (Figure 10 h, b = 0.944), when ICR starts to appear. The stronger positive correlation with $N_{>500}$ in phase 2 is likely due to primary ice nucleation (such as heterogeneous nucleation) playing a major role in phase 2 when ice crystals first start to appear. On the other hand, secondary ice production may occur more frequently in phase 3, and secondary ice production via rime-splintering is less effective when concentrations of cloud condensation nuclei are higher. For the correlations with $N_{>100}$, a positive trend is still seen with respect to ice spatial ratio in MCR and ICR, indicating possible pathways of ice formation via condensation freezing and immersion freezing assisted by smaller aerosols. Overall, the weaker positive correlations with $N_{>100}$ in MCR and ICR compared with $N_{>500}$ indicates that larger aerosols play a more dominant role for initiating ice nucleation than smaller aerosols. Stronger positive correlations between IWC and $N_{>500}$ compared with $N_{>100}$ are also shown in the previous work by Yang et al. (2021), although that study did not differentiate the four phases of clouds nor examine aerosol indirect effects in relation to cloud macrophysical properties, i.e., the spatial expansion of ice-containing cloud segments.

**5 Discussion and Conclusions**

Mixed-phase clouds are ubiquitous in the atmosphere and in order to fully capture their extent of impacts on Earth's climate,
more studies need to be conducted in order to investigate their formation, evolution, and aerosol effects on their microphysical and macrophysical characteristics. Therefore, in this study, a novel method that categorizes mixed-phase clouds into four phases was presented, which represent different conditions of partition between liquid and ice. This method allows an investigation on cloud macrophysical and microphysical properties as well as the related aerosol indirect effects at different levels of partitioning between supercooled liquid water and ice particles, as the phase change occurs among vapor, liquid, and
solid phase of water molecules.

The relationships between microphysical and macrophysical properties are examined, which addresses the question of whether the dominance of ice crystals in hydrometeor mass or number concentration also leads to the dominance of ice-containing regions in a consecutive in-cloud segment. Two spatial extent parameters – mixed spatial ratio and ice spatial ratio – are used to quantify the spatial distributions of hydrometeors within supercooled liquid water-dominant and ice-dominant mixed-phase
clouds. Positive correlations of IPNF and IWC in relation to mixed spatial ratio and ice spatial ratio are seen in both phases 2 and 3, respectively. Comparing phases 2 and 3, the latter phase shows higher rates of changes in four microphysical properties with increasing ice spatial ratio, including faster increase of IPNF, faster increase of IWC, faster decrease of LWC, and faster increase of ice mass fraction (Figures 7 and 8). These results indicate that when ice crystals become more dominant and pure ice segments start to appear, both the mass and number partitions between liquid phase and ice phase experience a higher rate
of phase change with respect to the spatial ratio of ice-containing regions (note that this rate of change is not with respect to time).

The correlations between various cloud macro- and microphysical properties are further demonstrated by using three methods to define ice, liquid, and mixed phases. Following the generic definition of mixed-phase clouds described in Korolev (1998) and Korolev et al. (2017), $\mu_{ice} = \alpha_{ice} / (\alpha_{ice} + \alpha_{liq})$, where $\mu_{ice}$ is ice phase fraction, and $\alpha_{ice}$ and $\alpha_{liq}$ are specific cloud microphysical
properties. We examined $\alpha_{ice}$ being ice mass fraction or IPNF at 1-Hz resolution, but also extended the definition to include $\alpha_{ice}$ being ice spatial ratio in a consecutive cloud segment, which is a macrophysical property that has not been investigated before. All three methods follow the same thresholds of $< 0.1$, $0.1 - 0.9$, and $> 0.9$ to separate $\mu_{ice}$ into liquid, mixed and ice phases, respectively. As a result, all three methods identify a significant transition from liquid to ice around a similar temperature at -17.5°C. A minor difference among three methods is that mixed-phase cloud frequency between -20°C to 0°C
is slightly higher when defined by ice number fraction and ice spatial ratio ($0.1 - 0.2$) compared with that defined by ice mass fraction and IPNF ($0.05 - 0.1$). Such comparison on various phase definition methods indicates that a spatial extent-based cloud phase identification method, such as using number of pixels in remote sensing data by Yip et al. (2021), Desai et al. (2023), and Wang et al. (2024), can produce similar statistical distributions of liquid and ice phases compared with other methods based on ice mass fraction, e.g., D'Alessandro et al. (2019) and Yang et al. (2021), while the spatial extent-based
method produces a slightly higher mixed-phase cloud frequency. Future analysis of cloud phase distributions based on different

types of observation techniques and model simulations is recommended to consider this comparison result, especially when evaluating model output against observations using different definitions of mixed-phase clouds.

Differing from previous studies on the coexistence of ice crystals and supercooled liquid water, the method presented in this work allows one to compare the cloud segments when ice crystals are surrounded by supercooled liquid water in MCR with those when pure ICR starts to appear. Aerosol indirect effects on mixed-phase clouds during different levels of phase partitioning can also be examined separately. In both phases 2 and 3, $N_{>500}$ show stronger positive correlations with mixed spatial ratio and ice spatial ratio compared with $N_{>100}$. This indicates that the larger aerosols are more likely to act as INPs to initiate primary ice nucleation. Phase 3 shows a slightly weaker positive correlation of ice spatial ratio with aerosol number concentrations (i.e., $N_{>500}$ and $N_{>100}$) compared with phase 2, indicating that the aerosol indirect effects are more prominent when ice crystals first start to appear amongst supercooled liquid water in MCR. Such weaker aerosol indirect effects in phase 3 are possibly due to a competition between the positive correlation of primary ice nucleation with aerosol number concentrations and the negative correlation of secondary ice production with aerosol number concentrations. When pure ice segments (ICR) start to appear, it is possible that secondary ice production plays a more important role and therefore the net aerosol indirect effects become weaker.

Thermodynamic and dynamic conditions are examined for each phase, especially for the key stage of mixed-phase clouds – the maintenance of supercooled liquid droplets when they coexist with ice. Previously, several dynamic mechanisms were proposed in the study of Korolev and Field (2008), highlighting the critical thresholds of vertical motion for sustaining supercooled liquid water. Our analysis shows that both higher updrafts and stronger in-cloud turbulence are more frequently observed in phases containing both ice and liquid (i.e., phases 2 and 3) compared with pure liquid or pure ice phase (i.e., phases 1 and 4, respectively). Even higher updrafts and turbulence are seen in phase 3 when pure ice segments start to appear, compared with phase 2 with only mixed-phase segments, indicating that higher updrafts are needed to sustain supercooled liquid water when they are surrounded by ice-dominated segments. This observation-based method can be used to assess the contribution from different dynamic mechanisms for maintaining different evolution stages of mixed-phase clouds in various field campaigns.

Parameterizations of mixed-phase clouds in climate models often rely on a tunable parameter that can modify the mixing volume between ice and liquid (e.g., Tan and Storelvmo, 2016; Zhang et al., 2019). In other words, the model parameterization assumes that when ice crystals are mixed more uniformly amongst supercooled liquid water within a model grid box (i.e., when mixed spatial ratio or ice spatial ratio is higher), the WBF process become more effective and the transition from liquid to ice would be faster. This study illustrates that the mass and number partitioning between liquid and ice hydrometeors in mixed-phase clouds are not only correlated with the mixed spatial ratio or ice spatial ratio which reflects the spatial fraction of ice-containing regions, but also are correlated with the existence of pure ice segments (Figures 7 and 8). Future model parameterization is recommended to quantify the varying rates of phase change throughout a cloud's lifetime by considering two main factors – the type of phases (especially phase 2 versus phase 3 depending on the existence of pure ice segments) and the spatial fraction of ice-containing region.

Overall, the method proposed in this work provides a unique perspective to assess mixed-phase cloud properties in both macrophysical and microphysical perspectives, especially for phases when supercooled liquid droplets and ice particles coexist. Such partition can be reflected in particle number fraction, mass fraction, and spatial ratio. We note that this is an idealized method with its own caveats. For example, the evolution of mixed-phase clouds may not always follow a simple direction from phase 1 to 4. In addition, the aircraft observations used here only capture the 1-D structure of a cloud segment, while cloud layers above and below the aircraft flight track may show a different ice spatial ratio on a 2-D or 3-D view. Nevertheless, this method helps to provide a statistical categorization of different phases of mixed-phase clouds solely based on Eulerian-view sampling of aircraft data. Future studies may derive such statistical distributions of phases based on 2-D remote sensing observations and 3-D model simulations. Examining individual phases of mixed-phase clouds may also provide more direct comparisons between observations and simulations.

**Data availability**

Observations from the NSF SOCRATES campaign are accessible at https://data.eol.ucar.edu/.

**Author contributions**

F. Maciel, M. Diao, and C.A. Yang contributed to the development of the ideas, conducted quality control to aircraft-based observations, and conducted data analysis. F. Maciel and M. Diao wrote the manuscript.

**Competing interests**

The authors declare that they have no conflict of interest.

**Acknowledgments**

F. Maciel, C.A. Yang and M. Diao acknowledge funding from NSF OPP #1744965. M. Diao acknowledges the funding support from U.S. Department of Energy (DOE) Atmospheric System Research (ASR) grant DE-SC0021211 and RDPP grant DE-SC0023155. C.A. Yang and M. Diao acknowledge funding from SJSU Division of Research and Innovation award number 22-LUG-08-006. F. Maciel and C.A. Yang also acknowledges support from the San Jose State University Walker Fellowship.

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

**Table 1.** Definitions of four phases of mixed-phase clouds based on ratios of lengths of LCR, MCR, and ICR over the length of TCR within a consecutive cloud segment, i.e., $\frac{L_{LCR}}{L_{TCR}}$, $\frac{L_{MCR}}{L_{TCR}}$, and $\frac{L_{ICR}}{L_{TCR}}$, respectively.

| Phase | Description | Number of 1-second samples | Number of TCR segments | Spatial Ratio of LCR | Spatial Ratio of MCR | Spatial Ratio of ICR |
|---|---|---|---|---|---|---|
| 1 | Only LCR | 8243 | 1163 | $\frac{L_{LCR}}{L_{TCR}} = 1$ | $\frac{L_{MCR}}{L_{TCR}} = 0$ | $\frac{L_{ICR}}{L_{TCR}} = 0$ |
| 2 | MCR appears | 12557 (LCR: 11096, MCR: 1461) | 142 | $0 \le \frac{L_{LCR}}{L_{TCR}} < 1$ | $0 < \frac{L_{MCR}}{L_{TCR}} \le 1$ | $\frac{L_{ICR}}{L_{TCR}} = 0$ |
| 3 | Pure ICR must appear | 11988 (LCR: 3478, MCR: 2973, ICR: 5537) | 249 | $0 \le \frac{L_{LCR}}{L_{TCR}} < 1$ | $0 \le \frac{L_{MCR}}{L_{TCR}} < 1$ | $0 < \frac{L_{ICR}}{L_{TCR}} < 1$ |
| 4 | Only ICR | 8646 | 1193 | $\frac{L_{LCR}}{L_{TCR}} = 0$ | $\frac{L_{MCR}}{L_{TCR}} = 0$ | $\frac{L_{ICR}}{L_{TCR}} = 1$ |

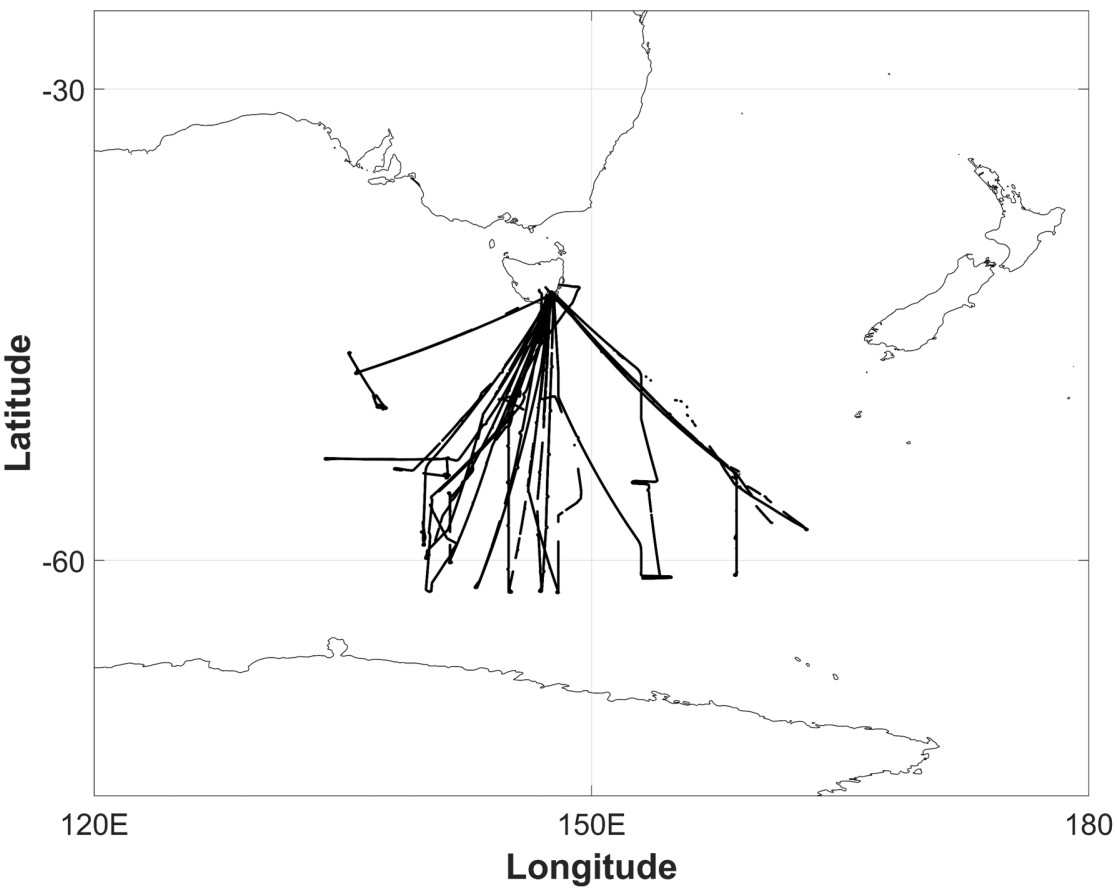

**Figure 1.** Map of the flight tracks for SOCRATES for only temperatures between 0°C and -40°C.


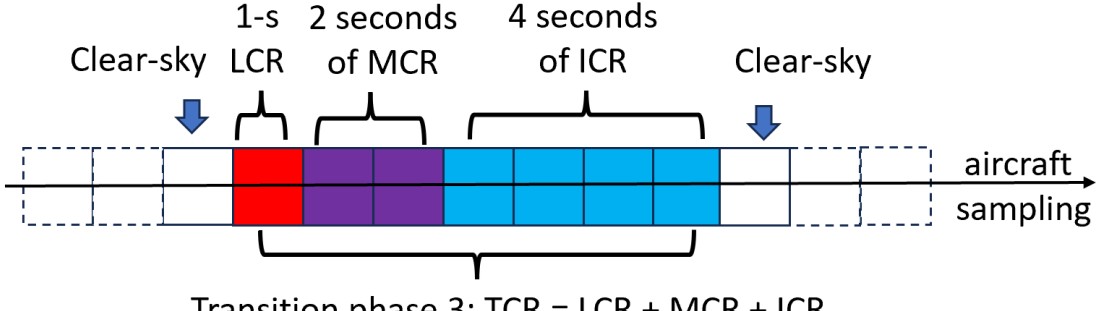

**Figure 2.** A schematic diagram that illustrates the identification of a total cloud region (TCR) sample, with 1 second of LCR (red), 2 seconds of MCR (purple), and 4 seconds of ICR (blue) embedded inside this TCR. All 7 seconds of samples inside this TCR are used in the following analysis of cloud properties.


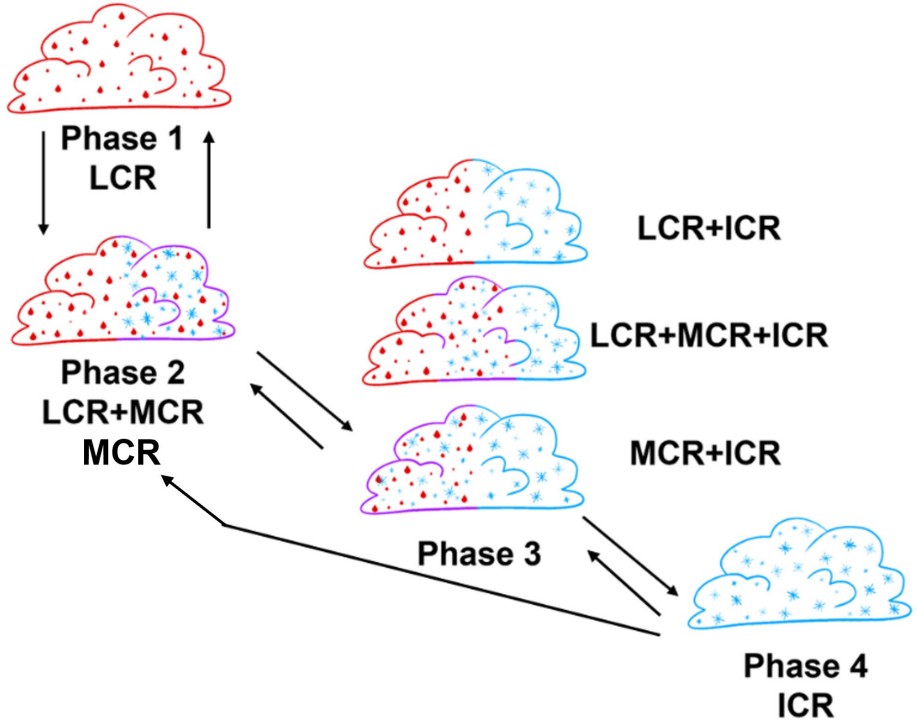

**Figure 3.** A conceptual diagram of the four phases for the phase exchange between supercooled liquid water and ice particles in mixed-phase clouds. Red, blue, and purple shading indicates liquid cloud region (LCR), ice cloud region (ICR) and mixed cloud region (MCR), respectively.


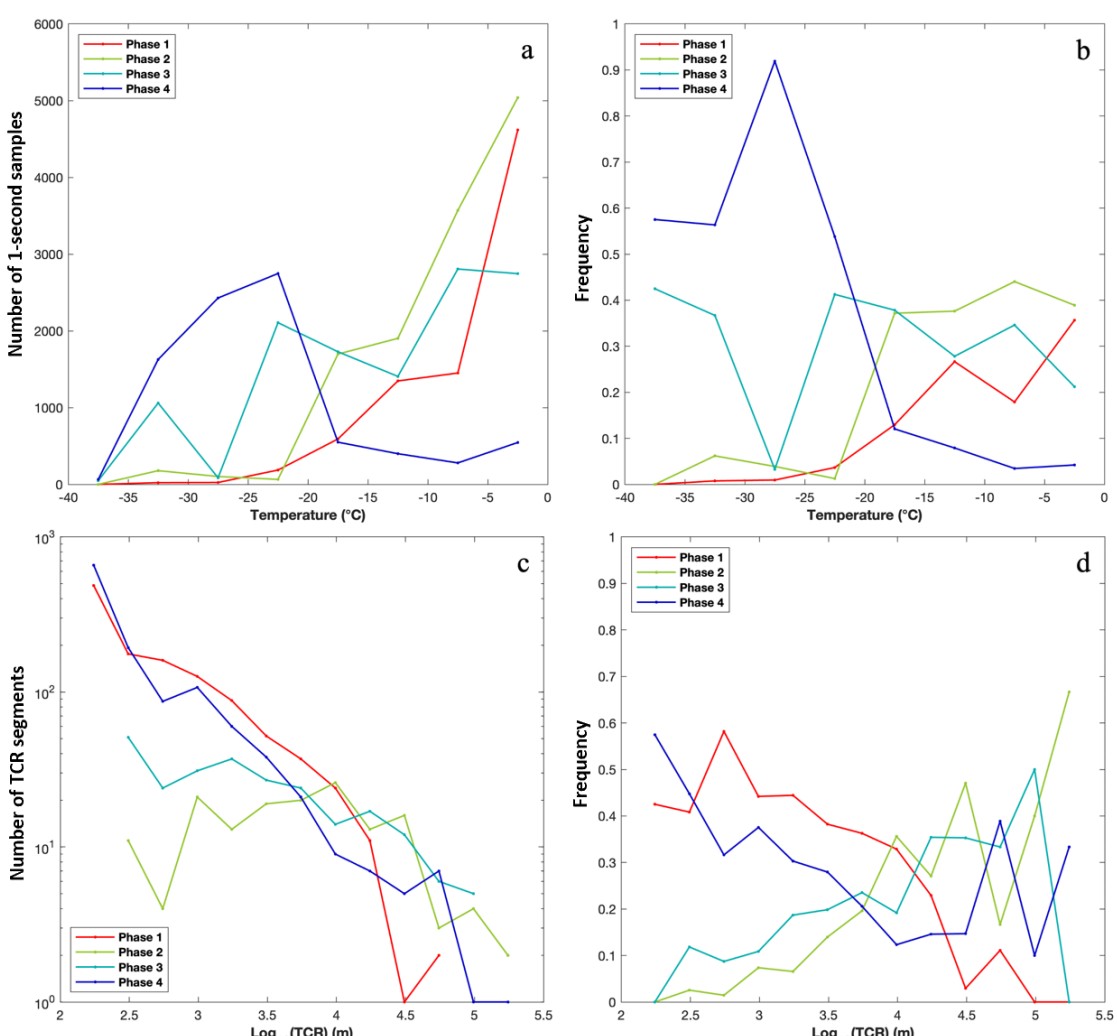

**Figure 4.** Distributions of 1-Hz samples in four phases at various temperatures in the top row. (a) Number of 1-second samples and (b) frequency of 1-second samples in each phase within various temperature bins. In (b), the frequency of 1-second samples in each phase is normalized by the total number of 1-second samples of all phases in each 5-degree temperature bin. Distributions of various lengths of TCR segments are analysed in the bottom row. (c) Number of TCR segments and (d) frequency of cloud segments in each phase associated with various lengths in log10-scale. In (d), frequency is calculated as the number of segments of a specific phase divided by the total number of segments in each $10^{0.25}$ bin.

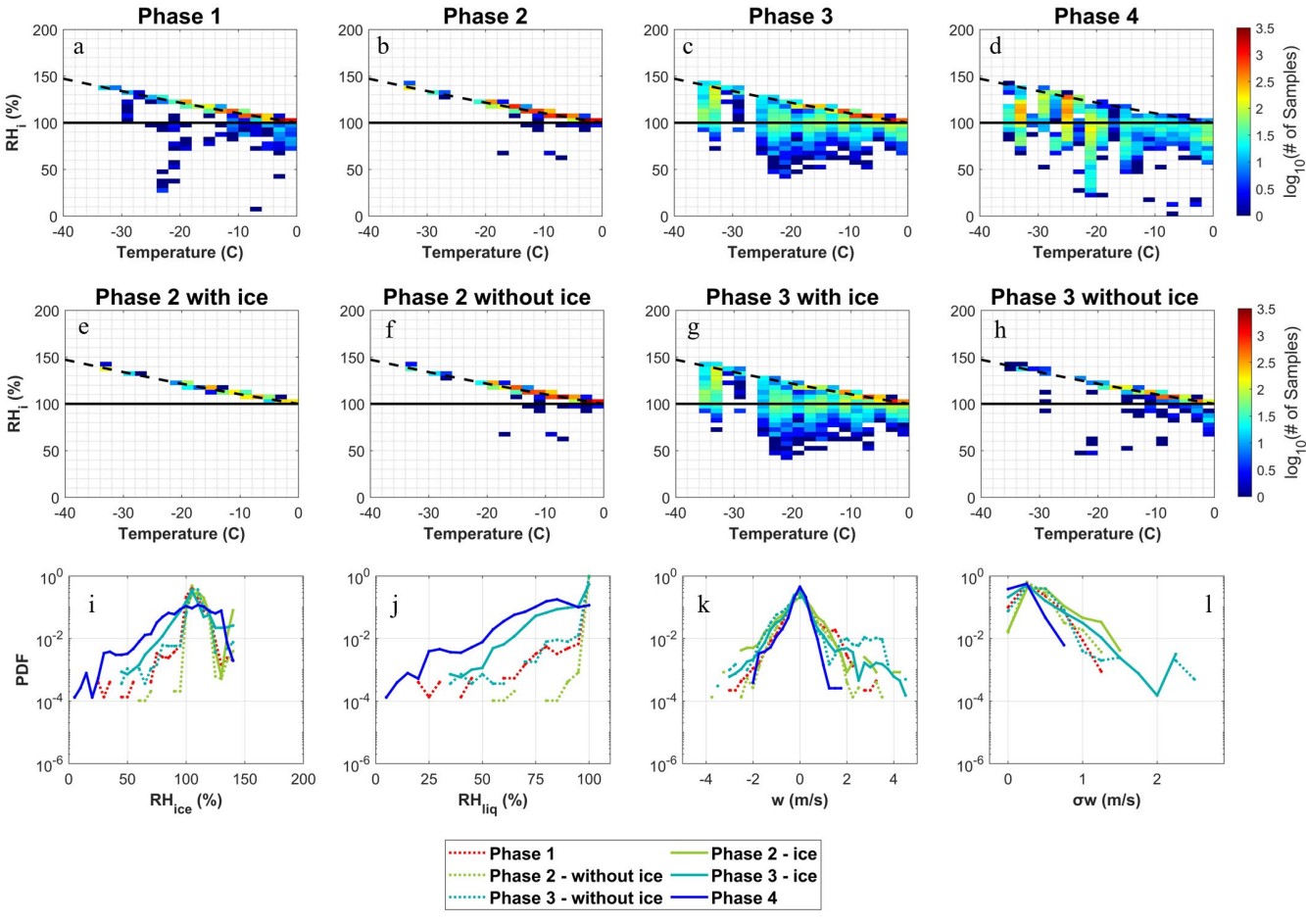


**Figure 5.** (a-h) Distributions of $RH_i$ as a function of temperature. (i-l) PDFs of $RH_i$, $RH_{liq}$, vertical velocity (w) and $\sigma_w$ of various phases, respectively. Dashed lines in (a) – (h) indicate liquid saturation.

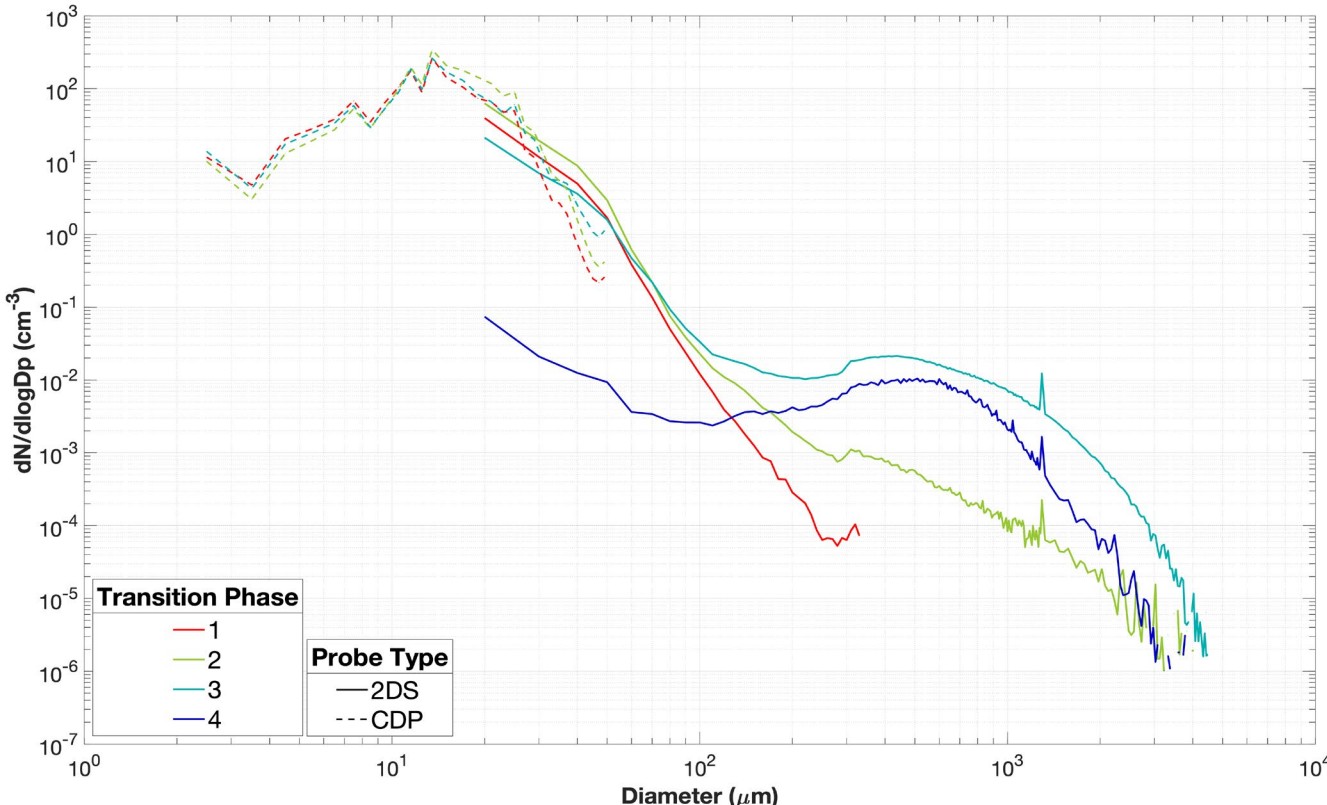

**Figure 6.** Particle size distribution of the four phases for mixed-phase clouds separated by probe types. The entire dataset at the temperature range of -40°C to 0°C is shown. Phase 4 only shows 2DS measurements because ice particles measured by CDP are excluded from the analysis.

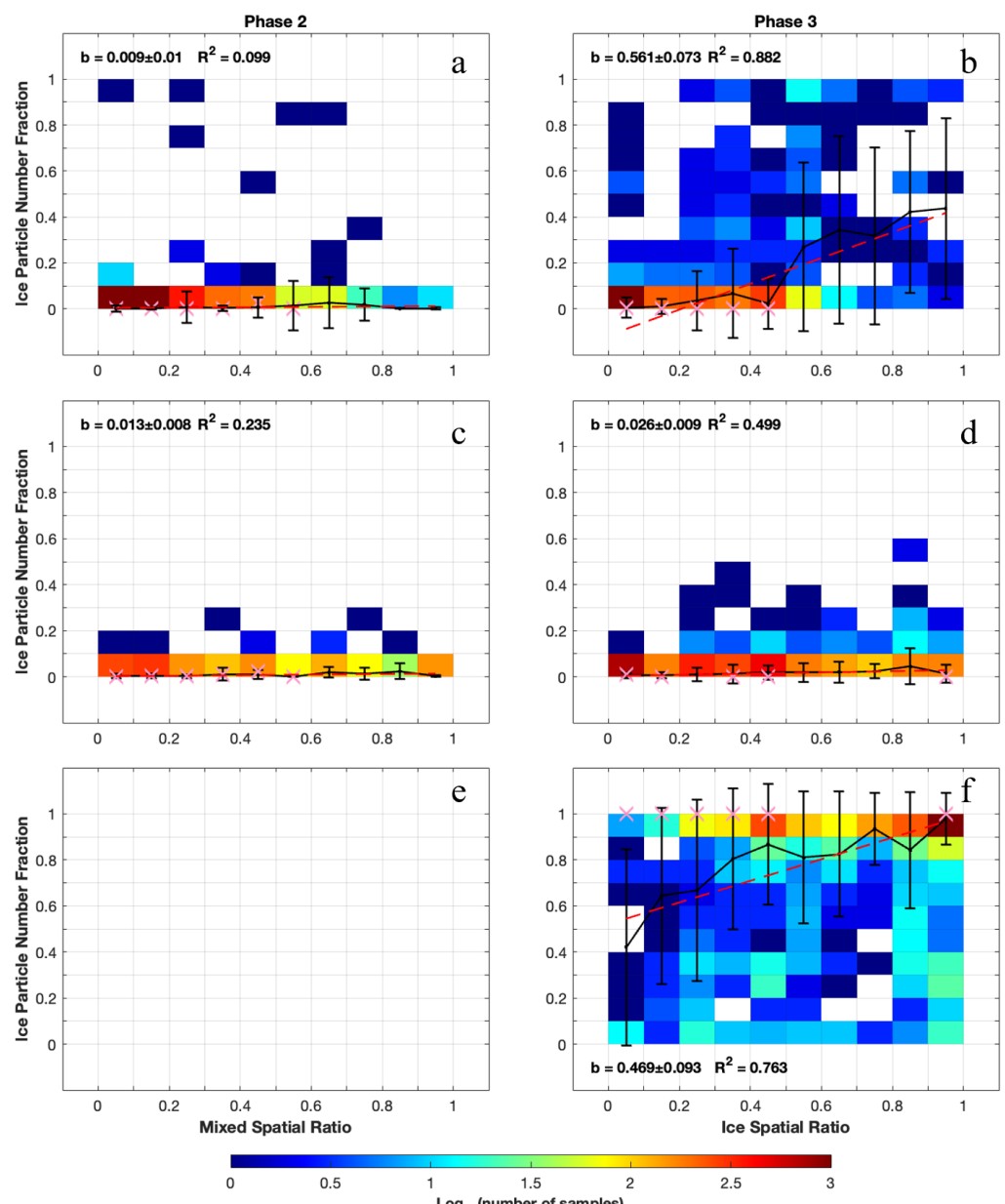

**Figure 7.** Relationship between ice particle number fraction (IPNF) and mixed spatial ratio or ice spatial ratio, separated by the phase type (phase 2 in column 1 and phase 3 in column 2) and by various cloud segments – (a, b) LCR, (c, d) MCR and (e, f) ICR. Average values for each ice spatial ratio bin are shown in black solid lines, with vertical bars representing standard deviations. Linear fit is shown in red dashed line. Average values of generating cells (time series obtained from Wang et al. (2020)) are in pink "X" markers. The slope value b, its associated standard deviation, and the ordinary R-squared value are shown in the legend.

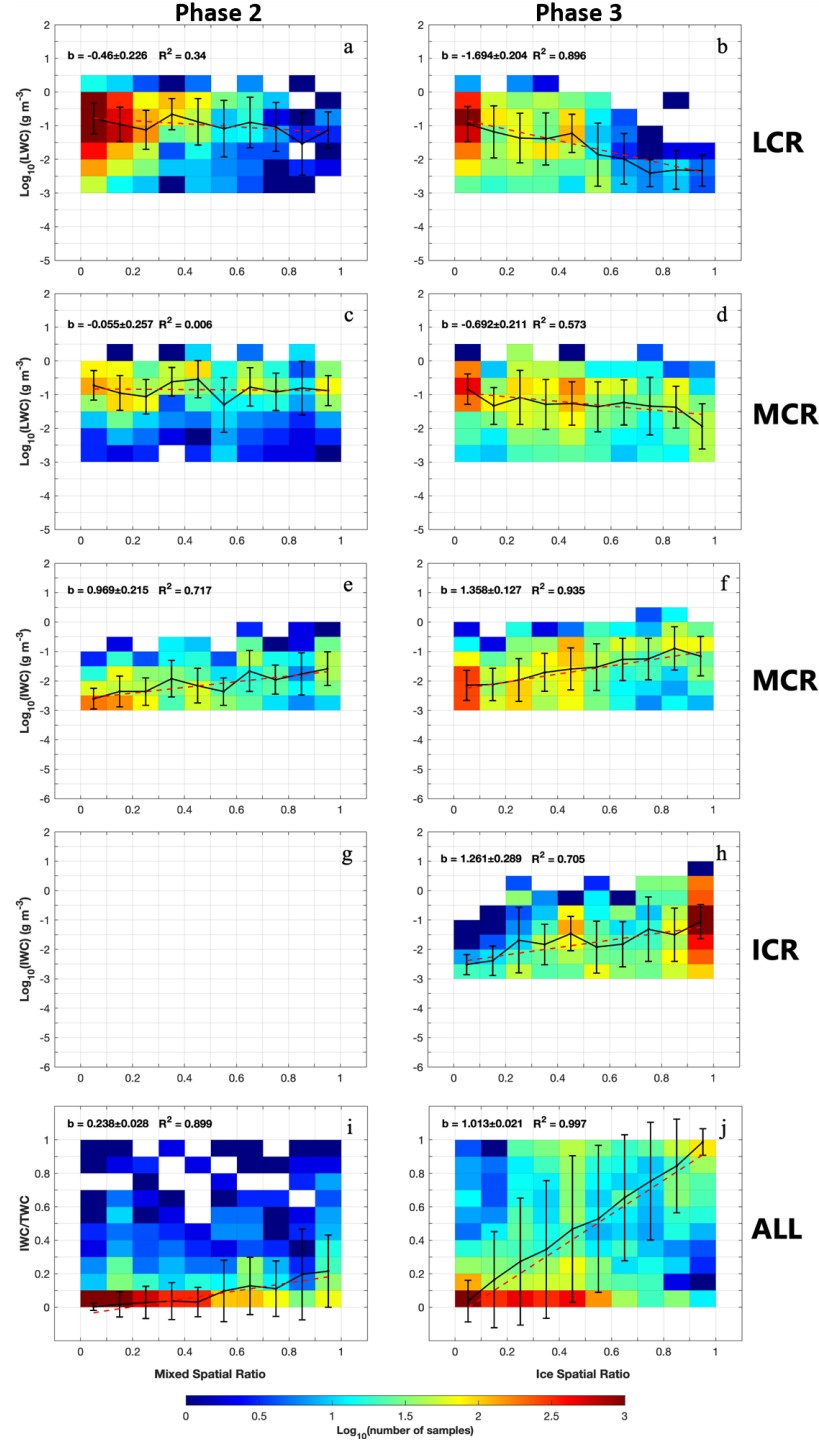

**Figure 8.** Similar to Figure 7, but showing (a-d) LWC (unit: g m$^{-3}$), (e-h) IWC (unit: g m$^{-3}$), and (i and j) ice mass fraction in relation to mixed spatial ratio for phase 2 and ice spatial ratio for phase 3, separated by the phase type and cloud regions.

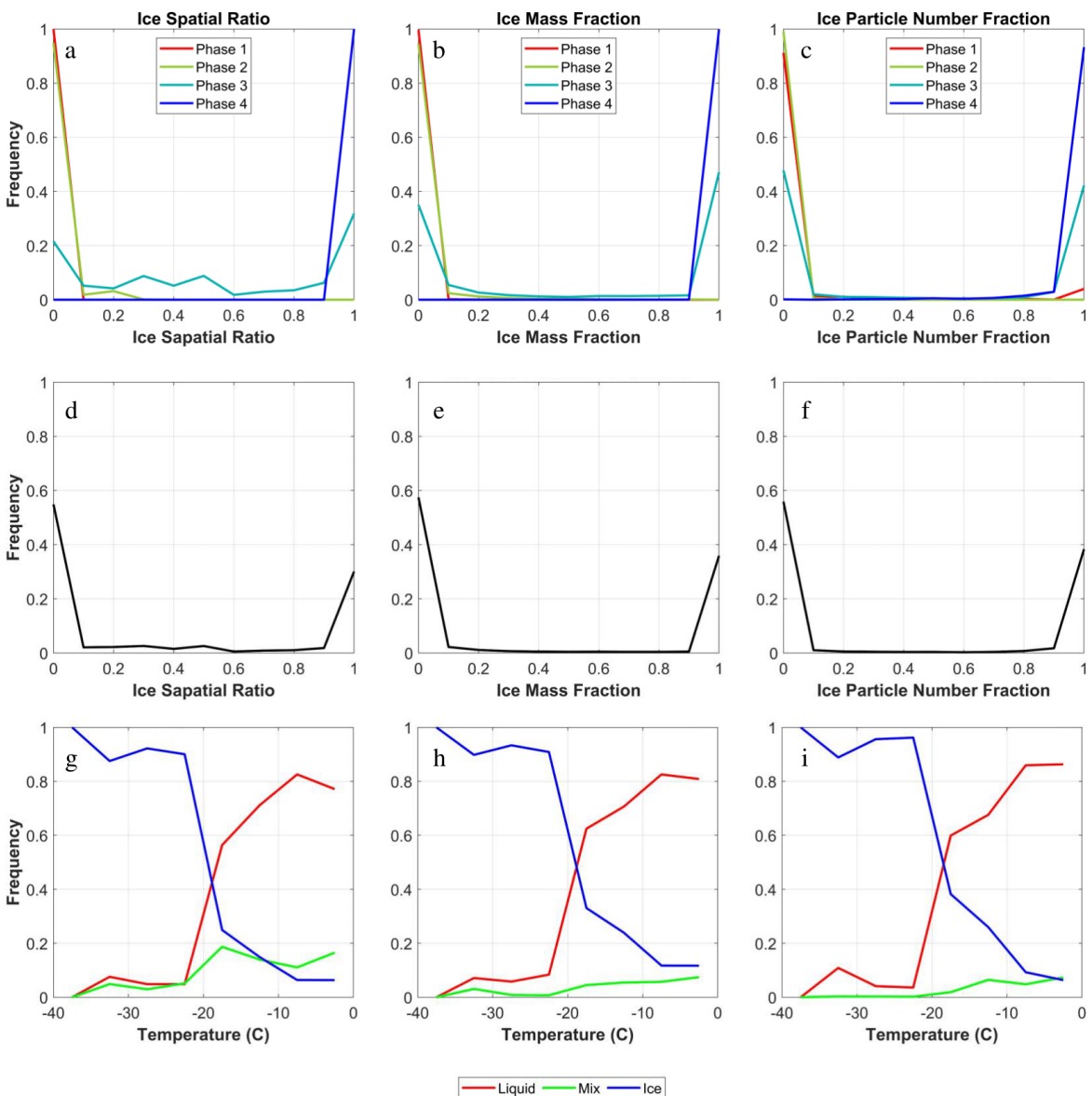

**Figure 9.** Frequency distributions of (a) ice spatial ratios calculated for individual consecutive TCR, (b) ice mass fraction per second, and (c) IPNF per second for four phases. (d-f) Similar to (a-c), but for the four phases combined representing the entire in-cloud conditions. (g-i) cloud phase frequency distributions defined based on the respective parameter in each column.

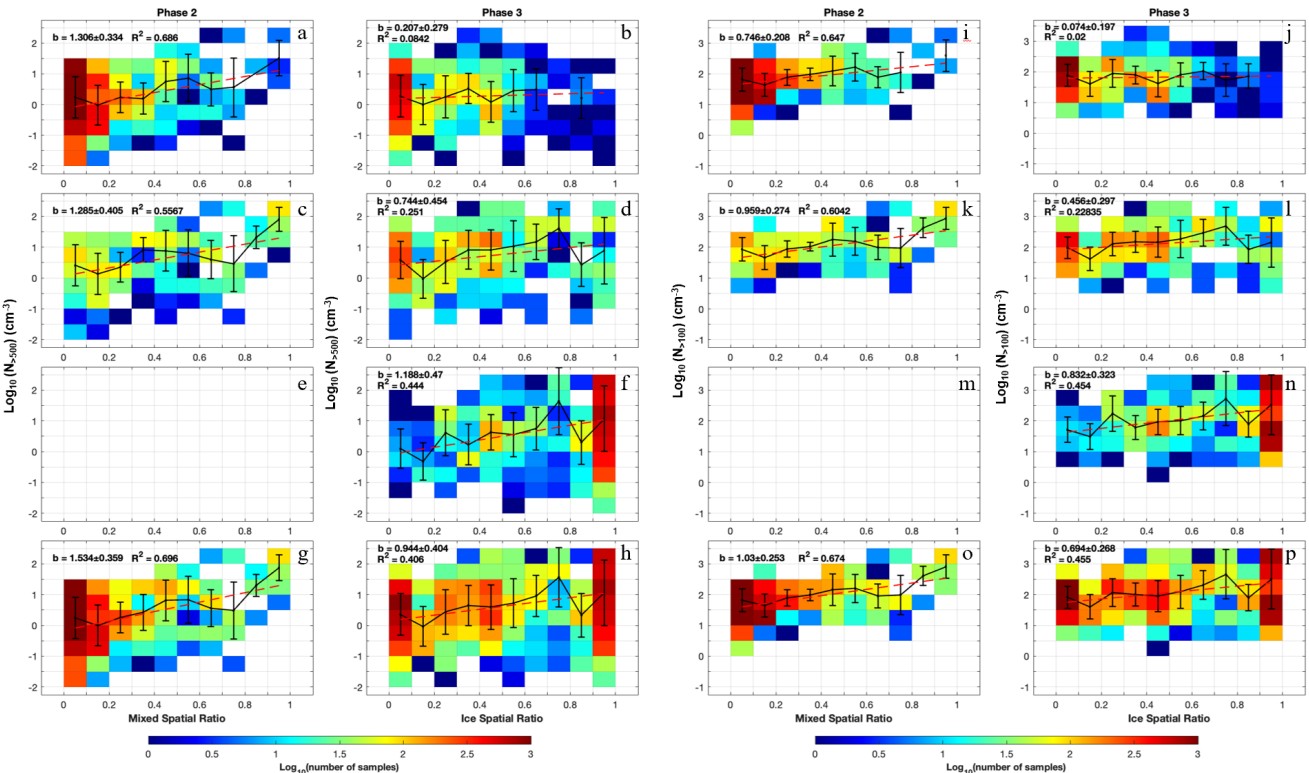

**Figure 10.** Similar to Figure 7, but showing logarithmic scale (a-h) $N_{>500}$ and (i-p) $N_{>100}$ in relation to mixed spatial ratio or ice spatial ratio, separated by the phases and cloud regions. The first, second, and third rows represent LCR, MCR, and ICR, respectively. The last row represents all cloud regions in a specific phase. The aerosol number concentrations represent the moving average values of every 50 seconds.