# Peer review of "Partition between Supercooled Liquid Droplets and Ice Crystals in Mixed-phase Clouds based on Airborne In-situ Observations"

_Atmospheric Measurement Techniques, 2022_

## Referee Comment (RC1)

Review of "The Transition from Supercooled Liquid Water to Ice Crystals in Mixed-phase Clouds based on Airborne In-situ Observations", by Maciel and Diao.

This is an interesting and I'd say novel study that characterizes the degree of glaciation of supercooled clouds in the Southern Ocean, using data from the SOCRATES field program. They use a combination of "bulk" data from the 2DS and CDP probes, augmented with the King liquid water content and Rosemont Icing Probes for phase discrimination. The methodology is based on Yang et al. (2021), which is based on D'Alessandro et al. (2019). A second method uses the 2DS images and a machine learning tool to identify individual images of liquid and ice particles. The methodology and some of the results bear a striking similarity to the Yang et al. (2021) article. This includes the relationship of the phase partitioning to the aerosol concentration. Some mention of the results of that study should be given and how the techniques and results differ.

The partitioning of the data into liquid only, mostly liquid, mostly ice and all ice phases is interesting, and relating it to the macrophysical properties of the cloud layer (ice content, relative humidity with respect to water and ice, and vertical velocity is interesting. Likewise, the ice fraction partitioning is rather interesting.

I tried to think of the factors that determine the glaciation of a stratiform cloud layer. First, in stratiform ice cloud layers, typically liquid water regions, the stronger vertical motions, and ice nucleation mostly occur at cloud top. Below cloud top and depending on the vertical velocity which is responsible for the degree of ice supersaturation or subsaturation, can contain a growth or sublimation layer. The measurements you report on do not account for where vertically within the cloud layer relative to cloud top the aircraft penetrations were made. Thus, you may be sampling in a subsident or upward moving parcel of air. Perhaps your measurements at temperatures below -15°C are near cloud top, and those at the warmer temperatures in the middle or lower parts of the cloud layers. The relationship of the relative humidity to turbulence might be a manifestation of where in the cloud layer the sampling is done, and whether generating cells were penetrated. You do mention that the aircraft observations only captures the 1-D structure of a cloud segment, while cloud layers above and below the aircraft flight track may show a different ice spatial ratio on a 2-D or 3-D view. Nonetheless, I think a weakness of the study is that there is no partitioning of where in the cloud layer the measurements are made. (Note that it's unlikely that the vertical motions measured by the aircraft system are sufficiently accurate to determine what zone the measurements are made in, unless generating cells are penetrated).

Figure 10, which shows the distributions of RHi, RHliq, vertical velocity and standard deviation of vertical velocity which be presented before Figure 3 as it provides a context for how the various transition regions relate to the large-scale properties of the cloud layer. What is shows is that transition region 4 is subsaturated. This why there is no liquid water in that region, not a stage of development of the ice cloud. It also shows up in the PSDs-transition region 4 has fewer ice particles and smaller maximum ice particle sizes than transition region 3. These might

be the trails of generating cells, where the growth is aloft and sublimation lower down in the cloud layer.

**Minor comments**

Line 6: determines the "ice cloud lifetime" 45: resilience to what?

62: "growing"?

66: aerosol

97: remove "various"

183: I don't think the CDP can reliably differentiate liquid drops from small ice

186. Phase 4. Perhaps this is due to aggregation reducing the concentration of ice crystals. Another possibility is that this region is subsaturated. Indeed, the RHi in transition region 4 is subsaturated.

190. This is likely due to sublimation in transition region 4.

Section 3.4. This is similar to the Yang et al. (2021) study.

260: Phase 4 has the lowest RHi and RHliq values. In fact, it is subsaturated at most temperatures (Fig. 10a). Consider moving Section 3.5 much earlier, perhaps before charactering the different transition phases. It explains a lot.

---

## Referee Comment (RC2)

**Review of "The Transition from Supercooled Liquid Water to Ice Crystals in Mixed-phase Clouds based on Airborne In-situ Observations" by F.V. Maciel and M. Diao et al. (2022)**

**Overview**

This study is focused on the microphysical characterization of mixed-phase environments and on linking it to the various stages of phase transition. The data explored here was collected in-situ during the SOCRATES field campaigns over the Southern Ocean. Identification of the phase transition stage is based on the assessment of the presence ice, liquid and mixed-phase cloud segments coexisting in the same cloud. Depending on the combination of these three thermodynamic states, the clouds were separated into four categories: (1) liquid, (2) mixed-phase ∧ liquid; (3) mixed-phase ∧ (liquid ∨ ice ∨ (liquid ∧ ice)); (4) ice. Aerosol concentration, in-cloud dynamics and atmospheric state conditions were quantified for each of these four categories. The applied method enabled conclusions regarding the effect of aerosols, atmospheric state and cloud dynamics parameters of the evolution of mixed-phase clouds. This paper is interesting and deserves attention, however, I have concerns about the general approach, data quality and clarity of presentation. I would also recommend a more thorough acknowledgement of past studies on mixed-phase clouds.

**Recommendation**: I regret to say that, in my opinion, the paper is not suitable for publication in ACP in its present form. I would recommend rewriting the manuscript addressing the comments below and resubmitting the paper.

**Major comments**

*Methodology and basic assumptions*

1.  The proposed method is based on a preconception that, during their lifetime, mixed-phase clouds pass through the stages (1)=>(2)=>(3)=>(4) as described in the paper, i.e., the cloud is initiated as liquid under supercooled conditions; then it experiences nucleation of ice and turns into mixed-phase; after that some section of the mixed-phase cloud glaciates and turns into ice, and in the final stage, the entire cloud is glaciated. The conceptual diagram of this process is shown in Fig.2, and it is used as the basis for the following interpretation of the data. This kind of "classical" evolution of mixed-phase clouds was observed and documented over 35 years ago (e.g., Hobbs and Rangno, 1985). However, besides the classical progression of mixed-phase, there are two other routes of evolution of mixed-phase clouds. The first scenario is when, after nucleation of INPs and turning liquid cloud into mixed-phase, all ice particles precipitate out of the cloud, turning the mixed-phase back into liquid. In other words, the thermodynamic phase evolution of such cloud can be described by the diagram: liquid => mixed-phase => liquid (i.e. (1)=>(2)=>(1)). The imbalance between the water vapor supply and the bulk ice mass crystal growth, required for the maintentenance of mixed-phase clouds, was discussed in Rauber and Tokay (1991), Pinto (1998). An interesting aspect of maintenance of supercooled liquid clouds was discussed by Westbrook and Illingworth (2011). There is a fair amount of modelling attempts to find an explanation of maintenance of mixed-phase clouds through the balance of INPs and dynamic forcing (e.g., Avramov, A., et al. 2011; Fan et al. 2009, 2011; Smith et al., 2009; to name a few). The second mixed-phase evolution scenario is related

to the generation of mixed-phase clouds in a pre-existing ice cloud due to dynamic forcing, which can presented by a diagram ice=>mixed-phase (i.e. (4)=>(2)). Note, that Fig.2 considers stage (4) as a final stage, whereas in the second scenario, (4) is an initial stage. The theoretical basis explaining such process was developed in Korolev and Mazin, 2003; Korolev and Field, 2008, Field et al. 2014; Hill et al, 2014). These studies were supported by earlier observations of mixed-phase clouds embedded in pre-existing, deep ice clouds (e.g., Hogan et al., 2002; Field et al. 2004). To summarize the above, the direction of the evolution of a mixed-phase environment may differ from the classical consideration (as in Fig.2), which was assumed in this work. Since the present study does not contain evidence justifying the classical evolution ( (1)=>(2)=>(3)=>(4) ) of the sampled mixed-phase clouds, it would be relevant to rewrite the sections of the paper discussing the "transition phases" and make a disclaimer of two other scenarios of the mixed-phase evolution.

2. As follows from the explanation in section 3.1, this study considers two types of mixed phase clouds as "genuine" where ice particles and liquid droplets are spatially mixed on a small scale, and "conditionally" mixed clouds, where ice and liquid are spatially separated (see Korolev et al. 2017 (Fig.5-1); Korolev and Milbrandt, 2022 (Fig.1)). In conditionally mixed-phase clouds the WBF process is disabled due to spatial separation of ice and liquid. Thus, the clouds identified in this study as (3), may form a sequence of spatially adjacent cloud segments …-ice-liquid-ice-liquid-… Such clouds are thermodynamically stable, and their lifetime will be determined by processes other than the interaction between ice and liquid (e.g., WBF, riming). Therefore, the term "transition phase" is not directly applicable to the "conditionally" mixed clouds considered in this paper, and it is relevant only to "genuinely" mixed-phase clouds. On the same note, in the frame of classical consideration of the mixed-phase evolution, the ice stage (4) is stable; (in terms mixed-phase transformation, other types of instabilities are not considered). Therefore, the term "transition phase" should not be applied to stage (4).  Having said that, the term "transition phase" should be reconsidered in this study and used cautiously and applied only to "genuinely" mixed clouds.

3. Since the direction of the evolution of mixed-phase environment may go backward, in contrast to the classical evolution, it makes sense to consider the microphysical properties of cloud thermodynamic states (1-4) without connection to the evolution of mixed-phase or to do so cautiously. This may also involve changing of the title of the paper.

*Data quality*
I have some concerns regarding the results of the particle size distributions (PSD), humidity and vertical wind measurements presented in this paper. The details are described below.
4. Measurements of DSD in liquid clouds (red lines):
   (a) The DSDs, measured by the 2DC in liquid clouds in all temperature subranges (including -30<T<-20C and -40<T<-30C), extend up to 3mm in diameter. These are exceptionally large raindrops for stratiform clouds at temperatures below -20C. To my best knowledge, I have never seen reports of observation of 2-3mm supercooled raindrops at -40<T<-20C.
   (b) As follows from Fig. 6, the concentration of supercooled drops with D>200um measured by 2DC in liquid clouds (red lines) appears to be higher than the

concentration of ice particles measured in mixed-phase clouds (light green lines) by 2DS. At temperatures $-40<T<-20C$ (Figs.6c,d) the concentration of drops $D>\sim2mm$ measured by 2DC is higher than the concentration of ice particles in ice clouds. Such behaviour appears to be anomalous.

(c) $D_{max}$ measured by 2DS and 2DC are expected to be approximately close to each other. This statement is well satisfied for PSDs in cloud types (2), (3) and (4) in Figs. 5 & 6. However, in liquid clouds (type (1)) there is a well pronounced difference between $D_{max}$ measured by the 2DS (~300um) and that measured by the 2DC (~3mm).

Items (a)-(c) are indicative that the SLD measurements by 2DC in liquid clouds are compromised.

5. The 2DS DSD in Fig.6a,c,d in the temperature subranges $-10<T<-0C$, $-30<T<-20C$ and $-40<T<-30C$ appear to be nearly the same. All three DSDs have the same $D_{max}=\sim300um$. Based on the past in-situ observations the concentration of SLD and $D_{max}$ is expected to decrease with the decrease of temperature due to an increasing of the probability of droplet freezing with the decrease of T and increase of their D. Both the absence of the temperature dependence of 2DS DSDs and observations of SLDs with $D\sim200-300um$ below $-30C$ are highly questionable.

6. The particles counted by CDP in ice cloud (type (4)) are most likely artifacts related to counting ice (e.g. Korolev et al. 2013), and therefore, their contribution to ice should be excluded.

7. The diagrams in Fig.7 show observations of LWC in liquid clouds as low as $10^{-6}$ g/m$^3$ (b), and IWC in ice clouds as low as $10^{-5.5}$ g/m$^3$ (h). Such low LWC and IWC values are below the minimum threshold, which can be measured from aircraft at 1s-averaging time by the particle probes employed in this study (e.g. Baumgardner et al. 2017).

8. Both theoretical and observational studies (Korolev and Mazin, 2003; Korolev and Isaac, 2006) showed $RH_{liq}$ in mixed phase clouds is close to 100%. Due to the short time of phase relaxation (typically 0.1-10s) in liquid and mixed-phase clouds, the evaporating droplets will rapidly bring the system of "droplets-water vapor" to quasi-equilibrium and saturate the environment. In this regard, the observations at -25C in liquid and mixed-phase clouds (with no ice) of $RH_{liq}$ ~88%, 82% and 75%, respectively (Fig.10b), is suggestive of large biases in $RH_{liq}$ measurements. The low accuracy of $RH_{liq}$ does not allow for the conclusions made in the paper about the relationships between humidity and microphysical parameters of cloud type (1)-(4).

9. Numerous in-situ observations (including those, cited in the present study e.g., Wang et al. 2020) showed that in stratiform clouds the distribution of the vertical wind is centered close to zero. A visual assessment of the diagram in Fig.10c suggests systematic biases of the vertical wind with an average speed of ~-0.2m/s or lower. For mixed-phase clouds (type 2) at -25C the biases in vertical wind reached -0.5m/s and ~-0.9m/s. Clouds subsiding at such speed are expected to evaporate within a relatively short time due to adiabatic heating. Thus, for LWC(0)=0.1g/m$^3$ at -10C, a cloud parcel descending at 0.2m/s will evaporate within 8 minutes.

*Clarity or presentation*
There are several items that require an explanation or need a more detailed description.

10. What is the definition of a cloud employed in this study? E.g., LWC>X, or N>Y or something else?

11. In section 3.1, I had a hard time understanding what the total cloud region (TCR) is. Is it a cloud separated from other clouds by a clear sky segment? Or is it an entire cloud domain sampled during a field campaign? If it is the latter, was a cloud free environment included in the statistics? If TCR refers to separate clouds, then how was the calculation of cloud statistics performed? i.e., were TCRs normalized on their spatial extension?

12. Definition of mixed-phase clouds:

    (a) The definition of mixed-phase based on LWC (or IWC) mass fraction LWC/TWC (or IWC/TWC) has been used in the cloud physics community for approximately thirty years. It is worth acknowledging this in the paper.

    (b) The second definition of mixed-phase, based on particle concentrations, is $N_{liq}/(N_{liq}+N_{ice})$, where $N_{liq}$ and and $N_{ice}$ are the concentrations of droplets and ice particles, respectively. Since, for most clouds, $N_{liq}$ is typically larger than $N_{ice}$ by 3 to 5 orders of magnitude, with a very few exceptions the ratio $N_{liq}/(N_{liq}+N_{ice}) \cong 1$. Therefore, since $N_{liq}/(N_{liq}+N_{ice}) > 0.9$, the majority of clouds should fall in the category of liquid clouds. This is clearly inconsistent with the results shown in the diagram on Fig.4b. This contradiction requires an explanation.

    (c) The spatial ratio is defined as Length(cloud type)/Length(total). In this regard, the statement on line 165: " *...for each TCR, ice spatial ratio is calculated as length of (ICR+MCR) / length of TCR*" sounds contradictory to this definition. If the definition of the spatial fraction is different from that stated above, then a more detailed explanation is required. Also, note that the spatial ratio was used for characterisation of mixed-phase clouds in Korolev et al. (2017, Fig.5-13a).

13. It would be beneficial for this work to discuss the effect of the WBF process and glaciation on the thermodynamic state of mixed-phase clouds. I found no mention of the glaciation process. The WBF was mentioned only once at the end of the paper.

14. The rapid increase of occurrence of ice clouds in the temperature range -15C to -20C was observed by other research groups (e.g., Wallace and Hobbs 1975; Moss and Johnson 1994; and others), which is worth acknowledging here.

15. It is worth indicating sampling statistics (cloud length) for each cloud type in Table 1.

*Concluding remarks*

Given the amount of work invested in this study, I would encourage the authors to rewrite the paper accounting the above comments. I did not consider any minor comments since they are eclipsed by the major issues of this work. My biggest concern is related to the data quality issues. Fixing other issues is just a matter of time.

Alexei Korolev

Avramov, A., et al. (2011), Toward ice formation closure in Arctic mixed-phase boundary layer clouds during ISDAC, J. Geophys. Res., 116, D00T08, doi:10.1029/2011JD015910.

Baumgardner, D., and Coauthors, 2017: Cloud ice properties: In situ measurement challenges. Ice Formation and Evolution in Clouds and Precipitation: Measurement and Modeling Challenges, Meteor. Monogr., No. 58, Amer. Meteor. Soc., doi:10.1175/AMSMONOGRAPHS-D-16-0011.1.

Fan, J., M. Ovtchinnikov, J. M. Comstock, S. A. McFarlane, and A. Khain, 2009: Ice formation in Arctic mixed-phase clouds: Insights from a 3-D cloud-resolving model with size-resolved aerosol and cloud microphysics. J. Geophys. Res., 114, D04205, doi:10.1029/2008JD010782.

Fan, J., S. Ghan, M. Ovchinnikov, X. Liu, P. J. Rasch, and A. Korolev, 2011: Representation of Arctic mixed-phase clouds and the Wegener-Bergeron-Findeisen process in climate models: Perspectives from a cloud-resolving study, J. Geophys. Res., 116, D00T07, doi:10.1029/2010JD015375

Field, P. R., R. J. Hogan, P. R. A. Brown, A. J. Illingworth, T. W. Choularton, P. H. Kaye, E. Hirst, and R. Greenaway, 2004: Simultaneous radar and aircraft observations of mixed-phase cloud at the 100 m-scale. Quart. J. Roy. Meteor. Soc., 130, 1877–1904, doi:10.1256/qj.03.102

Field, P. R., A. A. Hill, K. Furtado, and A. Korolev, 2014: Mixed-phase clouds in a turbulent environment. Part 2: Analytic treatment. Quart. J. Roy. Meteor. Soc., 140, 870–880. doi:10.1002/qj.2175.

Hobbs, P.V., and A. L. Rangno, 1985: Ice Particle Concentrations in Clouds. *J. Atmos. Sci.*, 42, 2523-2549, DOI: https://doi.org/10.1175/1520-0469(1985)042<2523:IPCIC>2.0.CO;2

Hill, A. A., P. R. Field, K. Furtado, A. Korolev, and B. J. Shipway, 2014: Mixed-phase clouds in a turbulent environment. Part 1: Large-eddy simulation experiments. Quart. J. Roy. Meteor. Soc., 140, 855–869, doi:10.1002/qj.2177.

Hogan, R. J., P. R. Field, A. J. Illingworth, R. J. Cotton, and T. W. Choularton, 2002: Properties of embedded convection in warm-frontal mixed-phase cloud from aircraft and polarimetric radar. Quart. J. Roy. Meteor. Soc., 128, 451–476, doi:10.1256/003590002321042054.

Korolev, A. V., and I. P. Mazin, 2003: Supersaturation of water vapor in clouds. J. Atmos. Sci., 60, 2957–2974, doi:10.1175/1520-0469(2003)060,2957:SOWVIC.2.0.CO;2.

Korolev, A. V., and G. A. Isaac, 2006: Relative humidity in liquid, mixed phase and ice clouds. J. Atmos. Sci., 63, 2865–2880, doi:10.1175/JAS3784.1.

Korolev, A. V., and P. R. Field, 2008: The effect of dynamics on mixed-phase clouds: Theoretical considerations. J. Atmos. Sci., 65, 66–86, doi:10.1175/2007JAS2355.1.

Korolev, A.V., E. Emery, J. W. Strapp, S. G. Cober, and G. A. Isaac, 2013b: Quantification of the effects of shattering on airborne ice particle measurements. J. Atmos. Oceanic Technol., 30, 2527–2553, doi:10.1175/JTECH-D-13-00115.1.

Korolev, A., McFarquhar, G., Field, P. R., Franklin, C., Lawson, P., Wang, Z., et al. 2017:. Mixed-phase clouds: Progress and challenges. *Meteorological Monographs*, *58*, 5.1–5.50. https://doi.org/10.1175/AMSMONOGRAPHS-D-17-0001.1

Korolev, A., & Milbrandt, J., 2022: How are mixed-phase clouds mixed? *Geophysical Research Letters,49*, e2022GL099578. https://doi.org/10.1029/2022GL099578

Morrison H, de Boer G, Feingold G, Harrington J, Shupe MD, Sulia K. 2011: Resilience of persistent Arctic mixed-phase clouds. *Nature Geosci.* DOI:10.1038/ngeo1332.

Moss, S. J. and Johnson, D.W. 1994 Aircraft measurements to validate and improve numerical model parametrization of ice to water ratios in clouds. *Atmos. Res.*, **34,** 1–25

Pinto, J. O., 1998: Autumnal mixed-phase cloudy boundary layers in the Arctic. *J. Atmos. Sci.*, 55, 2016–2037, doi:10.1175/1520-0469(1998)055,2016:AMPCBL.2.0.CO;2.

Rauber, R.M, Tokay A. 1991: An explanation for the existence of supercooled liquid water at the top of cold clouds. *J. Atmos. Sci.* **48**: 1005–1023.

Smith, A.J, Larson V.E, Niu J, Kankiewicz J.A, Carey L.D. 2009: Processes that generate and deplete liquid water and snow in thin midlevel mixed-phase clouds. *J. Geophys. Res.* **114**: D12203, DOI: 10.1029/2008JD011531

Wallace, J. M. and Hobbs, P. V. 1975 *Atmospheric Science: An introductory survey*. Academic Press, New York, USA

Westbrook, C. D., and A. J. Illingworth (2011), Evidence that ice forms primarily in supercooled liquid clouds at temperatures > −27°C, *Geophys. Res*. Lett., 38, L14808, doi:10.1029/2011GL048021.

---

## Author Response (AR1)

**Overarching Responses to the Reviewers**

Format: The reviewer's comments are quoted in italic

Line number in the response refers to the revised manuscript with tracked changes

Quotation in red color stands for revised/added text in the revised manuscript

We thank our reviewers for the helpful comments they took the time to provide us with. We would also like to thank the three reviewers and the leading editor for their patience on our responses. Earlier this year, my former student (Flor, who is now a PhD candidate at UCLA) and I myself each experienced unexpected difficulties and stress with our close family member's health. We each had to take the main responsibility for our family care for a while, which led to an unprecedented disruption to both of our regular work. I also invited my current graduate student, Ching An Yang (our new third coauthor), for her help with our preparation of the new figures since Ching had previous experience with the NSF SOCRATES data in her paper Yang et al. (2021) published in JGR-Atmosphere.

Below is a summary of the main revisions that we conducted, followed by our individual response to each reviewer. Overall, we revised almost all original figures and table 1, added new Figure 4, new supplemental Figures S1 – S10 and Table S1.

1. We have done a thorough data quality control on multiple instruments, including measurements of relative humidity, vertical velocity, and several cloud probes, 2DS, CDP and 2DC. We elaborate our quality control results in our response below to reviewer 2.

2. We conducted analysis of two new remote sensing instruments (i.e., HIAPER Cloud Radar and High Spectral Resolution Lidar) onboard the Gulfstream-V aircraft during the SOCRATES campaign, in order to address the concerns of precipitation by reviewer 3 and as a possible approach to examine cloud vertical layers in a comment raised by reviewer 1.

3. We expanded the possible pathways for cloud evolution as recommended by reviewer 2, and added more context on how the transition phases in this work are related to the previous studies.

**Response to comments from Reviewer 1**

*Review of "The Transition from Supercooled Liquid Water to Ice Crystals in Mixed-phase Clouds based on Airborne In-situ Observations", by Maciel and Diao.*

*This is an interesting and I'd say novel study that characterizes the degree of glaciation of supercooled clouds in the Southern Ocean, using data from the SOCRATES field program. They use a combination of "bulk" data from the 2DS and CDP probes, augmented with the King liquid water content and Rosemont Icing Probes for phase discrimination. The methodology is based on Yang et al. (2021), which is based on D'Alessandro et al. (2019). A second method uses the 2DS images and a machine learning tool to identify individual images of liquid and ice particles. The methodology and some of the results bear a striking similarity to the Yang et al. (2021) article. This includes the relationship of the phase partitioning to the aerosol concentration. Some mention of the results of that study should be given and how the techniques and results differ.*

We thank the reviewer for the helpful comments and below is our response to each of them.

Regarding the differences with Yang et al. (2021) paper, the main differences are that this work separates the transition phases of clouds instead of analyzing all clouds together, and this work compares different methods to define liquid, mixed and ice phase, including using the microphysical properties of hydrometeor mass concentrations or number concentrations, and the macrophysical properties of the

spatial fraction of ice-containing segments. Below are two examples where we distinguish this current work from that of Yang et al. (2021).

One example is that we modified the cloud phase identification for the 2DS probe compared with Yang et al. (2021): "… The second modification is about the treatment of large particles identified as liquid droplets. The previous method restricts particles with maximum dimensions ($D_{max}$) > 312.5 µm as ice particles, while those with $D_{max}$ between 112.5 and 312.5 µm can be either liquid or ice depending on the standard deviation of particle sizes measured by 2DS in that second. In this work, we restrict particles with $D_{max}$ > 212.5 µm to be ice particles, reducing the number of large particles being categorized as liquid droplets."

Another example is about the aerosol indirect effect analysis: "Stronger positive correlations between IWC and $N_{>500}$ compared with $N_{>100}$ are also shown in the previous work by Yang et al. (2021), although that study did not differentiate the transition phase of clouds nor examine aerosol indirect effects in relation to cloud macrophysical properties, i.e., the spatial expansion of ice-containing cloud segments."

*The partitioning of the data into liquid only, mostly liquid, mostly ice and all ice phases is interesting, and relating it to the macrophysical properties of the cloud layer (ice content, relative humidity with respect to water and ice, and vertical velocity is interesting. Likewise, the ice fraction partitioning is rather interesting.*

*I tried to think of the factors that determine the glaciation of a stratiform cloud layer. First, in stratiform ice cloud layers, typically liquid water regions, the stronger vertical motions, and ice nucleation mostly occur at cloud top. Below cloud top and depending on the vertical velocity which is responsible for the degree of ice supersaturation or subsaturation, can contain a growth or sublimation layer. The measurements you report on do not account for where vertically within the cloud layer relative to cloud top the aircraft penetrations were made. Thus, you may be sampling in a subsident or upward moving parcel of air. Perhaps your measurements at temperatures below -15⁰C are near cloud top, and those at the warmer temperatures in the middle or lower parts of the cloud layers. The relationship of the relative humidity to turbulence might be a manifestation of where in the cloud layer the sampling is done, and whether generating cells were penetrated. You do mention that the aircraft observations only captures the 1-D structure of a cloud segment, while cloud layers above and below the aircraft flight track may show a different ice spatial ratio on a 2-D or 3-D view. Nonetheless, I think a weakness of the study is that there is no partitioning of where in the cloud layer the measurements are made. (Note that it's unlikely that the vertical motions measured by the aircraft system are sufficiently accurate to determine what zone the measurements are made in, unless generating cells are penetrated).*

We thank the reviewer for this helpful comment. We agree that identifying the location of the aircraft in-situ sampling relative to the cloud vertical layer would be helpful. We tried the approach of analyzing the two airborne remote sensing instruments onboard – HIAPER Cloud Radar (HCR) and High Spectral Resolution Lidar (HSRL), which provide some profiling either above or below the aircraft flight track. But we ran into issues that the remote sensing data are only pointed toward one direction at a given time, so we can only inspect either above or below the aircraft for what the cloud layers were like. Also, the remote sensing instruments cannot measure in close proximity to the aircraft, which leave an area without remote sensing data surrounding the flight track. Since trying this approach does not provide a straightforward answer, we also tried looking into the field catalogue for possible identification of cloud top, middle, or base sampling. We found that only a few cloud segments have been identified with clear sampling strategies, which would severely affect the sample size that we can utilize. Thus, we decided to address this comment by conducting analysis on regions with different magnitudes of liquid and ice water content (LWC and IWC), and relative humidity with respect to ice ($RH_i$), since these values typically vary with vertical levels within a cloud as the reviewer mentioned.

Different ranges of LWC, IWC and $RH_i$ are used in the identifications of transition phases in **supplemental Figure S2 - S4**, respectively (shown below), which show small variations in the number of samples of four transition phases. In addition, we contrasted the differences between using all RHi values from the entire in-cloud layer and using only the higher RHi values ($RH_i > 80\%$) that potentially represent the higher levels within a cloud. We found that the differences between them are very small for several key analyses in this work, shown in **supplemental material Figures S7 – S9**, which are comparable with Figures 7, 8 and 9, respectively. Below we put these analyses side-by-side to illustrate the small differences between them. We discussed these additional tests and added suggestions for future field campaigns.

We added discussion in section 3.1: "Since the 1-D aircraft sampling can be at any vertical level relative to a cloud layer, we further examine the impacts of restricting the analysis to different ranges of LWC, IWC, and RHi values (supplementary Figures S2 – S4). Previous studies such as Wang et al. (2012) and D'Alessandro et al. (2023) have shown that cloud top usually contains higher LWC than cloud base, while IWC increases from the cloud top to cloud base. D'Alessandro et al. (2019) also showed that in-cloud samples with higher liquid mass fraction have higher RH values closer to liquid saturation. By using different ranges of LWC, IWC and RHi as proxies for vertical levels within cloud layers, we found that the number of samples of the four transition phases are relatively similar unless very high LWC or IWC are used ($> 0.1$ g m$^{-3}$)."

We also added comments in section 4: "Future investigation that compares 1-D aircraft sampling with 2-D remote sensing observations and 3-D model simulations is recommended to further examine the quasi-Lagrangian evolution of mixed-phase clouds."

[Figure]

Figure S2. Similar to Figure 3 a, number of samples for four transition phases but using different liquid water content (LWC) values (unit: g m$^{-3}$) as the threshold for defining in-cloud conditions.

[Figure]

Figure S3. Similar to Figure 3 a, number of samples for four transition phases but using different ice water content (IWC) values (unit: g m$^{-3}$) as the threshold for defining in-cloud conditions.

[Figure]

Figure S4. Similar to Figure 3 a, number of samples for four transition phases but using different ranges of relative humidity with respect to ice (RH$_i$) for analysis of in-cloud conditions.

[Figure]

Figure R1-1. Side-by-side comparisons of using all RH$_i$ data (left) versus using only high RH$_i$ (>80%) (right). The figure on the left is Figure 7, on the right is Figure S7.

[Figure]

Figure R1-2. Side-by-side comparisons of using all RH$_i$ data (left) versus using only high RH$_i$ (>80%) (right). The figure on the left is Figure 8, on the right is Figure S8.

[Figure]

Figure R1-3. Side-by-side comparisons of using all RH$_i$ data (top) versus using only high RH$_f$ (>80%) (bottom). The figure on the top is Figure 9, on the bottom is Figure S9.

*Figure 10, which shows the distributions of RHi, RHliq, vertical velocity and standard deviation of vertical velocity which be presented before Figure 3 as it provides a context for how the various transition regions relate to the large-scale properties of the cloud layer. What is shows is that transition region 4 is subsaturated. This why there is no liquid water in that region, not a stage of development of the ice cloud. It also shows up in the PSDs-transition region 4 has fewer ice particles and smaller maximum ice particle sizes than transition region 3. These might be the trails of generating cells, where the growth is aloft and sublimation lower down in the cloud layer.*

We moved the original Figures 10 and 11 forward and combined them into the **new Figure 5**. We decided to show Figure 3 first since it provides the number of samples for each phase. We also added new **Figure 4** (shown below) to illustrate the full distribution of RH$_i$ and standard deviation of vertical velocity ($\sigma_w$) for each phase. Please note that the RH$_i$ shown in the original Figures 10 and 11 only represents the mean value in each bin. When examining the entire distribution of RH$_i$, all phases have RH$_i$ being ice supersaturated, saturated and subsaturated. For example, phase 4 has ice supersaturation as well. This is consistent with what reviewer 2 pointed out that another possible pathway of the cloud evolution is that phase 4 pure ice segments can turn into phase 2 with coexisting liquid and ice. We added discussion on this part in section 2.2: "The third pathway of mixed-phase evolution is related to the generation of mixed-phase clouds in a pre-existing ice cloud due to dynamic forcing, which can be presented as ice=>mixed-phase, i.e., phases (4)=>(2), or (4)=>(3)=>(2). Note that the numerical order of phases 1 – 4 does not necessarily represent the evolution direction. For example, phase 4 may either be the final stage in the first classical pathway, whereas in the third pathway, phase 4 is an initial stage."

We agree with reviewer's comments that some of the phase 4 segments could be trails of generating cells, and we added this comment as a new discussion in section 3.3 when describing the particle size distributions (PSDs): "It is possible that some of the phase 4 samples may represent the trails of generating cells, where the growth is aloft, and sublimation is at the lower part of the cloud layer.".

[Figure]

Figure 4. Distributions of (a-h) $RH_i$ and (i-p) $\sigma_w$ in various transition phases as a function of temperature. Dashed lines in (a) – (h) indicate liquid saturation.

[Figure]

Figure 5. Distributions of (a) $RH_i$, (b) $RH_{liq}$, (c) vertical velocity (w) and (d) standard deviation of vertical velocity ($\sigma_w$) for various transition phases at different temperatures. (e-h) Similar to (a-d), but in relation to various mixed spatial ratios or ice spatial ratios. Phases 1 and 4 show ice spatial ratio at 0 and 1, respectively, and therefore only a single dot is shown for phases 1 and 4 in (e-h).

*Minor comments*

*Line 6: determines the "ice cloud lifetime"*

Revised.

*45: resilience to what?*

Revised to "The persistent existence of mixed-phase cloud systems".

*62: "growing"?*

Removed "growing".

*66: aerosol*

Revised.

*97: remove "various"*

Removed.

*183: I don't think the CDP can reliably differentiate liquid drops from small ice*

We thank the reviewer for pointing out this point. We removed the identification of ice by CDP, as also recommended by reviewer 2. In the revised manuscript, only when CDP identifies particles as liquid droplets, these measurements are being used.

*186. Phase 4. Perhaps this is due to aggregation reducing the concentration of ice crystals.*

*Another possibility is that this region is subsaturated. Indeed, the RHi in transition region 4 is subsaturated.*

We revised the text to add these new possible explanations: "The decreasing ice crystal concentrations per size bin from phase 3 to phase 4 may be caused by stronger aggregation, sublimation, and/or sedimentation of ice crystals in phase 4, as well as by stronger secondary ice production in phase 3."

*190. This is likely due to sublimation in transition region 4.*

We added these new possible explanations: "The significant decrease (1 to 4 orders of magnitude) of hydrometeor concentrations per size bin at $20 - 100$ µm in phase 4 compared with the other three phases suggests that most supercooled liquid water may have evaporated and transitioned into ice phase through WBF process or riming, instead of the freezing of individual droplets, while the small ice crystals may have sublimated."

*Section 3.4. This is similar to the Yang et al. (2021) study.*

We added this citation and explained the difference between this study and that previous study: "Stronger positive correlations between IWC and $N_{>500}$ compared with $N_{>100}$ are also shown in the previous work by Yang et al. (2021), although that study did not differentiate the transition phase of clouds nor examine aerosol indirect effects in relation to cloud macrophysical properties, i.e., the spatial expansion of ice-containing cloud segments."

*260: Phase 4 has the lowest RHi and RHliq values. In fact, it is subsaturated at most temperatures (Fig. 10a). Consider moving Section 3.5 much earlier, perhaps before charactering the different transition phases. It explains a lot.*

We moved the original Figures 10 and 11 forward and they now are combined into one new **Figure 5**. We also added a new **Figure 4** as mentioned above. The section has also been moved forward and it is the new section 3.2.

**Response to comments from Reviewer 2**

*Review of "The Transition from Supercooled Liquid Water to Ice Crystals in Mixed-phase Clouds based on Airborne In-situ Observations" by F.V. Maciel and M. Diao et al. (2022)*

*Overview*

*This study is focused on the microphysical characterization of mixed-phase environments and on linking it to the various stages of phase transition. The data explored here was collected in- situ during the SOCRATES field campaigns over the Southern Ocean. Identification of the phase transition stage is based on the assessment of the presence ice, liquid and mixed-phase cloud segments coexisting in the same cloud. Depending on the combination of these three thermodynamic states, the clouds were separated into four categories: (1) liquid, (2) mixed-phase ⋀ liquid; (3) mixed-phase ⋀ (liquid ⋁ ice ⋁ (liquid ⋀ ice)); (4) ice. Aerosol concentration, in-cloud dynamics and atmospheric state conditions were quantified for each of these four categories. The applied method enabled conclusions regarding the effect of aerosols, atmospheric state and cloud dynamics parameters of the evolution of mixed-phase clouds. This paper is interesting and deserves attention, however, I have concerns about the general approach, data quality and clarity of presentation. I would also recommend a more thorough acknowledgement of past studies on mixed-phase clouds.*

*Recommendation: I regret to say that, in my opinion, the paper is not suitable for publication in ACP in its present form. I would recommend rewriting the manuscript addressing the comments below and resubmitting the paper.*

We thank the reviewer for the helpful comments and below is our response to each of them.

*Major comments*

*Methodology and basic assumptions*

*1. The proposed method is based on a preconception that, during their lifetime, mixed-phase clouds pass through the stages (1)=>(2)=>(3)=>(4) as described in the paper, i.e., the cloud is initiated as liquid under supercooled conditions; then it experiences nucleation of ice and turns into mixed-phase; after that some section of the mixed-phase cloud glaciates and turns into ice, and in the final stage, the entire cloud is glaciated. The conceptual diagram of this process is shown in Fig.2, and it is used as the basis for the following interpretation of the data. This kind of "classical" evolution of mixed-phase clouds was observed and documented over 35 years ago (e.g., Hobbs and Rangno, 1985). However, besides the classical progression of mixed-phase, there are two other routes of evolution of mixed-phase clouds. The first scenario is when, after nucleation of INPs and turning liquid cloud into mixed-phase, all ice particles precipitate out of the cloud, turning the mixed-phase back into liquid. In other words, the thermodynamic phase evolution of such cloud can be described by the diagram: liquid => mixed-phase => liquid (i.e. (1)=>(2)=>(1)). The imbalance between the water vapor supply and the bulk ice mass crystal growth, required for the maintentenance of mixed-phase clouds, was discussed in Rauber and Tokay (1991), Pinto (1998). An interesting aspect of maintenance of supercooled liquid clouds was discussed by Westbrook and Illingworth (2011). There is a fair amount of modelling attempts to find an explanation of maintenance of mixed-phase clouds through the balance of INPs and dynamic forcing (e.g., Avramov, A., et al. 2011; Fan et al. 2009, 2011; Smith et al., 2009; to name a few). The second mixed-phase evolution scenario is related to the generation of mixed-phase clouds in a pre-existing ice cloud due to dynamic forcing, which can presented by a diagram ice=>mixed-phase (i.e. (4)=>(2)). Note, that Fig.2 considers stage (4) as a final stage, whereas in the second scenario, (4) is an initial stage. The theoretical basis explaining such process was developed in Korolev and Mazin, 2003; Korolev*

*and Field, 2008, Field et al. 2014; Hill et al, 2014). These studies were supported by earlier observations of mixed-phase clouds embedded in pre-existing, deep ice clouds (e.g., Hogan et al., 2002; Field et al. 2004). To summarize the above, the direction of the evolution of a mixed-phase environment may differ from the classical consideration (as in Fig.2), which was assumed in this work. Since the present study does not contain evidence justifying the classical evolution ( (1)=>(2)=>(3)=>(4) ) of the sampled mixed-phase clouds, it would be relevant to rewrite the sections of the paper discussing the "transition phases" and make a disclaimer of two other scenarios of the mixed-phase evolution.*

We thank the reviewer for these very helpful and detailed comments on the possible transition/evolution pathways among liquid, mixed and ice phase clouds. We revised the conceptual diagram in **Figure 2** to illustrate these additional possible pathways, allowing the two-way transitions between each transition phase to occur.

[Figure]

Figure 2. A conceptual diagram of the four transition phases for mixed-phase cloud evolution. Red, blue, and purple shading indicates liquid cloud region (LCR), ice cloud region (ICR) and mixed cloud region (MCR), respectively.

In addition, we added a paragraph in section 3.1 to discuss these pathways and provided more context of how the transition phases are related to the previous studies: "Several potential evolution pathways have been documented and discussed in previous literature, which can be linked with the separation of the four transition phases described above. A "classical" type of evolution pathway follows phases (1)=>(2)=>(3)=>(4), which was observed and documented over 35 years ago (e.g., Hobbs and Rangno, 1985). This type of evolution describes the situation that a cloud is initiated as liquid phase under supercooled conditions; then it experiences ice nucleation and turns into mixed-phase; after that some section of the mixed-phase cloud glaciates and turns into ice; and in the final stage, the entire cloud is glaciated. Besides the classical progression of mixed-phase, there are two other routes of evolution of mixed-phase clouds. The second pathway is when, after nucleation of INPs and turning liquid clouds into mixed-phase, all ice particles precipitate out of the clouds, turning the mixed-phase back into liquid. In other words, the thermodynamic phase evolution of such clouds can be described as liquid => mixed-phase => liquid, i.e., phases (1)=>(2)=>(1). The imbalance between the water vapor supply and the bulk ice mass crystal growth, required for the maintenance of mixed-phase clouds, was discussed in Rauber and Tokay (1991), Pinto (1998), and Westbrook and Illingworth (2011). There is a fair amount of modelling attempts to find an explanation of maintenance of mixed-phase clouds through the balance of

INPs and dynamic forcing (e.g., Avramov et al., 2011; Fan et al., 2009, 2011; Smith et al., 2009). The third pathway of mixed-phase evolution is related to the generation of mixed-phase clouds in a pre-existing ice cloud due to dynamic forcing, which can be presented as ice=>mixed-phase, i.e., phases (4)=>(2), or (4)=>(3)=>(2). Note that the numerical order of phases 1 – 4 does not necessarily represent the evolution direction. For example, phase 4 may either be the final stage in the first classical pathway, whereas in the third pathway, phase 4 is an initial stage. The theoretical basis explaining such process was developed in several previous studies (e.g., Korolev and Mazin, 2003; Korolev and Field, 2008, Field et al., 2014; Hill et al., 2014). These studies were supported by earlier observations of mixed-phase clouds embedded in pre-existing, deep ice clouds (e.g., Hogan et al., 2002; Field et al., 2004). We caution that a mixed-phase cloud may or may not follow these exact pathways in the real atmosphere, as certain transition phases may be skipped, the evolution direction could be reversed, and multiple phases can appear in the same cloud in a 3-D view. Nevertheless, this method provides a statistical separation of the cloud transition phases and allows a more focused analysis of the coexistence of supercooled liquid water and ice crystals that cannot be achieved solely based on second-by-second measurements (i.e., if one only analyzes seconds with coexisting ice and liquid)."

*2. As follows from the explanation in section 3.1, this study considers two types of mixed phase clouds as "genuine" where ice particles and liquid droplets are spatially mixed on a small scale, and "conditionally" mixed clouds, where ice and liquid are spatially separated (see Korolev et al. 2017 (Fig.5-1); Korolev and Milbrandt, 2022 (Fig.1)). In conditionally mixed-phase clouds the WBF process is disabled due to spatial separation of ice and liquid. Thus, the clouds identified in this study as (3), may form a sequence of spatially adjacent cloud segments ...-ice-liquid-ice-liquid-... Such clouds are thermodynamically stable, and their lifetime will be determined by processes other than the interaction between ice and liquid (e.g., WBF, riming). Therefore, the term "transition phase" is not directly applicable to the "conditionally" mixed clouds considered in this paper, and it is relevant only to "genuinely" mixed-phase clouds. On the same note, in the frame of classical consideration of the mixed-phase evolution, the ice stage (4) is stable; (in terms mixed-phase transformation, other types of instabilities are not considered). Therefore, the term "transition phase" should not be applied to stage (4). Having said that, the term "transition phase" should be reconsidered in this study and used cautiously and applied only to "genuinely" mixed clouds.*

We thank the reviewer for pointing out the stability of some conditions compared with others. We would like to point out that the scenario of "LCR+ICR" is one of the three scenarios for phase 3, since phase 3 can also have "LCR+MCR+ICR" and "MCR+ICR". The conditionally mixed phase clouds (i.e., represented by LCR+ICR) only contribute to 840 seconds out of the total 11988 seconds of phase 3.

We added more discussion about this in section 4: "The definition of LCR, MCR and ICR is also related to the two types of mixed-phase clouds – genuinely versus conditionally mixed, separated by the level of mixing between supercooled liquid water and ice crystals (e.g., Korolev et al., 2017, their Fig. 5-1; Korolev and Milbrandt, 2022, their Fig. 1). The scenario of "LCR+ICR" identified as one sub-category in phase 3 would be considered a conditionally mixed-phase cloud, which may form a sequence of spatially adjacent cloud segments ...-ice-liquid-ice-liquid-.... Such clouds may be thermodynamically stable, and their lifetime would be determined by processes other than the interaction between ice and liquid (e.g., WBF and riming). This special scenario when only "LCR+ICR" exist in the TCR without the existence of MCR has 840 seconds of samples, which is a small fraction of the total 11988 seconds of transition phase 3 samples. This result suggests that most of the clouds with coexisting supercooled liquid water and ice particles at least contain some partial segments as genuinely mixed phase, i.e., MCR."

*3. Since the direction of the evolution of mixed-phase environment may go backward, in contrast to the classical evolution, it makes sense to consider the microphysical properties of cloud thermodynamic*

*states (1-4) without connection to the evolution of mixed-phase or to do so cautiously. This may also involve changing of the title of the paper.*

We have rewritten the manuscript carefully, to tune down the description of the cloud evolution direction, but rather focus on these transition phases as various stages of clouds. Since we have revised the cartoon diagram of Figure 2 as illustrated above, the transition phases are not limited to a single direction of evolution but allow two-way transitions and a cycle between these phases. In addition, we changed the title of the manuscript to: "Transition between  Supercooled Liquid Water and  Ice Crystals in Mixed-phase Clouds based on Airborne In-situ Observations".

Because of this, we feel that the transition phase term does not indicate a specific connection with the evolution direction of the clouds anymore, and therefore we keep using the term.

*Data quality*

*I have some concerns regarding the results of the particle size distributions (PSD), humidity and vertical wind measurements presented in this paper. The details are described below.*

*4. Measurements of DSD in liquid clouds (red lines):*

*(a) The DSDs, measured by the 2DC in liquid clouds in all temperature subranges (including -30<T<-20C and -40<T<-30C), extend up to 3mm in diameter. These are exceptionally large raindrops for stratiform clouds at temperatures below -20C. To my best knowledge, I have never seen reports of observation of 2-3mm supercooled raindrops at -40<T<-20C.*

*(b) As follows from Fig. 6, the concentration of supercooled drops with D>200um measured by 2DC in liquid clouds (red lines) appears to be higher than the concentration of ice particles measured in mixed-phase clouds (light green lines) by 2DS. At temperatures -40<T<-20C (Figs.6c,d) the concentration of drops D>~2mm measured by 2DC is higher than the concentration of ice particles in ice clouds. Such behaviour appears to be anomalous.*

*(c) Dmax measured by 2DS and 2DC are expected to be approximately close to each other. This statement is well satisfied for PSDs in cloud types (2), (3) and (4) in Figs. 5 & 6. However, in liquid clouds (type (1)) there is a well pronounced difference between Dmax measured by the 2DS (~300um) and that measured by the 2DC (~3mm).*

*Items (a)-(c) are indicative that the SLD measurements by 2DC in liquid clouds are compromised.*

We thank the reviewer for giving us detailed comments on the data quality of various measurements. Regarding the large particles reported by 2DC in the original PSD figures, we conducted a comparison between Fast-2DC and 2DS probes for each research flight using time series and statistical comparison. We concluded that the Fast-2DC measurements had issues with particle imaging during various flight segments in SOCRATES and decided to exclude Fast-2DC probe from this analysis. Nevertheless, our main analysis relies on the other two probes (i.e., 2DS and CDP) and the main conclusion is not affected by the exclusion of Fast-2DC.

*5. The 2DS DSD in Fig.6a,c,d in the temperature subranges -10<T<-0C, -30<T<-20C and -40<T<-30C appear to be nearly the same. All three DSDs have the same Dmax=~300um. Based on the past in-situ observations the concentration of SLD and Dmax is expected to decrease with the decrease of temperature due to an increasing of the probability of droplet freezing with the decrease of T and*

*increase of their D. Both the absence of the temperature dependence of 2DS DSDs and observations of SLDs with D~200-300um below -30C are highly questionable.*

We thank the reviewer for pointing out these detailed features. We applied the following revisions to address the reviewer's question of erroneous, large liquid particles measured by 2DS with diameter > 200 micron in phase 1. Previously, we defined 2DS detected particles with $D_{max}$ (maximum dimension in a second) greater than 312.5 micron as ice particles, following the same definition described in D'Alessandro et al. (2019) in their Figure 1. For particles with $D_{max}$ between 112.5 and 312.5 micron, they are defined as liquid if the standard deviation of particles within that second ($\sigma_D$) ≤50 micron, and as ice if $\sigma_D$ > 50 micron. That is, if a particle has diameter > 200 micron, it can be either ice or liquid as the reviewer pointed out. In the revised method, we restrict particles with $D_{max}$ > 212.5 micron to be ice particles, reducing the large particles being categorized as liquid droplets. The revised PSD in the new **Figure 6** is copied below. The 2DS measurements in phase 1 show maximum dimensions up to 212.5 micron. Phase 4 does not show CDP measurements since ice particles detected by CDP are excluded from this analysis.

We added the explanations to section 2.2: "Two modifications are applied to the previous cloud phase identification method of D'Alessandro et al. (2019) and Yang et al. (2021). The first modification is that only when CDP measurements are categorized as liquid droplets, these samples are used in the analysis. Measurements categorized by CDP as ice particles are excluded since previous work has shown that these measurements related to counting ice are most likely artifacts (e.g., Korolev et al., 2013). The second modification is about the treatment of large particles identified as liquid droplets. The previous method restricts particles with maximum dimensions ($D_{max}$) > 312.5 µm as ice particles, while those with $D_{max}$ between 112.5 and 312.5 µm can be either liquid or ice depending on the standard deviation of particle sizes measured by 2DS in that second. In this work, we further restrict particles with $D_{max}$ > 212.5 µm to be ice particles, reducing the number of large particles being categorized as liquid droplets."

Regarding the comment about temperature dependence of PSD, upon a closer examination, we found that phase 4 (blue line in Figure 6) shows decreasing frequencies of large ice particles with decreasing temperatures. For example, particles with 3000 µm dimension shows that dN/dlogDp value in the y axis decreases from $10^{-4}$ to $10^{-6}$ when temperature decreases (new Figure 6 a – d). But for phase 3, where ice particles still coexist with liquid droplets, it shows almost no trend of decreasing $D_{max}$ with decreasing temperature. Our speculation is that since phase 3 still has supercooled liquid water coexisting with ice, these ice particles may still be subject to growth via WBF process, vapor depositional growth if there is ice supersaturation, and/or riming. Thus, the ice particle growth process is less limited in phase 3 compared with phase 4. We added some discussions on this in section 3.3: "Phase 4 shows a trend of decreasing frequency of large ice particles (e.g., $D_{max}$ > 2000 µm) with decreasing temperature, which could be due to an increasing probability of droplet freezing with decreasing temperature given the same dimension that reduces the available amount of large supercooled liquid droplets for glaciation or riming at lower temperatures. On the other hand, phase 3, which still has supercooled liquid water coexisting with ice particles, does not show such trend, probably because ice crystal growth may occur via various processes in phase 3, such as WBF process, glaciation, vapor depositional growth under ice supersaturation, and/or riming."

[Figure]

Figure 6. Particle size distribution of the four transition phases for mixed-phase clouds separated by probe types and temperature ranges. Phase 4 only shows 2DS measurements because ice particles measured by CDP are excluded from the analysis.

*6. The particles counted by CDP in ice cloud (type (4)) are most likely artifacts related to counting ice (e.g. Korolev et al. 2013), and therefore, their contribution to ice should be excluded.*

We agree with the reviewer and in the revised manuscript, all CDP measurements initially categorized as ice particles are excluded from the analysis. In other words, only CDP measurements categorized as liquid droplets are used. Section 2.2 is revised: "Two modifications are applied to the previous cloud phase identification method of D'Alessandro et al. (2019) and Yang et al. (2021). The first modification is

that only when CDP measurements are categorized as liquid droplets, these samples are used in the analysis. Measurements categorized by CDP as ice particles are excluded since previous work has shown that these measurements related to counting ice are most likely artifacts (e.g., Korolev et al., 2013)."

*7. The diagrams in Fig.7 show observations of LWC in liquid clouds as low as 10-6 g/m3 (b), and IWC in ice clouds as low as 10-5.5 g/m3 (h). Such low LWC and IWC values are below the minimum threshold, which can be measured from aircraft at 1s-averaging time by the particle probes employed in this study (e.g. Baumgardner et al. 2017).*

Following the suggestion of the reviewer, we revised the thresholds of in-cloud conditions to a higher threshold of total water content (TWC), described in section 2.1: "In-cloud conditions are defined as the 1-Hz measurements with total water content (TWC = IWC+LWC) greater than $0.001 \ \mathrm{g \ m^{-3}}$. Lower IWC and LWC values have also been reported by the two probes, but the threshold of $0.001 \ \mathrm{g \ m^{-3}}$ is chosen here due to the larger uncertainties of these cloud probes reporting lower mass concentrations of hydrometeors (e.g., Baumgardner et al., 2017)."

In addition, we compared the impacts of using different IWC and LWC values as in-cloud thresholds in **supplemental Figures S2 and S3** (copied below). The analysis shows that using IWC or LWC thresholds of $10^{-6}$, $10^{-5}$, $10^{-4}$, and $10^{-3} \ \mathrm{g \ m^{-3}}$ do not show significant differences in the number of samples of four phases.

[Figure]

Figure S2. Similar to Figure 3 a, number of samples for four transition phases but using different liquid water content (LWC) values (unit: $\mathrm{g \ m^{-3}}$) as the threshold for defining in-cloud conditions.

[Figure]

Figure S3. Similar to Figure 3 a, number of samples for four transition phases but using different ice water content (IWC) values (unit: g m$^{-3}$) as the threshold for defining in-cloud conditions.

*8. Both theoretical and observational studies (Korolev and Mazin, 2003; Korolev and Isaac, 2006) showed RHliq in mixed phase clouds is close to 100%. Due to the short time of phase relaxation (typically 0.1-10s) in liquid and mixed-phase clouds, the evaporating droplets will rapidly bring the system of "droplets-water vapor" to quasi-equilibrium and saturate the environment. In this regard, the observations at -25C in liquid and mixed-phase clouds (with no ice) of RHliq ~88%, 82% and 75%, respectively (Fig.10b), is suggestive of large biases in RHliq measurements. The low accuracy of RHliq does not allow for the conclusions made in the paper about the relationships between humidity and microphysical parameters of cloud type (1)-(4).*

We appreciate the concern from the reviewer. We took a few approaches to address the concerns about the relative humidity measurements.

First, we would like to mention that in the summer of 2016 and 2018, we spent 3 months each year at the National Center for Atmospheric Research / Earth Observing Laboratory (NCAR/EOL) to conduct laboratory calibration of the VCSEL water vapor hygrometer. Below is a schematic diagram (**Figure R2-1**) and some results based on these calibrations. Even though such laboratory calibration has improved the original water vapor data, we note that combining the uncertainties from temperature data and water vapor data, the RHi and RHliq data would still have about 6% - 7% uncertainties.

[Figure]

Figure R2-1. Laboratory calibration of the VCSEL hygrometer conducted by M. Diao. (a) A schematic diagram of the calibration system using the NCAR EOL environmental chamber. (b-e) An example of RH frequency distribution for research flight 10 in the NSF SOCRATES campaign. (b) RHliq frequency distribution that peaks at 96% for liquid and mixed phase clouds and (d) RHice frequency distribution that peaks at 106% for ice phase clouds using the older calibrations from Minghui Diao and Josh DiGangi at Princeton University in 2013. (c) New RHliq and (e) RHice frequency distributions using the Diao (2021) calibration for the SOCRATES campaign that peak at 101% and 99%, respectively. The new calibration of Diao (2021) has improved the peak position of RHliq and RHice for in-cloud conditions.

Second, we have noticed that the sub-saturated conditions for in-cloud conditions are detected not only by several NSF flight campaigns using the VCSEL hygrometer, but also by several other campaigns using completely different instruments and aircraft platforms. We speculate that there is some real physical explanation for these sub-saturated conditions. For example, it is possible that within a 1-second aircraft measurement, which is usually ~200 m resolution, the liquid droplets inside this segment may not immediately equilibrate with the entire volume of this segment for it to reach liquid saturation. In **Figure R2-2** below, we show the probability density function (PDFs) of RHi for three flight campaigns, NSF SOCRATES, NASA SEAC4RS and DOE ACME-V campaigns. We noticed that using two thresholds to define in-cloud conditions, i.e., TWC > $10^{-5}$ or > $10^{-3}$ g m$^{-3}$, the in-cloud RHi frequency distributions consistently show some subsaturated conditions. The NASA SEAC4RS campaign was based on NASA DC-8 research aircraft, and the water vapor was measured by the NASA Diode Laser Hygrometer (DLH) instrument. The DOE ACME-V was based on the DOE Gulfstream-1 research aircraft, using the Cavity Ring Down (CRD) instrument to measure gas phase including water vapor. This research topic regarding the high-resolution spatial variability of water vapor in relation to the spatial heterogeneity of the cloud hydrometeor distribution would be an interesting topic for high-resolution observations or LES simulations. However, limited by the available 1-Hz measurements in the SOCRATES campaign, we feel that testing such hypothesis on the existence of subsaturated conditions at sub-1 Hz resolution requires another study using different types of datasets at higher resolution. Nevertheless, we quantified the impacts of these sub-saturated conditions on our main conclusions in this work, which is discussed in the following paragraph.

[Figure]

Figure R2-2. Probability density functions (PDFs) of RHi and vertical velocity for three flight campaigns – NSF SOCRATES, NASA SEAC4RS and DOE ACME-V. (a-f) Clear-sky, in-cloud and all-sky conditions, with in-cloud conditions defined by TWC > $10^{-5}$ g m$^{-3}$. (g-l) Similar to the left panels but using TWC > 0.001 g m$^{-3}$ as the in-cloud threshold.

Third, we conducted a sensitivity test to the number of samples of four phases in **supplemental Figure S4**, using different restrictions of RH$_i$ data, i.e., RH$_i$ greater than 50%, 70%, 75%, 80%, 85%, and 90%. The results show similar distributions for the number of samples among these restrictions especially for RH$_i$ > 70%. In addition, for the main analysis, we applied a sensitivity test using only RH$_i$ > 80% and the results are shown in **supplemental Figures S7 – S9**, which are very similar to the results shown in Figures 7, 8 and 9, respectively. We contrast these analyses below in **Figure R1-1, R1-2, and R1-3**.

In summary, even though we cannot 100% verify the reasons behind these sub-saturated conditions for in-cloud samples, either including or excluding these sub-saturated conditions does not show a large impact on our main conclusions, which indicates that these main results are robust. We added a comment on these sub-saturated conditions in section 3.2: "For phase 1, a small amount of ice sub-saturated conditions is seen. Besides the 6% – 7% uncertainties in RH$_i$ values that originate from the combination of water vapor and temperature measurement uncertainties, the deviation of relative humidity from liquid saturation line may be due to spatial variability of RHi at sub-1 Hz resolution and/or lower LWC to provide sufficient vapor-liquid phase equilibrium at a 200-m scale."

We also added comments in section 3.2: "Previous theoretical and observational studies (Korolev and Mazin, 2003; Korolev and Isaac, 2006) showed that RH$_{liq}$ in mixed-phase clouds is close to 100%, due to evaporating droplets rapidly via the WBF process, bringing the system of "droplets-water vapor" to quasi-equilibrium and therefore saturating the environment. As liquid droplets glaciate into ice particles, the peak of RH frequency would also shift towards ice saturation (e.g., D'Alessandro et al., 2019). The in-cloud samples used in this study contain some sub-saturated conditions that deviate from liquid saturation in phases 1 – 3 or from ice saturation in phase 4 (as shown in Figure 4 a – d), which may be attributed to a combination of reasons, such as 6%–7% uncertainties in RH values originated from water vapor and temperature measurement uncertainties, heterogeneous distributions of LCR, MCR and ICR that lead to an uneven distribution of supercooled liquid water, as well as non-equilibrated states between vapor/liquid or vapor/ice phase due to a larger volume being sampled by fast aircraft measurements (~172 m horizontal resolution for 1-Hz measurements used here)."

Also in section 3.3: "To assess the impacts of the sub-saturated conditions within TCR on the main findings of this work, we examine the impacts of excluding the lower RH$_i$ samples from the analysis of cloud micro- and macrophysical properties. Supplemental Figures S7 and S8 show the results of excluding RH$_i$ < 80% based on the analysis similar to Figures 7 and 8, respectively. The relationships of ice particle number fraction, IWC, LWC and ice mass fractions with respect to mixed spatial ratio and ice spatial ratio show similar results when lower RH$_i$ are excluded, demonstrating the robustness of these main conclusions."

We also commented in section 3.4: "The analysis of Figure 9 is also conducted for only RH$_i$ > 80% in supplementary Figure S9 and the results show consistent results of the cloud phase frequency distributions regardless of the exclusion of low RH$_i$ values."

[Figure]

Figure R1-1. Side-by-side comparisons of using all RH$_i$ data (left) versus using only high RH$_i$ (>80%) (right). The figure on the left is Figure 7, on the right is Figure S7.

[Figure]

Figure R1-2. Side-by-side comparisons of using all RH$_i$ data (left) versus using only high RH$_i$ (>80%) (right). The figure on the left is Figure 8, on the right is Figure S8.

[Figure]

Figure R1-3. Side-by-side comparisons of using all RH$_i$ data (top) versus using only high RH$_i$ (>80%) (bottom). The figure on the top is Figure 9, on the bottom is Figure S9.

*9. Numerous in-situ observations (including those, cited in the present study e.g., Wang et al. 2020) showed that in stratiform clouds the distribution of the vertical wind is centered close to zero. A visual assessment of the diagram in Fig.10c suggests systematic biases of the vertical wind with an average speed of ~-0.2m/s or lower. For mixed-phase clouds (type 2) at -25C the biases in vertical wind reached -0.5m/s and ~-0.9m/s. Clouds subsiding at such speed are expected to evaporate within a relatively short time due to adiabatic heating. Thus, for LWC(0)=0.1g/m3 at -10C, a cloud parcel descending at 0.2m/s will evaporate within 8 minutes.*

We thank the reviewer for this inspection of the vertical velocity. We examined the distributions of vertical velocity for clear-sky, in-cloud, and all-sky conditions more closely and decided to apply a adjustment of +0.125 m/s to the 1-Hz vertical velocity observations. After this correction, the PDFs of velocity in are now centered at 0 m/s, which are illustrated **Figure R2-2** above as well as in supplemental **Figure S6**. This correction is mentioned in the main text in section 2.1: "The vertical velocity measurements are derived from several instruments, including Honeywell LASEREF IV Inertial Reference Unit, radome pressure, static pressure, pitot tubes, temperature probe, and differential Global Positioning System with an accuracy of ~±0.15–0.30 m/s and precision ~0.01 m/s (Diao et al., 2015). When examining the in-cloud and clear-sky conditions in the SOCRATES campaign, we noticed a low bias of the original vertical velocity measurements, and therefore applied a correction of +0.125 m/s for the vertical velocity values. After this correction, the peak of the frequency distributions of vertical velocity is centered at 0 m/s for both in-cloud and clear-sky conditions." After correcting the w distributions, the original Figures 9 and 10 are now updated and combined into the new **Figure 5** (below).

[Figure]

Figure 5. Distributions of (a) RH$_i$, (b) RH$_{liq}$, (c) vertical velocity (w), and (d) standard deviation of vertical velocity ($\sigma_w$) for various transition phases at different temperatures. (e-h) Similar to (a-d), but in relation to various mixed spatial ratios or ice spatial ratios. Phases 1 and 4 show ice spatial ratio at 0 and 1, respectively, and therefore only a single dot is shown for phases 1 and 4 in (e-h).

In addition, we would like to note that even though the mean w values seem to have more downdrafts when plotting w against ice spatial ratio or mixed spatial ratio (new Figure 5), the number of samples are not evenly distributed in various bins of ice spatial ratio or mixed spatial ratio (new **supplementary Figure S5**). In fact, the PDF of w shows higher frequencies of updrafts than downdrafts for phases 1 – 3 (**Figure S6 b**). We added discussion on this point in section 3.2: "The mean values of w do not vary

significantly with mixed spatial ratio or ice spatial ratio (Figure 5 g), but the PDFs of vertical velocity in supplemental Figure S6 b show higher frequencies of updrafts for phases 2 and 3 compared with phases 1 and 4, meaning that the segments containing both supercooled liquid droplets and ice particles are subject to relatively more updrafts, compared with the segments containing only liquid droplets or only ice crystals. This finding is consistent with Shupe et al. (2008) which pointed out the importance of updrafts for sustaining mixed-phase clouds. Differing from the previous studies, our method can further specify that the highest updrafts and vertical velocity fluctuations are found in transition phase 3 when pure ice segments start to appear (~4.5 m/s in Figure S6 b and ~2.3 m/s in Figure S6 c), consistent with the fact that $RH_{liq}$ deviates more from liquid saturation in phase 3 (Figure 5 f), and therefore higher updrafts would be required to maintain supercooled liquid droplets."

[Figure]

Figure S6. Probability density functions (PDFs) of (a) relative humidity with respect to ice, (b) vertical velocity (w), and standard deviations of w ($\sigma_w$) separated by different transition phases. Phases 2 and 3 are further separated into seconds with or without ice in this analysis. Both phases 2 and 3 show higher frequencies of updrafts and $\sigma_w$ compared with phases 1 and 4.

*Clarity or presentation*

*There are several items that require an explanation or need a more detailed description.*

*10. What is the definition of a cloud employed in this study? E.g., LWC>X, or N>Y or something else?*

We define a 1-Hz sample as in-cloud condition when the TWC is greater than 0.001 g m$^{-3}$.

*11. In section 3.1, I had a hard time understanding what the total cloud region (TCR) is. Is it a cloud separated from other clouds by a clear sky segment? Or is it an entire cloud domain sampled during a field campaign? If it is the latter, was a cloud free environment included in the statistics? If TCR refers to separate clouds, then how was the calculation of cloud statistics performed? i.e., were TCRs normalized on their spatial extension?*

TCRs are individual, separate segments that are surrounded by clear-sky conditions. Each second within a TCR is counted as one 1-Hz sample, and all seconds are used in the rest of the analysis (e.g., new Figures 3 – 10) in this manuscript. We agree that this is an important concept and should be explained more clearly. We added a **supplemental Figure S1** and explained this in section 3.1: "In the second step, a total cloud region (TCR) that can potentially contain a combination of LCR, ICR and MCR is identified, which basically is a consecutive in-cloud segment surrounded by clear-sky conditions. If a TCR sample is surrounded by two adjacent seconds of NaN, then this sample is deleted, because one cannot determine if the NaN points are the edge of the cloud or if they are still part of the cloud. But if a TCR sample is surrounded by two adjacent seconds of clear-sky samples, then this in-cloud sample is valid, and its measurement can last from one second to many seconds. For instance, if five seconds of LCR are adjacent to one second of MCR, then both the LCR and MCR belong to the same TCR. An illustration of the identification of TCR is shown in supplemental Figure S1. All the 1-Hz samples within the TCR are used in the analysis in the following sections."

[Figure]

Figure S1. A schematic diagram that illustrates the identification of a total cloud region (TCR) sample, with liquid cloud region (LCR) and ice cloud region (ICR) embedded inside this TCR. All 7 seconds of samples inside this TCR are used in the analysis of cloud properties.

*12. Definition of mixed-phase clouds:*

*(a) The definition of mixed-phase based on LWC (or IWC) mass fraction LWC/TWC (or IWC/TWC) has been used in the cloud physics community for approximately thirty years. It is worth acknowledging this in the paper.*

We added acknowledgment on the history of this method in section 3.4: "This method of using ice mass fraction to define mixed-phase clouds has been used in the cloud physics community for approximately thirty years (e.g., Korolev et al., 1998; Korolev et al., 2017, their equation 5-1 and references therein)."

*(b) The second definition of mixed-phase, based on particle concentrations, is Nliq/( Nliq+Nice), where Nliq and and Nice are the concentrations of droplets and ice particles, respectively. Since, for most clouds, Nliq is typically larger than Nice by 3 to 5 orders of magnitude, with a very few exceptions the ratio Nliq/( Nliq +Nice) $\cong$ 1. Therefore, since Nliq/( Nliq +Nice)>0.9, the majority of clouds should fall in the category of liquid clouds. This is clearly inconsistent with the results shown in the diagram on Fig.4b. This contradiction requires an explanation.*

We thank the reviewer for catching this error. We initially made a mistake of not adding the CDP measured liquid droplets into the 2DS measured liquid droplets for calculating total liquid number

concentrations. We have corrected this mistake. The new **Figure 9** below shows higher liquid phase frequency when temperature is closer to 0°C using the ice number particle fraction method, compared with the other methods using ice spatial ratio or ice mass fraction.

[Figure]

Figure 9. Cloud phase occurrence frequencies at various temperatures. Cloud phase identification methods are based on (a) ice mass fraction per second, (b) ice particle number fraction per second, and (c) ice spatial ratio calculated for individual consecutive TCR.

*(c) The spatial ratio is defined as Length(cloud type)/Length(total). In this regard, the statement on line 165: " ...for each TCR, ice spatial ratio is calculated as length of (ICR+MCR) / length of TCR" sounds contradictory to this definition. If the definition of the spatial fraction is different from that stated above, then a more detailed explanation is required. Also, note that the spatial ratio was used for characterization of mixed-phase clouds in Korolev et al. (2017, Fig.5-13a).*

We can see how our definition of spatial ratio may be confused with the other way of defining spatial ratios in the previous studies. We clarified in the last paragraph of section 3.1 that the spatial ratio is calculated for each TCR segment, not for the bulk measurement of many cloud segments in a temperature bin: "In addition, we define two terms – mixed spatial ratio and ice spatial ratio, to represent the spatial fraction of ice-containing clouds in phases 2 and 3, respectively. Specifically, the mixed spatial ratio represents the fraction of MCR as part of an individual, consecutive TCR in phase 2, calculated as length of MCR / length of TCR. Ice spatial ratio represents the fraction of ice-containing segments as part of an individual, consecutive TCR in phase 3, calculated as (length of ICR + length of MCR * IWC/TWC) / length of TCR. The contribution of MCR to ice spatial ratio in phase 3 is weighted by the ice mass fraction, giving the MCR a smaller weighting function compared with ICR since MCR contains higher fractions of supercooled liquid droplets. Note that the definitions of mixed spatial ratio and ice spatial ratio differ from the spatial ratio previously used for characterization of mixed-phase clouds in Korolev et al. (2017, Fig.5-13a). In that previous method, the spatial ratio of a certain phase (liquid, mixed or ice) is calculated as the number of samples of that phase divided by the total cloud samples in a certain temperature bin. In this work, the mixed spatial ratio and ice spatial ratio are calculated for individual TCR segments, and therefore each TCR would produce one value for mixed spatial ratio and one value of ice spatial ratio. These values of mixed spatial ratio or ice spatial ratio are applied to every 1-second sample within this TCR."

*13. It would be beneficial for this work to discuss the effect of the WBF process and glaciation on the thermodynamic state of mixed-phase clouds. I found no mention of the glaciation process. The WBF was mentioned only once at the end of the paper.*

We added the new Figure 4 about the thermodynamic conditions (i.e., $RH_i$ distributions) for each transition phase, and added more comments on WBF and glaciation as mentioned above: "Previous theoretical and observational studies (Korolev and Mazin, 2003; Korolev and Isaac, 2006) showed that $RH_{liq}$ in mixed-phase clouds is close to 100%, due to evaporating droplets rapidly via the Wegner-Bergeron-Findeisen (WBF) process, bringing the system of "droplets-water vapor" to quasi-equilibrium and therefore saturating the environment. As liquid droplets glaciate into ice particles, the peak of RH frequency would also shift towards ice saturation (e.g., D'Alessandro et al., 2019). The in-cloud samples used in this study contain some sub-saturated conditions that deviate from liquid saturation in phases 1 – 3 or from ice saturation in phase 4 (as shown in Figure 4 a – d), which may be attributed to a combination of reasons, such as 6%–7% uncertainties in RH values originated from water vapor and temperature measurement uncertainties, heterogeneous distributions of LCR, MCR and ICR that lead to an uneven distribution of supercooled liquid water, as well as non-equilibrated states between vapor/liquid or vapor/ice phase due to a larger volume being sampled by fast aircraft measurements (~172 m horizontal resolution for 1-Hz measurements used here)." We mentioned glaciation in several other places, such as in section 3.3: "The decreasing ice crystal concentrations per size bin from phase 3 to phase 4 may be caused by stronger aggregation, sublimation, and/or sedimentation of ice crystals in phase 4, as well as by stronger glaciation and/or secondary ice production in phase 3… Phase 4 shows a trend of decreasing frequency of large ice particles (e.g., $D_{max}$ > 2000 µm) with decreasing temperature. This could be due to an increasing probability of droplet freezing with decreasing temperature given the same dimension that reduces the available amount of large supercooled liquid droplets for glaciation or riming at lower temperatures. On the other hand, phase 3, which still has supercooled liquid water coexisting with ice particles, does not show such trend, probably because ice crystal growth may occur via various processes in phase 3, such as WBF process, glaciation, vapor depositional growth under ice supersaturation, and/or riming."

*14. The rapid increase of occurrence of ice clouds in the temperature range -15C to -20C was observed by other research groups (e.g., Wallace and Hobbs 1975; Moss and Johnson 1994; and others), which is worth acknowledging here.*

We added this comment in the last paragraph of section 3.4: "The rapid increase of occurrence of ice clouds in the temperature range of -15°C to -20°C was also observed by previous studies (e.g., Wallace and Hobbs, 1977; Moss and Johnson, 1994)."

*15. It is worth indicating sampling statistics (cloud length) for each cloud type in Table 1.*

We revised Table 1 by adding these number of samples as recommended by the reviewer.

**Table 1.** Definitions of four transition phases of mixed-phase clouds, alongside their required spatial ratios of LCR, ICR, and MCR.

| Phase | Description | Number of seconds | Number of TCRs | Spatial Ratio of LCR | Spatial Ratio of ICR | Spatial Ratio of MCR |
|---|---|---|---|---|---|---|
| | | | | M1 = length of LCR / total segment length | M2 = length of ICR / total segment length | M3 = length of MCR / total segment length |
| 1 | Only LCR | 8243 | 1163 | M1 = 1 | M2 = 0 | M3 = 0 |

| 2 | MCR appears | 12557 (LCR: 11096, MCR: 1461) | 142 | 0 < M1 < 1 | M2 = 0 | 0 < M3 ≤ 1 |
|---|---|---|---|---|---|---|
| 3 | Pure ICR must appear | 11988 (LCR: 3478, MCR: 2973, ICR: 5537) | 249 | 0 ≤ M1 < 1 | 0 < M2 < 1 | 0 ≤ M3 < 1 |
| 4 | Only ICR | 8646 | 1193 | M1 = 0 | M2 = 1 | M3 = 0 |

*Concluding remarks*

*Given the amount of work invested in this study, I would encourage the authors to rewrite the paper accounting the above comments. I did not consider any minor comments since they are eclipsed by the major issues of this work. My biggest concern is related to the data quality issues. Fixing other issues is just a matter of time.*

*Alexei Korolev*

We added all the new citations recommended by the reviewer into the main text.

*Avramov, A., et al. (2011), Toward ice formation closure in Arctic mixed-phase boundary layer clouds during ISDAC, J. Geophys. Res., 116, D00T08, doi:10.1029/2011JD015910.*

*Baumgardner, D., and Coauthors, 2017: Cloud ice properties: In situ measurement challenges. Ice Formation and Evolution in Clouds and Precipitation: Measurement and Modeling Challenges, Meteor. Monogr., No. 58, Amer. Meteor. Soc., doi:10.1175/AMSMONOGRAPHS-D-16-0011.1.5*

*Fan, J., M. Ovtchinnikov, J. M. Comstock, S. A. McFarlane, and A. Khain, 2009: Ice formation in Arctic mixed-phase clouds: Insights from a 3-D cloud-resolving model with size-resolved aerosol and cloud microphysics. J. Geophys. Res., 114, D04205, doi:10.1029/2008JD010782.*

*Fan, J., S. Ghan, M. Ovchinnikov, X. Liu, P. J. Rasch, and A. Korolev, 2011: Representation of Arctic mixed-phase clouds and the Wegener-Bergeron-Findeisen process in climate models: Perspectives from a cloud-resolving study, J. Geophys. Res., 116, D00T07, doi:10.1029/2010JD015375*

*Field, P. R., R. J. Hogan, P. R. A. Brown, A. J. Illingworth, T. W. Choularton, P. H. Kaye, E. Hirst, and R. Greenaway, 2004: Simultaneous radar and aircraft observations of mixed-phase cloud at the 100 mscale. Quart. J. Roy. Meteor. Soc., 130, 1877–1904, doi:10.1256/qj.03.102*

*Field, P. R., A. A. Hill, K. Furtado, and A. Korolev, 2014: Mixed-phase clouds in a turbulent environment. Part 2: Analytic treatment. Quart. J. Roy. Meteor. Soc., 140, 870–880. doi:10.1002/qj.2175. Hobbs, P.V., and A. L. Rangno, 1985: Ice Particle Concentrations in Clouds. J. Atmos. Sci., 42, 2523-2549, DOI: https://doi.org/10.1175/1520-0469(1985)042<2523:IPCIC>2.0.CO;2*

*Hill, A. A., P. R. Field, K. Furtado, A. Korolev, and B. J. Shipway, 2014: Mixed-phase clouds in a turbulent environment. Part 1: Large-eddy simulation experiments. Quart. J. Roy. Meteor. Soc., 140, 855–869, doi:10.1002/qj.2177.*

Hogan, R. J., P. R. Field, A. J. Illingworth, R. J. Cotton, and T. W. Choularton, 2002: Properties of embedded convection in warm-frontal mixed-phase cloud from aircraft and polarimetric radar. Quart. J. Roy. Meteor. Soc., 128, 451–476, doi:10.1256/003590002321042054.

Korolev, A. V., and I. P. Mazin, 2003: Supersaturation of water vapor in clouds. J. Atmos. Sci., 60, 2957–2974, doi:10.1175/1520-0469(2003)060,2957:SOWVIC.2.0.CO;2.

Korolev, A. V., and G. A. Isaac, 2006: Relative humidity in liquid, mixed phase and ice clouds. J. Atmos. Sci., 63, 2865–2880, doi:10.1175/JAS3784.1.

Korolev, A. V., and P. R. Field, 2008: The effect of dynamics on mixed-phase clouds: Theoretical considerations. J. Atmos. Sci., 65, 66–86, doi:10.1175/2007JAS2355.1.

Korolev, A.V., E. Emery, J. W. Strapp, S. G. Cober, and G. A. Isaac, 2013b: Quantification of the effects of shattering on airborne ice particle measurements. J. Atmos. Oceanic Technol., 30, 2527–2553, doi:10.1175/JTECH-D-13-00115.1.

Korolev, A., McFarquhar, G., Field, P. R., Franklin, C., Lawson, P., Wang, Z., et al. 2017:. Mixed-phase clouds: Progress and challenges. Meteorological Monographs, 58, 5.1–5.50. https://doi.org/10.1175/AMSMONOGRAPHS-D-17-0001.1

Korolev, A., & Milbrandt, J., 2022: How are mixed-phase clouds mixed? Geophysical Research Letters,49, e2022GL099578. https://doi.org/10.1029/2022GL0995786

Morrison H, de Boer G, Feingold G, Harrington J, Shupe MD, Sulia K. 2011: Resilience of persistent Arctic mixed-phase clouds. Nature Geosci. DOI:10.1038/ngeo1332.

Moss, S. J. and Johnson, D.W. 1994 Aircraft measurements to validate and improve numerical model parametrization of ice to water ratios in clouds. Atmos. Res., 34, 1–25

Pinto, J. O., 1998: Autumnal mixed-phase cloudy boundary layers in the Arctic. J. Atmos. Sci., 55, 2016–2037, doi:10.1175/1520-0469(1998)055,2016:AMPCBL.2.0.CO;2.

Rauber, R.M, Tokay A. 1991: An explanation for the existence of supercooled liquid water at the top of cold clouds. J. Atmos. Sci. 48: 1005–1023.

Smith, A.J, Larson V.E, Niu J, Kankiewicz J.A, Carey L.D. 2009: Processes that generate and deplete liquid water and snow in thin midlevel mixed-phase clouds. J. Geophys. Res. 114: D12203, DOI: 10.1029/2008JD011531

Wallace, J. M. and Hobbs, P. V. 1975 Atmospheric Science: An introductory survey. Academic Press, New York, USA

Westbrook, C. D., and A. J. Illingworth (2011), Evidence that ice forms primarily in supercooled liquid clouds at temperatures > −27°C, Geophys. Res. Lett., 38, L14808, doi:10.1029/2011GL048021.

**Response to comments from Reviewer 3**

*AMT review 3 comments*

*The purpose of this study is to evaluate varying cloud properties during the evolution of mixed phase clouds (i.e., from the first appearance of ice within supercooled liquid clouds to their complete glaciation). It tests for this by determining the ratio of liquid, mixed, and ice phase samples within a continuous cloud sample. Some important findings are revealed by the analysis, such as the vertical air motion is much more variable at the first appearance of ice at temperatures less than -20°C compared with other parts of the evolution, suggesting dynamical factors may be a significant factor for ice initiation.*

*The research topic is very important, as there are still large uncertainties associated with the evolution of mixed phase properties; and the novel approach qualitatively determining the stage of macroscale mixed phase evolution could be valid for a statistical analysis as undertaken in the study (assuming that once ice occurs within a supercooled liquid cloud, it will tend towards complete glaciation). This stage classification is often combined with the spatial extent of mixed and ice phase samples (ice spatial ratio). Findings can vary from quite insightful and perhaps striking (e.g., similar rates of increase in ice water content within mixed phase samples and also ice phase samples with increasing ice spatial ratio), to rather speculative (e.g., a key step for the Wegener-Bergeron-Findeison process to occur is when pure ice segments are present).*

We thank reviewer 3 for the helpful comments and below is our individual response.

We revised the original sentence about the Wegener-Bergeron-Findeisen (WBF) process mentioned by the reviewer: "As ice crystals grow into pure ice segments (i.e., ICR), liquid phase starts to rapidly transition into ice phase, suggesting that the formation and growth of ice particles become more significant when pure ice segment appears."

*Aside from the analyses performed, there are major concerns associated with their methodology listed below:*

*1)    Specific combinations of LCR, MCR, and ICR ratios may be rather ambiguous. For example, are we sure a phase 2 cloud segment with >97.5% MCR is still considered to be in the earlier stage of glaciation than a phase 3 cloud segment with <2.5% ICR and >97.5% LCR? There is the potential for extreme ambiguity in what part of the mixed phase evolution a cloud region is currently in based on the current framework.*

We agree with the reviewer that the previous analysis using ice spatial ratio as the transition indicator for both phases 2 and 3 can lead to an ambiguity. Therefore, in the revised manuscript, we use the definition of "mixed spatial ratio" for analysis of phase 2, to distinguish from ice spatial ratio in phase 3. Mixed spatial ratio is defined as mixed-phase cloud region (MCR) length over total cloud region (TCR) length i.e., mixed spatial ratio = length of MCR / length of TCR, calculated for each consecutive TCR.

In the first version of the manuscript, ice spatial ratio (ISR) was defined as: ISR = (ice cloud region ICR length + MCR length) / TCR length, calculated for each consecutive TCR. In that previous definition, ICR and MCR carry the same weight when their spatial extent contributes to spatial fraction of ice-containing segments. We revised it to: ISR = (ICR length + MCR length * IWC/TWC) / TCR length, calculated for each consecutive TCR. In this revised definition, MCR's contribution to ice-containing segment spatial extent is reduced since its average ice mass fraction is less than 1.

All the figures are revised according to these new definitions of mixed spatial ratio and ice spatial ratio mentioned above. The new **Figure 7** is shown as an example using the revised mixed spatial ratio and ice spatial ratio below.

[Figure]

**Figure 7.** Relationship between ice particle number fraction and mixed spatial ratio or ice spatial ratio, separated by the transition phases (phase 2 in column 1 and phase 3 in column 2), and by various cloud segments – (a, b) LCR, (c, d) MCR and (e, f) ICR. Average values for each ice spatial ratio bin are shown in black solid lines, with vertical bars representing standard deviations. Linear fit is shown in red dashed line. Average values of generating cells (time series obtained from Wang et al. (2020)) are in pink "X" markers. The slope value b, its associated standard deviation, and the ordinary R-squared value are shown in the legend.

*2)    This study does not appear to account for whether the aircraft is sampling within the cloud or the precipitation underlying a given cloud. So a primarily liquid phase cloud (with few mixed phase samples) could be precipitating ice and if the aircraft samples the precipitation, it will be considered phase 3, although the cloud itself would be phase 2. Not to mention, the aircraft could have a majority of cloud samples/an entire length of continuous cloud sampling as precipitation. Including precipitation likely accounts for subsaturated conditions for both liquid and ice phase samples in Figures 10 and 11.*

We agree with the reviewer that the previous analysis did not account for the potential precipitation underlying a given cloud. To identify these precipitating clouds, we analyzed two new airborne remote sensing instruments – the HIAPER Cloud Radar (HCR) and High Spectral Resolution Lidar (HSRL). We used the particle identification (PID) value added product provided by the NCAR Earth Observing Laboratory (EOL) team to identify and remove precipitating samples. The PID includes 11 types of conditions. We made time series figures for every hour of each research flight (RF) and zoomed into each segment to manually inspect possible precipitating samples.

We added the text in section 2.1: "To provide a more focused analysis of cloud layers instead of precipitation below the clouds, we use two remote sensing instruments onboard the G-V aircraft – NSF/NCAR High-performance Instrumented Airborne Platform for Environmental Research (HIAPER) Cloud Radar (HCR) and High Spectral Resolution Lidar (HSRL) to identify potential precipitating samples. The particle identification (PID) product is used, which includes identifications of 11 categories – rain, supercooled rain, drizzle, supercooled drizzle, cloud liquid, supercooled cloud liquid, melting, large frozen, small frozen, precipitation and cloud (Romatschke and Vivekanandan, 2022). By manually inspecting hourly time series of this product, we remove segments that are identified as precipitation, supercooled drizzle, drizzle, supercooled rain, and rain. In addition, we further examined the NSF SOCRATES campaign field catalogue for each flight to ensure that we do not miss any precipitation segments that have been identified in the field catalogue. The time stamps of the beginning and end of these segments are stored in supplemental Table S1. For most flights, we identified on average about 5 – 20 minutes of samples of precipitating regions, except RF15 which has about an hour of precipitating samples. It is worth noting that most of these segments occur at temperatures above 0°C, while this study only focuses on -40°C to 0°C."

**Table S1**. Time stamps of precipitating segments identified in the NSF SOCRATES campaign that are excluded from this study.

| Research flight (RF) | Time stamps of precipitation segments removed |
| --- | --- |
| RF01 | N/A |
| RF02 | 5:55 to 6:32 UTC |
| RF03 | 23:50 to 00:00 UTC, 00:00 to 00:05 UTC, 00:34 to 00:40 UTC, and 01:16 to 01:32 UTC |
| RF04 | 03:47 to 03:52 UTC |
| RF05 | 04:40 to 05:00 UTC |
| RF06 | 02:28 to 02:32 UTC |
| RF07 | 05:10 to 05:15 UTC |
| RF08 | 05:00 to 05:10 UTC |
| RF09 | 04:29 to 04:32 UTC |
| RF10 | 02:44 to 02:56 UTC, and 03:58 to 04:00 UTC |
| RF11 | 03:41 to 03:46 UTC |
| RF12 | 05:09 to 05:15 UTC |
| RF13 | 03:40 to 03:50 UTC, and 04:38 to 04:43 UTC |
| RF14 | 03:19 to 03:28 UTC, and 04:38 to 04:42 UTC |
| RF15 | 07:20 to 08:36 UTC |

*Minor concern:*

*1)    The lengths of the clouds used in the mixed phase evolution appear (note: the lengths are not clearly defined in the text) to not be set to a constant length, so some lengths could vary from two or three cloud samples (are single 1 Hz cloud samples without neighboring cloud samples removed?) to hundreds of*

*samples. This means processes associated with different length scales will have different impacts on different cloud lengths and may not result in an apples-to-apples comparison.*

This is a very good point. In fact, in the beginning stage of our analysis, we tested two ways of analyzing these segments. One way is to average each TCR cloud segment into a single datum point. But we decided to choose the other way for this analysis, which keeps each second of a segment as a separate sample, that way if a shorter segment only has a few seconds while a longer segment has hundreds of seconds, the longer segment will carry a higher weight since it will provide more 1-Hz samples for the final analysis. The latter method reflects the radiative impacts in the real atmosphere better, since clouds with larger spatial extent also have larger radiative effects. We added a supplemental Figure S1 to illustrate this method more clearly.

With that said, we agree that the length of the segment needs to be clarified in the manuscript better. We also would like to clarify regarding the reviewer's concern about "*are single 1 Hz cloud samples without neighboring cloud samples removed*". This indeed has been discussed in our team in the method development stage, but we didn't mention this detailed filtering process in our first manuscript, and now we added this process in the section 3.1: "In the second step, a total cloud region (TCR) that can potentially contain a combination of LCR, ICR and MCR is identified, which basically is a consecutive in-cloud segment surrounded by clear-sky conditions. If a TCR sample is surrounded by two adjacent seconds of NaN, then this sample is deleted, because one cannot determine if the NaN points are the edge of the cloud or if they are still part of the cloud. But if a TCR sample is surrounded by two adjacent seconds of clear-sky samples, then this in-cloud sample is valid, and its measurement can last from one second to many seconds. For instance, if five seconds of LCR are adjacent to one second of MCR, then both the LCR and MCR belong to the same TCR. An illustration of the identification of TCR is shown in supplemental Figure S1. All the 1-Hz samples within the TCR are used in the analysis in the following sections. The length of each second of sample within an TCR is calculated based on the aircraft true air speed at that specific second. The length of each TCR is calculated as the sum of all in-cloud samples within that TCR. The mean true air speed of the G-V research aircraft between -40°C and 0°C during the SOCRATES campaign is ~172 m/s."

[Figure]

Figure S1. A schematic diagram that illustrates the identification of a total cloud region (TCR) sample, with liquid cloud region (LCR) and ice cloud region (ICR) embedded inside this TCR. All 7 seconds of samples inside this TCR are used in the analysis of cloud properties.

*I think a good start for addressing these issues is to take a good look at the individual total cloud regions: the distribution of their lengths as well as their phase ratios and how they look for the individual "phases of mixed phase evolution."*

As we clarified above, this work indeed examines individual, consecutive TCR and calculates the mixed spatial ratio or ice spatial ratio for each TCR. The distributions of the TCR lengths are added as two subpanels in **Figure 3,** which illustrates the number of samples and the frequency distributions of TCR lengths for 4 transition phases. The number of samples of the ice spatial ratios of individual TCRs is illustrated in new Figure 7 (shown in our response on a previous page), which shows the relationship between ice number particle fraction and mixed (or ice) spatial ratio.

[Figure]

Figure 3. Distributions of four transition phases at various temperatures in terms of (a) number of 1-Hz samples and (b) frequency of each phase. In (b), the frequency of each phase is normalized by the number of samples of all phases in each 5-degree temperature bin. (c) Number of 1-Hz samples and (d) frequency distribution of TCR lengths in logarithmic scale. In (d), frequency is calculated as the number of 1-Hz samples of a specific phase divided by the total number of 1-Hz in-cloud samples in each $10^{0.25}$ bin.

*There are also other major concerns aside from the mixed phase evolution methodology:*

*1)   The authors use the UHSAS probe to discern aerosol concentrations within the clouds. However, it appears as though aerosol measurements are taken within the cloud samples. This is problematic as there is a high likelihood the probe is sampling the residuals of cloud particles, and I suspect the positive correlation of aerosols with diameters greater than 500 nm within greater ICR is the UHSAS sampling cloud particle residuals. In order to use the UHSAS in the cloud, it is vital to provide sensitivity tests to confirm no such biases are occurring for both liquid and ice particles.*

We thank the reviewer for pointing out this potential issue. To address this issue, we first examined segments by segments of in-cloud samples of concurrent measurements of UHSAS probe and cloud probes, and we did not see any evidence of cloud hydrometeor mass or number concentrations correlated with aerosol number concentrations at 1-Hz resolution. In addition to this inspection, we revised **Figure 10** by applying a moving average on aerosol number concentrations for every 50 seconds, which is about 10-km resolution. We also tested the results when using the 100-second moving averages of aerosol number concentrations (**supplemental Figure S10**), which is about 20-km resolution. The relationships

between average aerosol number concentrations for diameters greater than 500 nm and 100 nm (i.e., $N_{>500}$ and $N_{>100}$, respectively) and cloud microphysical properties (i.e., LWC, IWC, Nliq, and Nice) are examined. Similar aerosol indirect effects on ice and liquid are shown in these new analyses compared with our previous analyses in the original manuscript. We added the comments in section 3.5: "Due to the possible complication of in-cloud measurements of aerosol number concentrations, we applied a moving average to calculate logarithmic scales of aerosol concentrations at every 50 seconds in Figure 10. A coarser spatial averaging using the 100-second moving average is also shown in supplementary Figure S10."

[Figure]

**Figure 10.** Similar to Figure 7, but showing logarithmic scale (a-h) $N_{>500}$ and (i-p) $N_{>100}$ in relation to mixed spatial ratio or ice spatial ratio, separated by the transition phases and cloud regions. The last row represents all cloud regions in a specific transition phase. The aerosol number concentrations represent the moving average values of every 50 seconds.

[Figure]

**Figure S10**. Similar to Figure 10 in the main manuscript but using 100-second moving averages of logarithmic scales of $N_{>500}$ and $N_{>100}$. The results using the coarser scale of aerosol number concentrations are very similar to those shown in Figure 10.

*2)    Phase frequencies are determined using a phase definition relating the phase fraction of liquid to the total number of particles using the UWLID product. However, it was stated that the 2DS has a size range of 40-5000um. Assuming the UWLID product provides phase information for this particle size range (does it? If so, the smaller particle sizes [<~200 um] should be associated with large uncertainties due to the loss of the 2DS imaging resolution), what about particles less than 40 um? It doesn't seem like the CDP measurements are incorporated into the analysis, which could significantly impact number concentration ratios.*

We thank the reviewer for catching this error. That is correct, we previously did not add the CDP measurements into the number of liquid droplets for this calculation. In the revised manuscript, we calculated the number concentrations of both liquid droplets and ice particles from 2DS, then added the liquid droplet number concentration together with the CDP liquid droplet number concentrations. This new calculation affects new **Figures 7 and 9** which both contain the ice particle number fraction analysis. We clarified this calculation in section 2.2: "We use the hydrometeor count defined by the maximum diameter in the UWILD dataset to calculate Nliq and Nice detected by the 2DS probe within each second. Then we further add Nliq detected by CDP to those detected by 2DS to derive the total Nliq. Finally, we define ice particle number fraction, which equals Nice / (Nice + Nliq) in one second."

*There were quite a few grammatical errors as well, and multiple citations were in error. And although there are some interesting findings presented in this study, I unfortunately cannot recommend that this manuscript be accepted for publication. I recommend doing a thorough revision of the paper and resubmitting.*

We appreciate the reviewer for the comments. We have conducted a more thorough proofreading of the text, corrected the grammatical errors and revised the reference list.

---

## Referee Report (RR1)

**Review of revised "The Transition from Supercooled Liquid Water to Ice Crystals in Mixed-phase Clouds based on Airborne In-situ Observations" by Maciel et al.**

**Overview**

I reviewed this paper more than a year ago, and it took me some time to go through my comments and replies and read the revised text. My original comments were split into three major categories: (a) Methodology and basic assumptions, (b) Data quality, and (c) Clarity or presentation. Many of my comments were addressed and clarified. However, after reading the revised text, I came across another set of issues. Most of the questions fall into the same three categories as stated above.

**Recommendation**: Unfortunately, the revised paper remains unsuitable for publication in ACP. I suggest another round of revision of the manuscript and addressing the comments listed below.

**Major comments**

1. This work is focused on the analysis of the link between the phase composition of clouds, cloud dynamics and aerosols. Clouds were split into four categories: pure liquid clouds (P1), conditionally mixed clouds consisting of spatially continuous liquid and genuinely mixed cloud segments (P2), conditionally mixed clouds consisting of spatially continuous segments of ice and (pure liquid and/or genuinely mixed phase) clouds (P3) and pure ice clouds (P4). In the first version of the paper, it was assumed that the direction of changes in the cloud thermodynamic state is as follows: (P1)=>(P2)=>(P3)=>(P4). However, as was indicated in the reviewer's previous comments, depending on the dynamic forcing, interaction between the cloud and ambient environment, and ice precipitation out of the cloud, the phase partitioning may go in any direction. The complexity of the interaction between three thermodynamic phases in clouds and its sensitivity to environmental conditions does not allow for simplified judgment about the evolution stage of the cloud. In this regard, the following statement in the conclusions (line 500-501)
*"Overall, the method proposed in this work provides a unique perspective to assess various evolution stages of mixed phase clouds, especially the transition from liquid to ice phase"* is an overstatement. This paper does not contain discussion of the cloud evolution. All that could be said is that the sampled cloud belongs to the one of the four preselected categories P1-P4. Linkage to the dynamics and humidity obtained from the instant in-situ (Eulerian) observations does not allow for judgment about the history of the cloud environment.

2. I have a hard time understanding the term "transition stage" throughout the manuscript. I brought this question up in my previous round of comments, however, I did not receive a clear answer. Employing the term "transition stage" implies that some clouds can exist in a non-transition stage. Generally speaking, any cloud can be described as an unstable colloidal system in a transition stage between water in gaseous and condensed stages (liquid and/or ice). There are several types of instabilities relevant to cloudy environments related to condensation/evaporation (e.g., due to dynamic forcing, entrainment & mixing, WBF, radiation effects, Ostwald ripening), mechanical interaction between particles (e.g., coalescence, aggregation, riming, fragmentation), and sedimentation. Each of these types of

instabilities is characterized by its own time scale. Specifically, in relation to this study, the use of the term "transition stage" assumes a discussion of time scales such as time of phase relaxation, glaciation time, and residence time of cloud particles, along with different types of forcing. However, none of these points have been discussed. Therefore, the use of the term "transition stage" is redundant and may be misleading to the reader.

3. Per the previous comment, the title of the paper, "*The Transition from Supercooled Liquid Water to Ice Crystals in Mixed-phase Clouds based on Airborne In-situ Observations*" is misleading. The paper does not discuss the transition from liquid to ice. In fact, the transition of the thermodynamic phase may go in the opposite direction, i.e., ice to mixed-phase. This was also mentioned in section 3.1 and indicated in Fig 1. This conflicts with the title of the paper, implying a one-directional transition, "liquid to ice".

4. Lines 505-507: "*Nevertheless, this method helps to provide a statistical categorization of different transition phases of mixed-phase clouds solely based on Eulerian-view sampling of aircraft data, which enables more detailed examination from a statistical, quasi-Lagrangian view that was not available previously.*"  There was no "*quasi-Lagrangian*" consideration of mixed-phase in the text, and this statement at the end of the paper is unexpected and confusing. I also have a hard time understanding how quasi-random sampling of clouds (e.g. Eulerian) can be linked to a quasi-Lagrangian consideration. What are the time and spatial scales of the quasi-Lagrangian consideration referred to?

5. Lines 267-270: "*Comparing RHi values in regions with and without ice, phase 2 shows higher RHi for regions with ice, while phase 3 shows higher RHi in regions without ice. This feature can be explained by the fact that higher RHi is required in order to initiate ice nucleation in phase 2, while ice crystals that continue to grow in phase 3 will further reduce RHi magnitude by vapor deposition.*" This is an unjustified statement. I believe the authors meant the dependence of INP nucleation on supersaturation. However, supersaturation in phase 2 (and any other type of cloud) is limited by saturation over liquid. This fact mitigates or eliminates the dependence of INP nucleation vs. vertical velocity in liquid and mixed-phase clouds. On the other hand, the differences between RHliq in P2 and P3 are within 1-4%. This is smaller than the accuracy of RH measurements (i.e., 6% - 7%). It applies limitations on relating these differences to physical processes, and it can be explained just by the error in RH measurements.

6. As can be seen from Figure 3a, the sampling statistics of measurements are distributed quite unevenly across the temperature range and cloud types P1-P4. The lengths of different cloud types in different temperature subranges vary from approximately 850km down to 1km or less. The points with low sampling statistics (e.g., less than 100 of 1Hz samples ~17km) have low statistical significance. This should be clearly discussed in the text.

7. Figs. 4 and 5 include clouds P2 and P3 with subdivisions in clouds "with ice" and "without ice". I have a hard time understanding what it means. Does it mean that P2 "without ice" are just liquid clouds with mixed-phase cloud regions (MCR) excluded from the data set? Whereas clouds P2 "with ice" are just P2 clouds? Does it mean that clouds P3 "without ice"

are clouds with excluded MCR and ICR? Or just with excluded ICR? Does it mean that P3 "with ice" is just P3 clouds or something else? With this ambiguity in the interpretation of the meaning of P2 & P3, "with ice" and "without ice" I found it difficult to follow the subsequent discussion.

8. Fig.3e shows that humidity in pure liquid clouds (P1, indicated by a red dot) is saturated over ice. This contradicts previous studies of humidity in liquid clouds (Korolev and Mazin, JAS, 2003; Korolev and Isaac, JAS, 2006, D'Alessandro et al., J.Clim., 2021). Something is fundamentally wrong here.

9. The diagram in Fig.S6a shows that the PDFs of RHi in liquid and mixed-phase clouds from this study are centered at saturation over ice, i.e., RHi=100%. This result is overly concerning. It raises many questions about the accuracy and data quality of RH measurements and results presented in the paper.

10. Could you please double-check that the points with sigma_w=1+ m/s for P2 clouds in Fig.3h are not related to the malfunctioning of the LASEREF? This point looks suspicious and very different from the rest of the points. This is an overly high value, which is relevant for strong convection. If the data quality of these points is justified, could you check the type of clouds?

11. Lines 273-275: "*For distributions of w in Figure 5 c, phase 1 has slightly higher w than other phases. Phases 2 – 4 show slightly negative average w values, suggesting weak downdrafts as the average condition in these phases*." This is a concerning statement. Subsiding clouds at the rate of 10cm/s to 40cm/s (Fig.5c) will dissipate within 5 to 20 minutes with initial LWC=0.1g/m3 at T=-10C. This estimated time scale of cloud dissipation is shorter than the sampling time of the cloud during the flight observations. After my previous comment about vertical measurements, the authors found a negative bias of the vertical velocity measurement in the clear sky at 0.125m/s. Along this way, could you please check the drift of w in the clear sky? Note that measurements of vertical wind during ascending, descending, and any other type of aircraft maneuvering may result in significant biases of w. It is also worth mentioning that the LASEREF is an internal system, and the vertical wind velocity is calculated from the aircraft acceleration, i.e., the aircraft body is used as a sensor. The accuracy of such measurements is relatively low.

12. Lines 275-276: "*For the σw distribution (Figure 5 d), regions with ice in phase 2 have the highest fluctuations of vertical velocity, indicating that stronger in-cloud turbulence induces high RHi (as shown in Figure 5 a), which further initiates ice nucleation in phase 2*." The last part of this statement relates the rate of INP nucleation with the vertical velocity, which induces higher RHi. This is an unjustified statement. Note that humidity in liquid clouds is always limited by quasi-steady supersaturation (i.e., Korolev and Mazin, 2003). In other words, in liquid clouds RHliq ~= 100%+ for a wide range of w. This is suggestive that the rate of the INP nucleation does not depend on w.

13. Lines 278-279: "*Such result is consistent with the finding of Buhl et al. (2019) which showed a positive correlation between IWC mass flux and vertical velocity fluctuation, but this study further illustrates that in-cloud turbulence is particularly important for transition phase 2 when ice crystals first start to appear inside MCR, surrounded by supercooled liquid water.*" There are several observational studies that show a correlation between ice and vertical velocity. However, the statement "*when ice crystals first start to appear inside MCR, surrounded by supercooled liquid water*" is an overstatement. This study did not present evidence to support it.

14. Fig. 5f shows that for P2 and P3 clouds, in most cases RHliq(no ice) <RHliq(with ice). Assuming that "no ice" category means "liquid" the results presented in Fig.5f contradict fundamentals of mixed-phase clouds, i.e., for the same environmental conditions, humidity in pure liquid clouds is expected to be higher compared to that in mixed-phase clouds. I am not sure what the cause of the obtained inequality is. However, in view of the importance of this result, an explanation of this phenomenon is required.

15. Lines 324-325: "… *because ice crystal growth may occur via various processes in phase 3, such as … vapor depositional growth under ice supersaturation*…" This is not true. For RHi=100%, $dM_{ice}/dt=0$.

16. In my previous comment, I brought up a concern regarding using the ice concentration fraction $\lambda_{ice}=N_{ice}/(N_{liq}+N_{ice})$ for the analysis on mixed-phase, where $N_{liq}$ and and $N_{ice}$ are the concentrations of droplets and ice particles, respectively. For most mixed phase cloud $N_{liq}$ >>$N_{ice}$ by a few orders of magnitude. Therefore, it is expected that in liquid clouds and mixed-phase clouds $\lambda_{ice}\cong0$, whereas in ice clouds $\lambda_{ice}\cong1$. The diagrams in Figs.7acd are consistent with this prediction, i.e. the cloud particle concentration in mixed-phase clouds is dominated by liquid droplet and therefore, $\lambda_{ice}\cong0$. However, diagrams in Figs.7bf caused questions. These diagrams show $\lambda_{ice}$ vs ice spatial ratio in P3 liquid and ice cloud regions (LCR & ICR). In Fig.7b ice concentration fraction varies in the range 0 < $\lambda_{ice}$ < 0.4, and in Fig.7f 0.3 < $\lambda_{ice}$ < 0.9. This is confusing, since in LCR the ice concentration fraction is expected to be $\lambda_{ice}\cong0$, whereas in ICR $\lambda_{ice}\cong1$. For the sake of argument, consider a case with $\lambda_{ice}\cong0.4$. Then, for the case of LCR (Fig.7b), the droplet concentration typical for the SOCRATES clouds $N_{liq}$ = 100 cm$^{-3}$ the concentration of ice particles will be $N_{ice} = \lambda_{ice} N_{liq} /(1- \lambda_{ice}) \cong 67$ cm$^{-3}$. To the best of my knowledge, such high concentrations of ice have never been reported in scientific literature. On the other hand, the presence of ice in P3 LCR raises questions about the accuracy of the identification of liquid cloud segments in P3. For the case of ICR (Fig.7f) assume $N_{ice} = 100L^{-1}$ (high end of ice concentration). Then $N_{ice} = N_{liq} (1- \lambda_{ice}) / \lambda_{ice} = 0.15$ cm$^{-3}$. This is an overly low concentration of liquid droplets. Such clouds are volatile, and they may exist at cloud interfaces, and their lifetime scale is expected to be short. Another question is related to measurements of such clouds. If the measurements were performed in ice clouds, then both CDP and 2DS can be contaminated by shattering artifacts, which can be confused with liquid drops. Please note, that both antishattering tips and antishattering algorithms are not capable of 100% filtering out all shattering artifacts (Korolev et al. JTECH, 2013).

17. The measurements of aerosol particles presented in section 3.5 were conducted by UHSAS inside clouds. I have a serious concern about this approach and the data quality. There is no need to say that aerosol inside clouds is modified by droplet activation, droplet collision-coalescence, scavenging by cloud particles, and precipitating out of the cloud. An example of aerosol processing can be seen in Fig.13b at https://doi.org/10.1175/2011BAMS3180.1. It is also not clear how a 50-second moving average would help eliminate this issue. Such averaging is expected to mix LCR, MCR, ICR, and out-of-cloud regions.

18. Please, replace the reference Korolev et al. JTECH, 1998 by Korolev, A.V., 1998: "About Definition of Liquid, Mixed and Ice Clouds." *FAA Workshop on Mixed-Phase and Glaciated Icing Conditions.* December 2-3, Atlantic City, NJ, 325-326.   This is a result of the error in referencing papers in Korolev et al. *AMS Monogr*. 2017, which propagated to the present study.

*Concluding remarks*
As in my previous comment, I did not consider any minor questions since they overlapped with the major issues of this work. My biggest concern remains the data quality of this paper.

Alexei Korolev

---

## Referee Report (RR2)

**Review: Partition between Supercooled Liquid Droplets and Ice Crystals in Mixed-phase Clouds based on Airborne In-situ Observations**

This study analyzes the in-situ observation of mixed-phase clouds collected during the SOCRATES field campaigns over the ocean. Each cloud segment is categorized into four phases: 1) liquid, 2) mixed-phase/liquid, 3) mixed-phase/liquid/ice, and 4) ice. The dependency of microphysical cloud properties, dynamical properties, and aerosol properties on each of these phases is examined. The paper introduces mixed and ice spatial ratios to describe the evolution of the phases.

This paper presents an intriguing approach for analyzing in-situ data of mixed-phase clouds. However, there are concerns regarding the equal treatment of cloud segments between 0.2 km and 180 km (see major comment). The quality of the presentation could be enhanced by focusing on fewer figures and discussing these figures more comprehensively (see minor comments).

**Recommendation:** I suggest reconsidering the paper after making major revisions.

**Major:**

Cloud segments vary in length from 0.2 to 180 km. What is the likelihood that a short cloud segment is incorrectly classified as liquid (phase 1) due to the low measurement volume missing ice crystals? Is there a possibility that the two edges of a long cloud segment interacted with each other? If not, what justifies treating them as one quantity in the analysis? How do the results depend on the length of the segments? Would splitting long cloud segments into smaller pieces (e.g., 1000 m), where cloud particles interact with each other, be advantageous?

**Minor:**

Abstract: Specify the dataset used and the types of clouds investigated.

Line 180: Figure S1 is crucial for understanding the approach and should be moved to the main manuscript.

Line 184: The introduction of M1, M2, M3 is confusing and unnecessary as it only appears in Table 1.

Line 215: It should be a second-by-second analysis. Could an analysis of larger intervals (e.g., 10 seconds) provide a better understanding of the cloud phases?

Line 225: "The lengths of cloud segments vary…"

Line 225: After the sampling statistic, I expected the number of samples, not a time.

Line 236 – 238: A mixed-phase cloud (MPC) consists of supercooled droplets and ice crystals. Which spatial fraction describes the macrophysical properties of MPCs? How do LCR, ICR, MCR, and TCR represent macrophysical properties, and why aren't they used in the analysis?

Line 240 – 245: Is it correct that the mixed spatial ratio equals the spatial ratio of MCR (M3), but the ice spatial ratio differs from the spatial ratio of ICR? If yes, try to find clearer names and perhaps add the definition of mixed spatial ratio and ice spatial ratio to Table 1.

Line 270 – 272: What is the percentage of observations over 1.25 m/s in phase 2 and 3? Is the difference significant? I suggest moving Figure 4 i-p to the appendix and Figure S4 b-d to the main manuscript.

Line 295 – 296: Which phase are you referring to? I suggest plotting all temperatures of the size distribution of this phase in one plot to emphasize the differences. As differences in temperature are not further discussed, I suggest moving the size distribution of the temperature interval to the appendix and showing only the size distribution of all temperatures in the main manuscript.

Line 305 – 306: What is the fraction of observations with ice particle number fraction > 0.1 in Figure 6b? Why was the linear regression calculated on the mean of each ice spatial ratio bin (which weighted each bin equally despite very different numbers of observations in each bin) and not based on individual observations? Please add more information on the calculation of the linear regression.

Line 315 – 320: I have difficulty following the argument. Are you referring to "ice crystals gradually dominating the total particle population" as high ice particle number fraction? How can "a particular TCR" be identified in the multiple subfigures? What is the spatial extent of the entire cloud segment?

Line 323 – 324: How can "ICR appear" when phase 3 always has some ICR?

Line 325: Please describe in more detail how you derived that "they experience similar rates of phase changes from liquid to ice" based on measurements of individual states of cloud microphysics?

Line 336: How do you conclude that generating cells contain lower ice particle number fractions? What is the uncertainty of the generating cells measurements?

Line 348-349: What do you mean by "... similar rate of increase between ice crystals embedded among supercooled liquid droplets..."?

Line 352: Should be Figure 7i.

Line 357 – 358: What effect would the formation and growth of ice particles have, and could the depletion of the liquid phase also play a significant role?

Line 405: Why should SIP be stronger in phase 3 when phase 2 has more large droplets (according to Fig. 5)?

Line 447-448: Please be precise if you are referring to MCR/ICR or phase 2/3.

Line 468- 476: Move this paragraph to the definition of the phases.

Line 480 – 484: Are you suggesting that once a small pocket of pure ice crystals appears in a cloud (phase 3), the rate of change from liquid to the ice phase accelerates for the whole cloud?

---

## Referee Report (RR3)

**Manuscript Title: "Partition between Supercooled Liquid Droplets and Ice Crystals in Mixed-phase Clouds based on Airborne In-situ Observations"**

**Key Scientific Question:** The manuscript addresses the question of how the macrophysical and microphysical properties of mixed-phase clouds, and the factors controlling their formation and evolution, impact their radiative forcing over the Southern Ocean. This research aims to understand the interaction between supercooled liquid water and ice crystals within these clouds and how aerosols influence these properties.

The study introduces a novel method for categorizing mixed-phase clouds into four distinct phases based on spatial relationships among segments containing pure ice (ICR or phase4), pure liquid (LCR or phase1), or both, liquid dominated mixed-phase (MCR or phase 2) and ice-dominated mixed-phase MCR or phase3).

Key findings include positive correlations between ice particle number fraction and ice water content (IWC) with mixed and ice spatial ratios in phases 2 and 3, with phase 3 showing faster changes. All methods identified a significant phase transition around $-17.5°C$. Larger aerosols were found to be more likely to act as ice nucleating particles (INPs), with phase 3 exhibiting weaker aerosol indirect effects due to secondary ice production. Higher updrafts and stronger in-cloud turbulence were observed in mixed-phase conditions, particularly in phase 3. These insights suggest that future climate models should account for varying phase change rates and spatial fractions of ice-containing regions.

However, the method has limitations, including its idealized nature, reliance on 1-D aircraft data, and potential lack of comprehensive spatial representation. Future research should integrate 2-D and 3-D observations (remote-sensing) and simulations to validate and refine the phase categorization method, enhancing our understanding of mixed-phase cloud dynamics and their climate impacts.

**Minor Comments:**

- Line 266: Typo → $- 0.1 - km$ → $0.1 - 1$ km
- Line 280: Typo → Wegner → Wegener
- Line 485: "Because of this, aerosol indirect effects on various stages of clouds can also be examined separately." - Because of what? Consider rephrasing for clarity!

**Conclusion**

The manuscript presents significant advancements in the understanding of mixed-phase clouds and their classification. The novel method proposed is promising and provides valuable insights into cloud microphysical and macrophysical properties. Addressing the major remarks and refining the methodology could further enhance the impact and robustness of the study. Overall, the manuscript is a valuable contribution to the field of cloud physics and climate science and is recommended for publication with minor revisions.

---

## Author Response (AR2)

*Responses to the Reviewers*

Format: The reviewer's comments are quoted in italic. Quotation in red color stands for revised/added text in the revised manuscript.

*Overall comment:*

We thank the reviewer for the detailed comments, especially after a long time since your first review. We have addressed individual comments from the reviewer in our response. Specifically, we have addressed these following main comments:

1.  We improved the data quality of relative humidity measurements by applying a new data quality control procedure for in-cloud measurements with a high amount of supercooled liquid water. That is, when the 1-Hz sample is categorized as liquid-containing (i.e., LCR or MCR) and for that second either CDP probe or King probe reported LWC > 0.001 g m$^{-3}$, the $RH_{liq}$ values are adjusted to liquid saturation. The $RH_i$ and $RH_{liq}$ distributions are shown in the new Figure 4.
2.  We examined the data quality of vertical velocity, especially for cases when the research aircraft was doing vertical profiles or when it stayed at constant pressures. We found no drifting issues with vertical velocity measurements. But to be more cautious, we applied a new data quality control procedure to the analysis of $\sigma_w$ (i.e., standard deviation of vertical velocity calculated for every 40 seconds), i.e., we remove the $\sigma_w$ values when the research aircraft experienced dPressure > 10 hPa within those 40 seconds.
3.  We also improved the calculation of ice particle number fraction (IPNF), which equals Nice / (Nice + Nliq). Specifically, we used an additional cloud imaging probe – the PHIPS probe, to verify cloud particle thermodynamic phases for each phase and each type of cloud region (LCR, MCR or ICR). We applied a series of new quality control procedures to remove the mistakenly reported ice or liquid from the UWILD 2DS data, which have significantly reduced the number of seconds with IPNF between 0.4 and 1, which are the scenarios that the reviewer was concerned about. New Figures 6 and 8 reflected such change, with IPNF values peak at 0 and 1. A sensitivity test to Figure 6 is also conducted by removing all high IPNF values, and our conclusions remain the same.
4.  We revised the title again, reduced the ambiguity in our writing, and rephrased the text that the reviewer had concerns about.
5.  We also revised the line colors for any line plots such as new Figures 3, 5, 8 to satisfy the color scheme requirement by the journal of Atmospheric Measurement Techniques (AMT).

*Response to comments from the Reviewer*

*Review of revised "The Transition from Supercooled Liquid Water to Ice Crystals in Mixed- phase Clouds based on Airborne In-situ Observations" by Maciel et al.*

*Overview*
*I reviewed this paper more than a year ago, and it took me some time to go through my comments and replies and read the revised text. My original comments were split into three major categories: (a) Methodology and basic assumptions, (b) Data quality, and (c) Clarity or presentation. Many of my comments were addressed and clarified. However, after reading the revised text, I came across another set of issues. Most of the questions fall into the same three categories as stated above.*

*Recommendation: Unfortunately, the revised paper remains unsuitable for publication in ACP. I suggest another round of revision of the manuscript and addressing the comments listed below.*

*Major comments*

*1. This work is focused on the analysis of the link between the phase composition of clouds, cloud dynamics and aerosols. Clouds were split into four categories: pure liquid clouds (P1), conditionally mixed clouds consisting of spatially continuous liquid and genuinely mixed cloud segments (P2), conditionally mixed clouds consisting of spatially continuous segments of ice and (pure liquid and/or genuinely mixed phase) clouds (P3) and pure ice clouds (P4). In the first version of the paper, it was assumed that the direction of changes in the cloud thermodynamic state is as follows: (P1)=>(P2)=>(P3)=>(P4). However, as was indicated in the reviewer's previous comments, depending on the dynamic forcing, interaction between the cloud and ambient environment, and ice precipitation out of the cloud, the phase partitioning may go in any direction. The complexity of the interaction between three thermodynamic phases in clouds and its sensitivity to environmental conditions does not allow for simplified judgment about the evolution stage of the cloud. In this regard, the following statement in the conclusions (line 500-501)"Overall, the method proposed in this work provides a unique perspective to assess various evolution stages of mixed phase clouds, especially the transition from liquid to ice phase" is an overstatement. This paper does not contain discussion of the cloud evolution. All that could be said is that the sampled cloud belongs to the one of the four preselected categories P1-P4. Linkage to the dynamics and humidity obtained from the instant in-situ (Eulerian) observations does not allow for judgment about the history of the cloud environment.*

We tuned down the description on evolution and only mentioned the coexistence between liquid and ice in this revised sentence in Section 4: "Overall, the method proposed in this work provides a unique perspective to assess mixed phase cloud properties in both macrophysical and microphysical perspectives, especially for phases when supercooled liquid droplets and ice particles coexist."

*2. I have a hard time understanding the term "transition stage" throughout the manuscript. I brought this question up in my previous round of comments, however, I did not receive a clear answer. Employing the term "transition stage" implies that some clouds can exist in a non-transition stage. Generally speaking, any cloud can be described as an unstable colloidal system in a transition stage between water in gaseous and condensed stages (liquid and/or ice). There are several types of instabilities relevant to cloudy environments related to condensation/evaporation (e.g., due to dynamic forcing, entrainment & mixing, WBF, radiation effects, Ostwald ripening), mechanical interaction between particles (e.g., coalescence, aggregation, riming, fragmentation), and sedimentation. Each of these types of instabilities is characterized by its own time scale. Specifically, in relation to this study, the use of the term "transition stage" assumes a discussion of time scales such as time of phase relaxation, glaciation time, and residence time of cloud particles, along with different types of forcing. However, none of these points have been discussed. Therefore, the use of the term "transition stage" is redundant and may be misleading to the reader.*

We did a global search and changed the term "transition phase" to just "phase" throughout the text to avoid confusion.

*3. Per the previous comment, the title of the paper, "The Transition from Supercooled Liquid Water to Ice Crystals in Mixed-phase Clouds based on Airborne In-situ Observations" is misleading. The paper does not discuss the transition from liquid to ice. In fact, the transition of the thermodynamic phase may go in the opposite direction, i.e., ice to mixed-phase. This was also mentioned in section 3.1 and indicated in Fig 1. This conflicts with the title of the paper, implying a one-directional transition, "liquid to ice".*

In our last revision, we changed the title to "Transition between Supercooled Liquid Water and Ice Crystals in Mixed-phase Clouds based on Airborne In-situ Observations".

Since this may still be a bit misleading, we further changed the title of the manuscript to: "Partition between  Supercooled Liquid  Droplets and  Ice Crystals in Mixed-phase Clouds based on Airborne In-situ Observations".

*4. Lines 505-507: "Nevertheless, this method helps to provide a statistical categorization of different transition phases of mixed-phase clouds solely based on Eulerian-view sampling of aircraft data, which enables more detailed examination from a statistical, quasi-Lagrangian view that was not available previously." There was no "quasi-Lagrangian" consideration of mixed-phase in the text, and this statement at the end of the paper is unexpected and confusing. I also have a hard time understanding how quasi-random sampling of clouds (e.g. Eulerian) can be linked to a quasi-Lagrangian consideration. What are the time and spatial scales of the quasi-Lagrangian consideration referred to?*

We revised this sentence to stay with the statistical analysis perspective of this approach: "Nevertheless, this method helps to provide a statistical categorization of different phases of mixed-phase clouds solely based on Eulerian-view sampling of aircraft data. Future studies may derive such statistical distributions of phases based on 2-D remote sensing observations and 3-D model simulations. Examining individual phases of mixed-phase clouds may also provide more direct comparisons between observations and simulations."

*5. Lines 267-270: "Comparing RHi values in regions with and without ice, phase 2 shows higher RHi for regions with ice, while phase 3 shows higher RHi in regions without ice. This feature can be explained by the fact that higher RHi is required in order to initiate ice nucleation in phase 2, while ice crystals that continue to grow in phase 3 will further reduce RHi magnitude by vapor deposition." This is an unjustified statement. I believe the authors meant the dependence of INP nucleation on supersaturation. However, supersaturation in phase 2 (and any other type of cloud) is limited by saturation over liquid. This fact mitigates or eliminates the dependence of INP nucleation vs. vertical velocity in liquid and mixed-phase clouds. On the other hand, the differences between RHliq in P2 and P3 are within 1-4%. This is smaller than the accuracy of RH measurements (i.e., 6% - 7%). It applies limitations on relating these differences to physical processes, and it can be explained just by the error in RH measurements.*

We agree that the previous comment was an overstatement and removed it. In addition, after carefully thinking about the layout of the original Figure 5, we decided to remove this figure because it has many bins with a low number of samples, yet it gives the misleading look as if all the bins share similar significance. Since the relative humidity and vertical velocity are also shown in the new Figure 4, these two figures also become too repetitive. After removing Figure 5, the original text in Section 3.2 has been re-written, with descriptions focusing on new Figure 4 and the new supplemental Figure S4.

*6. As can be seen from Figure 3a, the sampling statistics of measurements are distributed quite unevenly across the temperature range and cloud types P1-P4. The lengths of different cloud types in different temperature subranges vary from approximately 850km down to 1km or less. The points with low sampling statistics (e.g., less than 100 of 1Hz samples ~17km) have low statistical significance. This should be clearly discussed in the text.*

We thank the reviewer for the helpful comment. We added this discussion in Section 3.1: "The lengths of different phases vary from ~0.2 – 180 km in various temperature ranges, with low sampling statistics (i.e., less than 100 seconds) of continuous in-cloud segments longer than 3.5 km, which indicates a patchy horizontal structure with clear-sky gaps inside the clouds."

*7. Figs. 4 and 5 include clouds P2 and P3 with subdivisions in clouds "with ice" and "without ice". I have a hard time understanding what it means. Does it mean that P2 "without ice" are just liquid clouds with mixed-phase cloud regions (MCR) excluded from the data set? Whereas clouds P2 "with ice" are just P2*

*clouds? Does it mean that clouds P3 "without ice" are clouds with excluded MCR and ICR? Or just with excluded ICR? Does it mean that P3 "with ice" is just P3 clouds or something else? With this ambiguity in the interpretation of the meaning of P2 & P3, "with ice" and "without ice" I found it difficult to follow the subsequent discussion.*

We added more clarifications in Section 3.2 when describing Figure 4: "For phases 2 and 3, LCR represents seconds without ice particles, while MCR and ICR represent seconds with ice particles. These two conditions (i.e., without or without ice) are separately examined in Figure 4 e-h and m-p." For example, in the diagram in Figure S1, a TCR labelled as phase 3 has 3 seconds of LCR and 4 seconds of ICR, therefore these 3 seconds of LCR represent "without ice", while the 4 seconds of ICR represent "with ice".

*8. Fig.3e shows that humidity in pure liquid clouds (P1, indicated by a red dot) is saturated over ice. This contradicts previous studies of humidity in liquid clouds (Korolev and Mazin, JAS, 2003; Korolev and Isaac, JAS, 2006, D'Alessandro et al., J.Clim., 2021). Something is fundamentally wrong here.*

We thank the reviewer for this fundamental suggestion. We believe the reviewer was referring to Figure 5e instead of Figure 3e. We have conducted more examinations about the relative humidity measurements and applied a new data quality control procedure, which is described in Section 2.1: "For $RH_{liq}$ lower than 100%, an adjustment to 100% is applied if two criteria are satisfied for a 1-Hz sample: 1) it contains supercooled liquid water and 2) either CDP or the King probe measures LWC greater than 0.001 g m$^{-3}$."

With this new quality control procedure, the **new Figure 4** is copied below. It shows much improved distributions of $RH_{liq}$ for segments containing supercooled liquid water, such as phase 1 (Figure 4 a), phase 2 (Figure 4 b), and phase 3 without ice (Figure 4 h). The $RH_{liq}$ distributions of these conditions now show high frequencies around liquid saturation at various temperature ranges (new Figure S4 b). Section 3.2 is also re-written. In addition, as mentioned above, the old Figure 5 from the last submitted manuscript (the Nov 25, 2023 version) has been removed to reduce the redundancy with new Figure 4, as well as eliminating the misleading features from ice spatial ratio bins that have too few samples.

[Figure]

**Figure 4.** Distributions of (a-h) $RH_i$ and (i-p) $\sigma_w$ in various phases as a function of temperature. Dashed lines in (a) – (h) indicate liquid saturation.

*9. The diagram in Fig.S6a shows that the PDFs of RHi in liquid and mixed-phase clouds from this study are centered at saturation over ice, i.e., RHi=100%. This result is overly concerning. It raises many questions about the accuracy and data quality of RH measurements and results presented in the paper.*

We appreciate the concern from the reviewer. Upon the additional quality control to the RH data as mentioned in the above comments, we have also updated that previous supplemental figure, which is now the new supplemental **Figure S4** (copied below). Liquid and mixed phase clouds now show the peak frequency of $RH_{liq}$ at liquid saturation for phases 1 – 3 (Figure S4b).

[Figure]

**Figure S4**. Probability density functions (PDFs) of (a) relative humidity with respect to ice, (b) relative humidity with respect to liquid, (c) vertical velocity (w), and (d) standard deviations of w ($\sigma_w$) separated by different phases. Phases 2 and 3 are further separated into seconds with or without ice in this analysis. Both phases 2 and 3 show higher frequencies of updrafts and $\sigma_w$ compared with phases 1 and 4

*10. Could you please double-check that the points with sigma_w=1+ m/s for P2 clouds in Fig.3h are not related to the malfunctioning of the LASEREF? This point looks suspicious and very different from the rest of the points. This is an overly high value, which is relevant for strong convection. If the data quality of these points is justified, could you check the type of clouds?*

We thank the reviewer for the helpful comment. We believe the reviewer is referring to the old Fig. 5h, which shows a sudden spike of $\sigma_w$ value in a specific ISR bin. We took several steps to address this comment.

First, we examined this specific spike with a high $\sigma_w$ value above 1 m/s. We located this high vertical velocity fluctuation in RF03 from UTC 02:34:39 to 02:34:45, which lasted about 6 seconds. Upon inspection, these few seconds of measurements look good and legit, and they occurred when the plane was flying horizontally. Because of the way that the old Figure 5 was plotted at various bins of ISR, this bin of ISR only has a few seconds of data containing these few seconds of high vertical velocity values, which leads to the high spike of $\sigma_w$ value. This is also another example that made us realize that the old Figure 5 is not a good representation of the entire dataset, since some bins may only represent a few seconds of data. That old Figure 5 is removed and replaced by the **new Figure 4**. Second, we examined the data quality of vertical velocity, especially during the ascent or descent legs. We applied a new quality control procedure for $\sigma_w$ value, which will be described in detail in the following comment.

*11. Lines 273-275: "For distributions of w in Figure 5 c, phase 1 has slightly higher w than other phases. Phases 2 – 4 show slightly negative average w values, suggesting weak downdrafts as the average condition in these phases." This is a concerning statement. Subsiding clouds at the rate of 10cm/s to 40cm/s (Fig.5c) will dissipate within 5 to 20 minutes with initial LWC=0.1g/m3 at T=-10C. This estimated time scale of cloud dissipation is shorter than the sampling time of the cloud during the flight observations. After my previous comment about vertical measurements, the authors found a negative bias of the vertical velocity measurement in the clear sky at 0.125m/s. Along this way, could you please check the drift of w in the clear sky? Note that measurements of vertical wind during ascending, descending, and any other type of aircraft maneuvering may result in significant biases of w. It is also worth mentioning that the LASEREF is an internal system, and the vertical wind velocity is calculated from the aircraft acceleration, i.e., the aircraft body is used as a sensor. The accuracy of such measurements is relatively low.*

That old statement about downdraft was referring to old Figure 5, which has been removed. We further investigated the question from the reviewer, which is, would the vertical motion of the aircraft during ascent or descent cause any drifting in the vertical velocity measurements. We first examined the time series during ascent and descent, compared with the horizontal legs, and we found no indication of drifting of vertical velocity during rapid ascent or descent. Next, we plotted the distributions of vertical winds against the maximum pressure differences (dP) seen in a duration of 10-second, 20-second, 30-second, or 40-second periods, and found vertical velocity distribution is centered at 0 for all these dP ranges (shown in **Figure R1** below). But to be on the cautious side, we removed $\sigma_w$ values when high dP values (> 10 hPa in 40 seconds) were observed.

We added this comment in Section 2.1: "To minimize the impacts of ascent and descent and the possible associated biases of vertical velocity measurements, we restrict the analysis of vertical velocity fluctuations (i.e., standard deviations of vertical velocity, calculated for every 40 seconds) to segments where the maximum pressure difference (dP) within 40 seconds is less than 10 hPa."

[Figure]

**Figure R1.** Scatter plots of the relationships between vertical velocity (w, m/s) and the maximum changes of pressure (dP) calculated within a duration of 10, 20, 30, and 40 seconds, color coded by $\sigma_w$ values.

*12. Lines 275-276: "For the $\sigma_w$ distribution (Figure 5 d), regions with ice in phase 2 have the highest fluctuations of vertical velocity, indicating that stronger in-cloud turbulence induces high RHi (as shown in Figure 5 a), which further initiates ice nucleation in phase 2." The last part of this statement relates the rate of INP nucleation with the vertical velocity, which induces higher RHi. This is an unjustified statement. Note that humidity in liquid clouds is always limited by quasi-steady supersaturation (i.e., Korolev and Mazin, 2003). In other words, in liquid clouds RHliq ~= 100%+ for a wide range of w. This is suggestive that the rate of the INP nucleation does not depend on w.*

We revised Section 3.2 and removed Figure 5. That former comment which was unjustified has also been removed.

*13. Lines 278-279: "Such result is consistent with the finding of Buhl et al. (2019) which showed a positive correlation between IWC mass flux and vertical velocity fluctuation, but this study further illustrates that in-cloud turbulence is particularly important for transition phase 2 when ice crystals first start to appear inside MCR, surrounded by supercooled liquid water." There are several observational studies that show a correlation between ice and vertical velocity. However, the statement "when ice crystals first start to appear inside MCR, surrounded by supercooled liquid water" is an overstatement. This study did not present evidence to support it.*

We revised Section 3.2 and that former statement was removed.

*14. Fig. 5f shows that for P2 and P3 clouds, in most cases RHliq(no ice) <RHliq(with ice). Assuming that "no ice" category means "liquid" the results presented in Fig.5f contradict fundamentals of mixed-phase clouds, i.e., for the same environmental conditions, humidity in pure liquid clouds is expected to be higher compared to that in mixed-phase clouds. I am not sure what the cause of the obtained inequality is. However, in view of the importance of this result, an explanation of this phenomenon is required.*

We addressed the concern over RH measurements in our comments above. The new **Figure 4** shows that $RH_{liq}$ peaks at liquid saturation for phases 1 and 2, as well as for 1-Hz samples without ice in phase 3.

*15. Lines 324-325: "... because ice crystal growth may occur via various processes in phase 3, such as ... vapor depositional growth under ice supersaturation..." This is not true. For RHi=100%, $dM_{ice}/dt=0$.*

We revised that sentence: "…probably because ice crystal growth may occur via various processes in phase 3, such as WBF process, glaciation, and/or riming."

*16. In my previous comment, I brought up a concern regarding using the ice concentration fraction $\lambda_{ice}=N_{ice}/(N_{liq}+N_{ice})$ for the analysis on mixed-phase, where $N_{liq}$ and and $N_{ice}$ are the concentrations of droplets and ice particles, respectively. For most mixed phase cloud $N_{liq}>>N_{ice}$ by a few orders of magnitude. Therefore, it is expected that in liquid clouds and mixed-phase clouds $\lambda_{ice}\approx0$, whereas in ice clouds $\lambda_{ice}\approx1$. The diagrams in Figs.7acd are consistent with this prediction, i.e. the cloud particle concentration in mixed-phase clouds is dominated by liquid droplet and therefore, $\lambda_{ice}\approx0$. However, diagrams in Figs.7bf caused questions. These diagrams show $\lambda_{ice}$ vs ice spatial ratio in P3 liquid and ice cloud regions (LCR & ICR). In Fig.7b ice concentration fraction varies in the range $0 < \lambda_{ice}< 0.4$, and in Fig.7f $0.3 <\lambda_{ice} < 0.9$. This is confusing, since in LCR the ice concentration fraction is expected to be $\lambda_{ice}\approx0$, whereas in ICR $\lambda_{ice}\approx1$. For the sake of argument, consider a case with $\lambda_{ice}\approx0.4$. Then, for the case of LCR (Fig.7b), the droplet concentration typical for the SOCRATES clouds $N_{liq} = 100$ cm$^{-3}$ the concentration of ice particles will be $N_{ice} = \lambda_{ice} N_{liq} /(1- \lambda_{ice}) \approx 67$ cm$^{-3}$. To the best of my knowledge, such high concentrations of ice have never been reported in scientific literature. On the other hand, the presence of ice in P3 LCR raises questions about the accuracy of the identification of liquid cloud segments in P3. For the case of ICR (Fig.7f) assume $N_{ice} = 100L^{-1}$ (high end of ice concentration). Then $N_{ice} = N_{liq} (1- \lambda_{ice}) / \lambda_{ice} = 0.15$ cm$^{-3}$. This is an overly low concentration of liquid droplets. Such clouds are volatile, and they may exist at cloud interfaces, and their lifetime scale is expected to be short. Another question is related to measurements of such clouds. If the measurements were performed in ice clouds, then both CDP and 2DS can be contaminated by shattering artifacts, which can be confused with liquid drops. Please note, that both antishattering tips and antishattering algorithms are not capable of 100% filtering out all shattering artifacts (Korolev et al. JTECH, 2013).*

We appreciate the reviewer for pointing out this issue. We took several weeks to investigate this issue. In a quick summary, we have a few key findings:

1.  Almost all the high ice particle number fraction (IPNF) ≥ 0.4 occurred when the UWILD 2DS data were the only contributor to the particle measurements (i.e., a total of 5273 seconds), while the CDP probe reported 0 or NaN. Only 4 seconds are the exception, when IPNF ≥ 0.4 and the CDP probe reported non-zero values.

2.  Because the main issue happens with UWILD 2DS data, we applied more quality controls to that dataset. Please note that we initially used the published data from the UWILD paper (Atlas et al., 2021) and the authors acknowledged the fact that at lower temperatures and smaller dimensions there are higher chances of misidentification by the machine learning model.

    We examined cloud images from a new cloud probe, the PHIPS probe, when it has concurrent measurements as the 2DS probe. Please note that the PHIPS probe does not report cloud images for every second when 2DS data have values. Also, the PHIPS probe may have an underestimation of the particle number concentrations compared with 2DS and CDP probes, so we cannot use the PHIPS cloud probe images as a quantitative measure for Nice or Nliq per second. But we use it here to verify concurrent seconds when UWILD 2DS data reported values.

    We found that some of the UWILD 2DS particle phase identifications have misidentified real ice particles (usually small ice fragments) as liquid droplets. This type of misidentification often happens at temperatures below -20°C, when there are numerous small ice fragments with a small dimension (e.g., < 20 micron) and a shape very close to a sphere.

    On the other hand, we also found real valid measurements when IPNF is relatively high, when a few supercooled liquid droplets are surrounded by more ice particles. These samples are often in ICR at higher temperatures. Below in **Figure R2** we show a few examples of PHIPS images when we found IPNF values between 0.2 and 1 and they look realistic.

3.  With these new examination results using the PHIPS probe, we applied the following quality control procedures to the UWILD 2DS data, added to Section 3.3: "Note that additional quality control procedures are applied to the ice particle number fraction (IPNF) data, because the machine-learning based particle identifications of 2DS data may misidentify small ice fragments as supercooled liquid droplets, especially at lower temperatures. To minimize such misidentifications, the following two quality control procedures are applied, which are developed after inspecting the Particle Habit Imaging and Polar Scattering (PHIPS) airborne cloud probe: (1) for 1-Hz samples of ICR in phase 3 and 4, when temperatures are below -20°C and 0 < IPNF < 1, IPNF is reset to 1 to be pure ice. In addition, for 1-Hz samples of ICR in phase 3, when temperatures are between -20 and -10°C and 0.4 < IPNF < 1, these IPNF values are reset to 1."

4.  We conducted a sensitivity test in **Figure R3** below, by completely removing the high INPF between 0.4 and 1 that may be complicated by the difficulties of particle identification of small hydrometeors (i.e., small ice versus liquid droplets). The main findings of positive correlations between the ice particle number fraction and mixed or ice spatial ratio from Figure 6 remain unchanged. In addition, the main finding that phase 3 shows higher slope values than phase 2 is also unchanged. We mentioned this sensitivity test in the text in Section 3.3: "Note even after quality control is applied to IPNF, a small amount of high IPNF values is still seen (e.g., 0.4 ≤ IPNF < 1) in Figure 6 b and f. A sensitivity test is conducted by removing all 0.4 ≤ IPNF < 1 in Figure 6 and the results show consistent conclusions, that is, all phases show positive correlations between IPNF and the spatial expansion of ice-containing regions. In addition, phase 3 still shows

higher slopes of linear regressions compared with phase 2, indicating faster increases of IPNF in phase 3 when pure ice segments start to appear."

5. With these new quality control procedures, we revised the **new Figure 6** (old Figure 7) and the **new Figure 8** (old Figure 9). The main changes can be seen in Figure 8 c, with much higher frequency of ice phase at lower temperatures compared with the old version of Figure 9 c. In addition, we added the new types of analysis showing the frequency distributions of ISR, ice mass fraction (IMF) and IPNF in the top two rows of new Figure 8. For the combined phases 1 to 4 of all in-cloud conditions, the ISR, IMF and IPNF all peak at 0 and 1, with fewer samples in between.

Below we copied Figures R2 and R3, as well as Figures 6 and 8 mentioned above.

[Figure]

**Figure R2.** Sample images from the PHIPS probe and the matching time series for the cloud segments with $0.2 \leq \text{IPNF} < 1$ in RF2, RF5, and RF6.

[Figure]

**Figure 6.** Relationship between ice particle number fraction and mixed spatial ratio or ice spatial ratio, separated by two phases (phase 2 in column 1 and phase 3 in column 2), and by various cloud segments – (a, b) LCR, (c, d) MCR and (e, f) ICR. Average values for each ice spatial ratio bin are shown in black solid lines, with vertical bars representing standard deviations. Linear fit is shown in red dashed line. Average values of generating cells (time series obtained from Wang et al. (2020)) are in pink "X" markers. The slope value b, its associated standard deviation, and the ordinary R-squared value are shown in the legend.

[Figure]

**Figure R3.** A sensitivity test that is similar to Figure 6, but removing IPNF ≥ 0.4 for LCR and MCR, and 0.4 ≤ IPNF < 1 for ICR. Both phases 2 and 3 show positive slopes for linear regressions, and phase 3 shows higher slope values compared with phase 2, consistent with Figure 6.

[Figure]

**Figure 8.** Frequency distributions of (a) ice spatial ratios calculated for individual consecutive TCR, (b) ice mass fraction per second, and (c) ice particle number fraction per second for four phases. (d-f) Similar to (a-c), but for all the phases combined representing the entire in-cloud conditions. (g-i) cloud phase frequency distributions defined based on the respective parameter in each column.

*17. The measurements of aerosol particles presented in section 3.5 were conducted by UHSAS inside clouds. I have a serious concern about this approach and the data quality. There is no need to say that aerosol inside clouds is modified by droplet activation, droplet collision- coalescence, scavenging by cloud particles, and precipitating out of the cloud. An example of aerosol processing can be seen in Fig.13b at [https://doi.org/10.1175/2011BAMS3180.1](https://doi.org/10.1175/2011BAMS3180.1). It is also not clear how a 50-second moving average would help eliminate this issue. Such averaging is expected to mix LCR, MCR, ICR, and out-of-cloud regions.*

To address the issue of potential contamination of aerosol observations inside clouds, we revised the **new Figure 9** and **new Figure S6** (copied below) by limiting the averaging of aerosol concentrations to clear-sky conditions only. The positive correlations between the average aerosol number concentrations and mixed or ice spatial ratio are still seen in phase 2 and phase 3. In addition, phase 2 shows higher slope values compared with phase 3, which are similar to our conclusions before.

We also edited the discussion in Section 3.5: "Due to the possible complication of in-cloud measurements of aerosol number concentrations, we applied a moving average to calculate logarithmic scales of clear-sky aerosol concentrations at every 50 seconds in Figure 9. Furthermore, the average aerosol concentration is only analyzed if more than half of the entire 50 seconds satisfy the criteria of in-cloud conditions. A coarser spatial averaging using the moving average of clear-sky conditions of every 100 seconds is also shown in supplementary Figure S6."

All the main findings remain consistent with our previous manuscript, including positive correlations between MSR or ISR with aerosol number concentrations, higher slope values for phase 2 than phase 3, and higher slope values for $N_{>500}$ than $N_{>100}$.

[Figure]

**Figure 9.** Similar to Figure 6, but showing logarithmic scale (a-h) $N_{>500}$ and (i-p) $N_{>100}$ in relation to mixed spatial ratio or ice spatial ratio, separated by the phases and cloud regions. The first, second, and third rows represent LCR, MCR, and ICR, respectively. The last row represents all cloud regions in a specific phase. The aerosol number concentrations represent the moving average values of every 50 seconds for the clear-sky conditions only.

[Figure]

**Figure S6**. Similar to Figure 9 in the main manuscript but using 100-second moving averages of logarithmic scales of $N_{>500}$ and $N_{>100}$ for the clear-sky conditions only. The results using the coarser scale of aerosol number concentrations are very similar to those shown in Figure 9.

*18. Please, replace the reference Korolev et al. JTECH, 1998 by Korolev, A.V., 1998: "About Definition of Liquid, Mixed and Ice Clouds." FAA Workshop on Mixed-Phase and Glaciated Icing Conditions. December 2-3, Atlantic City, NJ, 325-326. This is a result of the error in referencing papers in Korolev et al. AMS Monogr. 2017, which propagated to the present study.*

We have replaced the reference Korolev et al. JTEch, 1998 by the following:

Korolev, A.V.: About Definition of Liquid, Mixed and Ice Clouds. FAA Workshop on Mixed-Phase and Glaciated Icing Conditions. December 2-3, Atlantic City, NJ, 325-326, 1998.

---

## Author Response (AR3)

***Responses to the Reviewers***

Format: The reviewer's comments are quoted in italic

Section number in the response refers to the revised manuscript with tracked changes

Quotation in red color stands for revised/added text in the revised manuscript

***Overall comment:***

We thank the reviewer for the detailed comments. We have addressed individual comments from the reviewer in our response. Some main changes include:

1) Adding a new Figure 2 (which is originally Figure S1 showing the schematic diagram);
2) Adding a supplemental Figure S4, showing a sensitivity test on the impact of length scales;
3) Revising new Figure 5 by combining the original Figure S4 into the original Figure 4;
4) Revising the figure of particle size distributions which becomes the new Figure 6 and new Figure S5 in the revised manuscript;
5) Adding a new Figure S6, showing the linear regressions applied to individual seconds of samples instead of average values inside each bin;
6) Adding clarifications in our revised writing to address most of the remaining minor comments.

***Response to comments from the Reviewer***

*Review: Partition between Supercooled Liquid Droplets and Ice Crystals in Mixed-phase Clouds based on Airborne In-situ Observations*

*This study analyzes the in-situ observation of mixed-phase clouds collected during the SOCRATES field campaigns over the ocean. Each cloud segment is categorized into four phases: 1) liquid, 2) mixed phase/liquid, 3) mixed-phase/liquid/ice, and 4) ice. The dependency of microphysical cloud properties, dynamical properties, and aerosol properties on each of these phases is examined. The paper introduces mixed and ice spatial ratios to describe the evolution of the phases.*

*This paper presents an intriguing approach for analyzing in-situ data of mixed-phase clouds. However, there are concerns regarding the equal treatment of cloud segments between 0.2 km and 180 km (see major comment). The quality of the presentation could be enhanced by focusing on fewer figures and discussing these figures more comprehensively (see minor comments).*

*Recommendation: I suggest reconsidering the paper after making major revisions.*

*Major:*

*Cloud segments vary in length from 0.2 to 180 km. What is the likelihood that a short cloud segment is incorrectly classified as liquid (phase 1) due to the low measurement volume missing ice crystals? Is there a possibility that the two edges of a long cloud segment interacted with each other? If not, what justifies treating them as one quantity in the analysis? How do the results depend on the length of the segments? Would splitting long cloud segments into smaller pieces (e.g., 1000 m), where cloud particles interact with each other, be advantageous?*

We can see why the previous writing may have caused the confusion. To clarify, our analysis is indeed using every second of data within a consecutive total cloud region (TCR). That is, if a TCR segment contains 10 seconds, then there will be 10 samples at 1-Hz resolution, while a segment containing 100 seconds will produce 100 of 1-Hz samples. We revised the Y axis label for new Figure 4 (below) to

distinguish the number of 1-second samples (panel a) from the number of TCR segments (panel c). That is, Figure 4 a shows the number of 1-second samples to be used in all the following analysis (i.e., Figures 5 – 10), while Figure 4 c shows the number of TCR segments associated with various length scales. The count of TCR segments and the distributions TCR lengths are only used in Figure 4 c, not in latter sections.

[Figure]

**Figure 4.** Distributions of 1-Hz samples in four phases at various temperatures in the top row. (a) Number of 1-second samples and (b) frequency of 1-second samples in each phase within various temperature bins. In (b), the frequency of 1-second samples in each phase is normalized by the total number of 1-second samples of all phases in each 5-degree temperature bin. Distributions of various lengths of TCR segments are analysed in the bottom row. (c) Number of TCR segments and (d) frequency of cloud segments in each phase associated with various lengths in log10-scale. In (d), frequency is calculated as the number of segments of a specific phase divided by the total number of segments in each $10^{0.25}$ bin.

To address the comments of "*Is there a possibility that the two edges of a long cloud segment interacted with each other? If not, what justifies treating them as one quantity in the analysis?*" and "*Would splitting long cloud segments into smaller pieces (e.g., 1000 m), where cloud particles interact with each other, be advantageous*", the in-cloud segments (i.e., TCRs) are already separated into 1-second samples when being analyzed for all the analyses shown in revised Figures 5 – 10. To improve the clarity of the text and figure, we revised supplemental Figure S1 and moved it to the main text as the **new Figure 2** (copied below). New descriptions were added in Section 3.1: "An illustration of the identification of TCR is shown in Figure 2. In that example, 1 second of LCR, 2 seconds of MCR, and 4 seconds of ICR are adjacent to each other. Then the 1 LCR sample, 2 MCR samples, and 4 ICR samples all belong to the

same TCR, which produces a total of 7 seconds of samples. All the 1-Hz samples within the TCR will be used in the analysis in Sections 3.3 – 3.8 (i.e., Figure 4 a and b, Figures 5 – 10)."

In addition, we tried to highlight the differences between the definition of LCR, MCR and ICR and the definition of TCR in Section 3.1: "In the first step, each second of observations are categorized into four conditions, including a second of clear-sky condition, liquid cloud region (LCR), ice cloud region (ICR), or mixed-phase cloud region (MCR). LCR is defined as a one-second sample where only supercooled liquid droplets were observed, while ICR is defined as a one-second sample with only ice crystals. MCR is a one-second sample with occurrence of both ice and liquid. In the second step, a total cloud region (TCR) that can potentially contain multiple seconds with a combination of LCR, ICR and MCR is identified, which basically is a consecutive in-cloud segment surrounded by clear-sky conditions. In other words, LCR, ICR and MCR are defined at the scale of each second, while TCR is defined at the scale of a consecutive in-cloud segment which can contain more than one second."

[Figure]

**Figure 2.** A schematic diagram that illustrates the identification of a total cloud region (TCR) sample, with 1 second of LCR (red), 2 seconds of MCR (purple), and 4 seconds of ICR (blue) embedded inside this TCR. All 7 seconds of samples inside this TCR are used in the following analysis of cloud properties.

To address the comment of "*What is the likelihood that a short cloud segment is incorrectly classified as liquid (phase 1) due to the low measurement volume missing ice crystals?*", we plotted the scatterplots of ice water content (IWC) versus liquid water content (LWC), as well as ice crystal number concentration (Nice) versus supercooled liquid droplet number concentration (Nliq), for the conditions when both ice and liquid are observed (shown below as **Figure R1**). We added this discussion in Section 3.1: "To investigate the possibility of misclassifying MCR as LCR due to the relatively lower number concentrations of ice particles compared with supercooled liquid droplets in a one-second sampling volume, distributions of mass and number concentrations of ice crystals are examined against those of supercooled liquid droplets (not shown). When liquid and ice coexist, the majority of the 1-second samples have both IWC > 0.01 g m$^{-3}$ and LWC > 0.01 g m$^{-3}$. In addition, the mass concentrations and number concentrations of ice and liquid are positively correlated with each other. This indicates that when ice and liquid coexist, most likely both types of hydrometeors have significant mass and number concentrations. Thus, it is less likely that the smaller sampling volume for ice crystals would lead to a misclassification of MCR as LCR. It is possible though, that some pure ICR pockets with very low number concentrations of ice crystals may be missing."

[Figure]

**Figure R1.** Distributions of (a) IWC with respect to IWC, and (b) Nice with respect to Nliq, all in log10-scale, plotted for the samples with coexistence of ice crystals and supercooled liquid droplets. The scatterplot dots are color coded by temperature.

To address the comment of "*How do the results depend on the length of the segments?*", we plotted the distributions of four phases similar to the revised Figure 4, but restricted to different length scales of TCR (**new supplemental Figure S3**, shown below). We added these comments in the text in Section 3.3: "The impact of length scales of TCR on the phase distributions is examined in supplemental Figure S3. TCR samples are separated into four scales – 0.1 – km, 1 – 10 km, 10 – 100 km, and > 100 km. The dependence on temperature for the distributions of four phases is consistently seen for various scales, e.g., phase 1 has more samples at higher temperatures, while phase 4 has more samples at lower temperatures. Comparing the shorter (Figure S3 a and b) and longer (c and d) TCR samples, the shorter ones have more samples in phase 1 (i.e., pure liquid phase), while the longer ones have more phases 2 and 3. This result indicates that the coexistence of ice and liquid occurs more frequently in clouds with larger spatial extent, such as stratocumulus and stratus clouds."

[Figure]

**Figure S3.** Similar to Figure 4 a, distributions of 1-Hz samples in four phases at various temperatures, separated by the length scales of TCR samples. E ach second within the TCR is counted as a sample.

*Minor:*

*Abstract: Specify the dataset used and the types of clouds investigated.*

We added more information to the abstract: "Using this method, we examine the relationship between the macrophysical and microphysical properties of Southern Ocean mixed-phase clouds at -40 to 0°C (e.g., stratiform and cumuliform clouds) based on the in-situ aircraft-based observations during the US National Science Foundation Southern Ocean Clouds, Radiation, Aerosol Transport Experimental Study (SOCRATES) flight campaign."

*Line 180: Figure S1 is crucial for understanding the approach and should be moved to the main manuscript.*

We revised this figure, which becomes the new Figure 2 as copied in our response above.

*Line 184: The introduction of M1, M2, M3 is confusing and unnecessary as it only appears in Table 1.*

We removed the terms of "M1, M2 and M3" and revised the text: "Within each TCR, the spatial ratios of LCR, MCR, and ICR relative to TCR are calculated. The definitions of each phase are based on these spatial ratios as described in Table 1. The number of one-second samples and the number of cloud segments for four phases are summarized."

The revised Table 1 is copied below.

**Table 1.** Definitions of four phases of mixed-phase clouds based on ratios of lengths of LCR, MCR, and ICR over the length of TCR within a consecutive cloud segment, i.e., $\frac{L_{LCR}}{L_{TCR}}$, $\frac{L_{MCR}}{L_{TCR}}$, and $\frac{L_{ICR}}{L_{TCR}}$, respectively.

| Phase | Description | Number of 1-second samples | Number of TCR segments | Spatial Ratio of LCR | Spatial Ratio of MCR | Spatial Ratio of ICR |
|---|---|---|---|---|---|---|
| 1 | Only LCR | 8243 | 1163 | $\frac{L_{LCR}}{L_{TCR}} = 1$ | $\frac{L_{MCR}}{L_{TCR}} = 0$ | $\frac{L_{ICR}}{L_{TCR}} = 0$ |
| 2 | MCR appears | 12557 (LCR: 11096, MCR: 1461) | 142 | $0 \leq \frac{L_{LCR}}{L_{TCR}} < 1$ | $0 < \frac{L_{MCR}}{L_{TCR}} \leq 1$ | $\frac{L_{ICR}}{L_{TCR}} = 0$ |
| 3 | Pure ICR must appear | 11988 (LCR: 3478, MCR: 2973, ICR: 5537) | 249 | $0 \leq \frac{L_{LCR}}{L_{TCR}} < 1$ | $0 \leq \frac{L_{MCR}}{L_{TCR}} < 1$ | $0 < \frac{L_{ICR}}{L_{TCR}} < 1$ |
| 4 | Only ICR | 8646 | 1193 | $\frac{L_{LCR}}{L_{TCR}} = 0$ | $\frac{L_{MCR}}{L_{TCR}} = 0$ | $\frac{L_{ICR}}{L_{TCR}} = 1$ |

*Line 215: It should be a second-by-second analysis. Could an analysis of larger intervals (e.g., 10 seconds) provide a better understanding of the cloud phases?*

We can see why the previous writing led to a misunderstanding. We indeed are using every second within a TCR as individual samples. We revised that sentence to clarify the difference between this work and previous studies: "Nevertheless, this method provides a statistical separation of the cloud phases and allows a more focused analysis of the coexistence of supercooled liquid water and ice crystals that cannot be achieved if a one-second sample is analyzed without the context of its surrounding conditions, for instance, if a one-second LCR is part of a pure liquid cloud segment, or is surround by MCR or ICR." Regarding the comment about intervals, we found that the longer TCR segments are generally associated with larger gaps between cloud segments, so the analysis restricting to larger gaps lead to a similar result as restricting to larger cloud segments, as discussed in Figure S3.

*Line 225: "The lengths of cloud segments vary..."*

We revised it to "The lengths of TCR segments vary…"

*Line 225: After the sampling statistic, I expected the number of samples, not a time.*

We thank the reviewer for catching the typo. That sentence has been revised: "The lengths of TCR segments vary from ~0.2 – 180 km in various temperature ranges, with low sampling statistics (i.e., less than 100 ) of continuous in-cloud segments longer than 60 km…"

*Line 236 – 238: A mixed-phase cloud (MPC) consists of supercooled droplets and ice crystals.*

We revised this sentence: "For macrophysical properties of mixed-phase clouds, we focus on investigating the lengths of cloud segments and the spatial fraction of a cloud segment containing ice, which is defined as mixed spatial ratio and ice spatial ratio."

*Which spatial fraction describes the macrophysical properties of MPCs? How do LCR, ICR, MCR, and TCR represent macrophysical properties, and why aren't they used in the analysis?*

The spatial fraction containing ice for phase 2 is defined as mixed spatial ratio. The spatial fraction containing ice for phase 3 is defined as ice spatial ratio. These two spatial ratios are analyzed in Figures 7 – 10 in the revised manuscript. The length of cloud segments is analyzed in Figure 4 panels c and d of the revised manuscript.

*Line 240 – 245: Is it correct that the mixed spatial ratio equals the spatial ratio of MCR (M3), but the ice spatial ratio differs from the spatial ratio of ICR? If yes, try to find clearer names and perhaps add the definition of mixed spatial ratio and ice spatial ratio to Table 1.*

Yes, the reviewer is correct that "*mixed spatial ratio equals the spatial ratio of MCR (M3), but the ice spatial ratio differs from the spatial ratio of ICR*", because phase 3 may contain both MCR and ICR. Because the original terms of M1, M2 and M3 caused confusion, we removed their name, and directly refer to them as length of LCR, MCR and ICR relative to the length of TCR (as described in Table 1 caption). In addition, the discussions of mixed spatial ratio and ice spatial ratios do not appear until we analyze them in Figure 7 of the revised manuscript, thus we moved the definitions of these two terms into Section 3.6, right before we introduced Figure 7.

*Line 270 – 272: What is the percentage of observations over 1.25 m/s in phase 2 and 3? Is the difference significant? I suggest moving Figure 4 i-p to the appendix and Figure S4 b-d to the main manuscript.*

We revised the new Figure 5, and moved the original Figure 4 i – p to be the new supplemental Figure S4.

The percentage of $\sigma_w$ values of one-second samples greater than 1.25 m/s in phases 2 and 3 are very small, which is 0.412% and 0.660% of the total samples of each phase, respectively. In addition, we added a new supplemental Table S2 to illustrate that phases 2 and 3 have higher frequencies of larger $\sigma_w$ values than phases 1 and 4.

**Table S2.** Number of one-second samples of $\sigma_w$ (i.e., standard deviation of vertical velocity) in four phases at various ranges. Percentages relative to the total number of $\sigma_w$ samples of each phase are shown in parentheses.

| Phase number | All $\sigma_w$ values | $\sigma_w \geq 0.5$ m/s | $\sigma_w \geq 1$ m/s | $\sigma_w \geq 1.25$ m/s |
|---|---|---|---|---|
| Phase 1 | 4549 | 621 (13.7%) | 15 (0.330%) | 0 |
| Phase 2 | 8730 | 1360 (15.6%) | 174 (1.99%) | 36 (0.41%) |
| Phase 3 | 8638 | 1491 (17.3%) | 251 (2.91%) | 57 (0.66%) |
| Phase 4 | 7814 | 176 (2.25%) | 0 | 0 |

We revised the text in Section 3.4 describing the new Figure 5, Figure S4 and Table S2: "Probability density functions (PDFs) of $RH_i$, $RH_{liq}$, vertical velocity, and $\sigma_w$ are further examined in Figure 5 i – l. The peak frequencies of $RH_{liq}$ are seen at liquid saturation for phases 1 – 3, consistent with the findings in Figure 5 a – d. The PDFs of vertical velocity show higher frequencies of updrafts for phases 2 and 3

compared with phases 1 and 4. In addition, PDFs of $\sigma_w$ show higher frequencies of large $\sigma_w$ values in phases 2 and 3 than phases 1 and 4. The number of 1-Hz $\sigma_w$ samples at various ranges (i.e., $\geq 0.5$ m/s, $\geq 1$ m/s, and $\geq 1.25$ m/s) and their percentages relative to the total samples in each phase are shown in supplemental Table S2. That analysis also shows higher percentages of larger $\sigma_w$ values in phases 2 and 3 compared with phases 1 and 4. Similarly, the distributions of $\sigma_w$ as a function of temperature in supplemental Figure S4 show more samples above 1 m/s across a wide range of temperatures from -36°C to 0°C in phases 2 and 3 than phases 1 and 4."

[Figure]

**Figure 5.** (a-h) Distributions of $RH_i$ as a function of temperature. The PDFs of (i) $RH_i$, (j) $RH_{liq}$, (k) vertical velocity (w) and (l) $\sigma_w$ of various phases. Dashed lines in (a) – (h) indicate liquid saturation.

*Line 295 – 296: Which phase are you referring to? I suggest plotting all temperatures of the size distribution of this phase in one plot to emphasize the differences. As differences in temperature are not further discussed, I suggest moving the size distribution of the temperature interval to the appendix and showing only the size distribution of all temperatures in the main manuscript.*

We revised this figure. The new Figure 6 now shows the entire temperature range between -40 and 0°C, while the separated temperature ranges are shown as the new supplemental Figure S5.

[Figure]

**Figure 6.** Particle size distribution of the four phases for mixed-phase clouds separated by probe types. The entire dataset at the temperature range of -40°C to 0°C is shown. Phase 4 only shows 2DS measurements because ice particles measured by CDP are excluded from the analysis.

[Figure]

**Figure S5.** Particle size distribution separated by four phases and various temperature ranges. Four temperature bins between -40°C to 0°C are shown in each panel. Phase 4 only shows 2DS measurements because ice particles measured by CDP are excluded from the analysis.

*Line 305 – 306: What is the fraction of observations with ice particle number fraction > 0.1 in Figure 6b? Why was the linear regression calculated on the mean of each ice spatial ratio bin (which weighted each bin equally despite very different numbers of observations in each bin) and not based on individual observations? Please add more information on the calculation of the linear regression.*

The original Figure 6 b is now Figure 7 b in the revised manuscript. We added this in the discussion in Section 3.6: "After the corrections, out of 2866 seconds of samples analyzed in Figure 7 b, 172 seconds (i.e., 6.00%) show IPNF > 0.1."

We clarified the reason of conducting linear regression for the mean microphysical properties, and also added a new supplemental Figure S6 to illustrate the differences if the linear regressions are applied directly to individual seconds of samples. The discussion is added to Section 3.6: "The linear regression analysis is applied to the average values of microphysical properties in each spatial ratio bin, in order to assign an equal weight to each bin of mixed or ice spatial ratio. When directly applying the linear regressions analysis to individual seconds of IPNF (as shown in supplemental Figure S6), similar slope values are seen compared with Figure 7, but the bins of mixed spatial ratio and ice spatial ratio have uneven distributions of samples."

[Figure]

**Figure S6**. Similar to Figure 7, but applying the linear regressions directly to individual seconds of samples.

*Line 315 – 320: I have difficulty following the argument. Are you referring to "ice crystals gradually dominating the total particle population" as high ice particle number fraction? How can "a particular TCR" be identified in the multiple subfigures? What is the spatial extent of the entire cloud segment?*

We revised this sentence to clarify our point: "This means that while ice crystals gradually dominate the total particle population (i.e., IPNF increases) in cloud segments, the spatial fraction containing ice particles (i.e., MCR+ICR) also approaches 1 from a macroscopic perspective."

*Line 323 – 324: How can "ICR appear" when phase 3 always has some ICR?*

We deleted that phrase in the revised sentence: "On the other hand, in phase 3, ice crystals start to become the dominant particles by number concentration  and supercooled liquid droplets become less dominant."

*Line 325: Please describe in more detail how you derived that "they experience similar rates of phase changes from liquid to ice" based on measurements of individual states of cloud microphysics?*

We can see that the original comment is a little far reaching. Thus we deleted that discussion about the rate of phase change.

*Line 336: How do you conclude that generating cells contain lower ice particle number fractions? What is the uncertainty of the generating cells measurements?*

We added the explanation in the text in Section 3.6: "Previously, Wang et al. (2020) used airborne remote sensing measurements from the SOCRATES campaign to identify generating cells of ice crystals. Based on the definition from American Meteorological Society (2013), generating cells are defined as cloud-top regions with high radar reflectivity, which often produce fall streaks of falling hydrometeors. Out of the 16 cases of generating cells detected by Wang et al. (2020), all 16 cases contain supercooled liquid droplets. The average LWC and Nliq inside generating cells were found to be greater than those outside the generating cells. In addition, larger ice particles and higher Nice were seen in the generating cells, associated with the updrafts inside the cells. These reported generating cells are also analyzed in Figure 7, with the average IPNF values shown in each mixed and ice spatial ratio bin. The generating cells associated with LCR and MCR contain lower IPNF (Figure 7 a – d). This is because when generating cells are associated with high concentrations of supercooled liquid droplets, Nice may be lower than Nliq, which leads to the lower IPNF. But when the generating cells are associated with ICR, significantly higher IPNF (close to 1) are seen for most ice spatial ratio bins (Figure 7 f). This result suggests that not all regions within the generating cells experience significant phase change from liquid to ice, unless the ice-containing regions become dominated by ice."

*Line 348-349: What do you mean by "... similar rate of increase between ice crystals embedded among supercooled liquid droplets..."?*

We can see that the original discussion is quite confusing. We deleted that comment and focused on comparing the slope values in various phases.

*Line 352: Should be Figure 7i.*

Thanks for catching the typo. Yes, it should be Figure 7 i.

*Line 357 – 358: What effect would the formation and growth of ice particles have, and could the depletion of the liquid phase also play a significant role?*

We can see that the original comment about "formation and growth of ice particles" is unnecessary and causes confusion, so we deleted that part of the sentence. The revised sentence is: "As ice crystals grow into pure ice segments (i.e., ICR), liquid phase starts to rapidly evolve into ice phase."

*Line 405: Why should SIP be stronger in phase 3 when phase 2 has more large droplets (according to Fig. 5)?*

Secondary ice production (SIP) is generally identified when Nice is much higher than number concentrations of ice nucleating particles. In other words, the very high Nice values are usually associated with SIP. Even though phase 2 has higher Nliq than phase 3, phase 3 also has higher Nice than phase 2 (as shown in the particle size distribution in revised Figure 6). Thus, it is more likely that phase 3 contains more SIP events than phase 2.

*Line 447-448: Please be precise if you are referring to MCR/ICR or phase 2/3.*

We can see why the original comment is confusing. We revised it to: "... the method presented in this work allows one to compare the cloud segments when ice crystals are surrounded by supercooled liquid water in MCR with those when pure ICR starts to appear."

*Line 468- 476: Move this paragraph to the definition of the phases.*

Thanks for the suggestion. We moved the paragraph to the end of Section 3.1, after defining the four phases.

*Line 480 – 484: Are you suggesting that once a small pocket of pure ice crystals appears in a cloud (phase 3), the rate of change from liquid to the ice phase accelerates for the whole cloud?*

We revised the comment indicating a causal relationship to a correlation: "This study illustrates that the rates of phase change are also correlated with the existence of pure ice segments (Figures 7 and 8), not only with the mixed spatial ratio or ice spatial ratio which reflects the spatial fraction of ice-containing regions. Future model parameterization is recommended to quantify the varying rates of phase change throughout a cloud's lifetime by considering two main factors – the type of phases (especially phase 2 versus phase 3) and the spatial fraction of ice-containing region."

---

## Author Response (AR4)

*Responses to the Reviewers*

Format: The reviewers' comments are quoted in italic

Section number in the response refers to the revised manuscript with tracked changes

Quotation in red color stands for revised/added text in the revised manuscript

**Overall comment:**

We thank the two reviewers for the detailed, helpful comments. We have addressed individual comments as shown below and revised the manuscript accordingly.

**Response to comments from Reviewer 5**

*The authors have made commendable revisions to the main text of the manuscript. However, these changes need to be reflected in the Discussion and Conclusion sections. Ensure that figures are referenced appropriately in both the Discussion and Conclusion sections to support the findings and arguments presented.*

We appreciate the helpful comments from Reviewer 5. We checked through the entire manuscript to make sure all descriptions directly follow the most up-to-date version of the figures and tables. We also made a few revisions in Section 5 Discussion and Conclusions as the reviewer recommended below.

*Revise argumentation:*

*Line 455: Elaborate on how this method can investigate the (temporal) evolution of cloud properties.*

We revised this sentence to avoid using the terminology of evolution in Section 5 Discussion and Conclusions: "This method allows an investigation on  cloud macrophysical and microphysical properties as well as the related aerosol indirect effects at different levels of partitioning between supercooled liquid water and ice particles, as the phase change occurs among vapor, liquid, and solid phase of water molecules."

*Lines 510-514: Remove or revise these lines, as the argumentation of growth rate has been removed from the main text.*

We revised this section and removed the terminology of growth rate in Section 5: "This study illustrates that the mass and number partitioning between liquid and ice hydrometeors in mixed-phase clouds are not only correlated with the mixed spatial ratio or ice spatial ratio which reflects the spatial fraction of ice-containing regions, but also are correlated with the existence of pure ice segments (Figures 7 and 8). Future model parameterization is recommended to quantify the varying rates of phase change throughout a cloud's lifetime by considering two main factors – the type of phases (especially phase 2 versus phase 3 depending on the existence of pure ice segments) and the spatial fraction of ice-containing region."

*Clarify specific terms:*

*Lines 462-463: Clarify what is meant by "all three microphysical properties". In line 350 four microphysical properties (IPNF, LWC, IWC, and ice mass fraction) are listed.*

We thank the reviewer for pointing this out. We revised this sentence in Section 5 to: "Comparing phases 2 and 3, the latter phase shows higher rates of changes in four microphysical properties with increasing ice spatial ratio, including faster increase of IPNF, faster increase of IWC, faster decrease of LWC, and faster increase of ice mass fraction (Figures 7 and 8)."

We also revised one sentence in the abstract to mention all 4 cloud microphysical properties: "The results show that the exchange between supercooled liquid water and ice crystals in a macrophysical perspective, represented by the increasing spatial ratio of regions containing ice crystals relative to the total in-cloud region (defined as ice spatial ratio), is positively correlated with the phase exchange in a microphysical perspective, represented by the increasing ice water content (IWC), decreasing liquid water content (LWC), increasing ice mass fraction, and increasing ice particle number fraction (IPNF)."

*Line 465: Specify that the higher rate of phase change is in respect to the spatial ratio, not time, to avoid confusion.*

We agree that it would be helpful to clarify this point. We revised this sentence in Section 5 to: "These results indicate that when ice crystals become more dominant and pure ice segments start to appear, both the mass and number partitions between liquid phase and ice phase experience a higher rate of phase change with respect to the spatial ratio of ice-containing regions (note that this rate of change is not with respect to time)."

**Response to comments from Reviewer 6**

*Manuscript Title: "Partition between Supercooled Liquid Droplets and Ice Crystals in Mixed-phase Clouds based on Airborne In-situ Observations"*

*Key Scientific Question: The manuscript addresses the question of how the macrophysical and microphysical properties of mixed-phase clouds, and the factors controlling their formation and evolution, impact their radiative forcing over the Southern Ocean. This research aims to understand the interaction between supercooled liquid water and ice crystals within these clouds and how aerosols influence these properties. The study introduces a novel method for categorizing mixed-phase clouds into four distinct phases based on spatial relationships among segments containing pure ice (ICR or phase4), pure liquid (LCR or phase1), or both, liquid dominated mixed-phase (MCR or phase 2) and ice-dominated mixed-phase MCR or phase3). Key findings include positive correlations between ice particle number fraction and ice water content (IWC) with mixed and ice spatial ratios in phases 2 and 3, with phase 3 showing faster changes. All methods identified a significant phase transition around -17.5°C. Larger aerosols were found to be more likely to act as ice nucleating particles (INPs), with phase 3 exhibiting weaker aerosol indirect effects due to secondary ice production. Higher updrafts and stronger in-cloud turbulence were observed in mixed-phase conditions, particularly in phase 3. These insights suggest that future climate models should account for varying phase change rates and spatial fractions of ice-containing regions. However, the method has limitations, including its idealized nature, reliance on 1-D aircraft data, and potential lack of comprehensive spatial representation. Future research should integrate 2-D and 3-D observations (remote-sensing) and simulations to validate and refine the phase categorization method, enhancing our understanding of mixed-phase cloud dynamics and their climate impacts.*

We appreciate the helpful comments from Reviewer 6. Below is our response to each of the comments.

*Minor Comments:*

*- Line 266: Typo → − 0.1 – km → 0.1 − 1 km*

We thank the reviewer for catching this typo. We revised it to "0.1 – 1 km".

*- Line 280: Typo → Wegner → Wegener*

We thank the reviewer for catching this typo. We revised it to "Wegener".

*- Line 485: "Because of this, aerosol indirect effects on various stages of clouds can also be examined separately." - Because of what? Consider rephrasing for clarity!*

We agree that this former sentence can cause confusion. We revised this sentence in Section 5 to: "Aerosol indirect effects on mixed-phase clouds during different levels of phase partitioning can also be examined separately."

*Conclusion*

*The manuscript presents significant advancements in the understanding of mixed-phase clouds and their classification. The novel method proposed is promising and provides valuable insights into cloud microphysical and macrophysical properties. Addressing the major remarks and refining the methodology could further enhance the impact and robustness of the study. Overall, the manuscript is a valuable contribution to the field of cloud physics and climate science and is recommended for publication with minor revisions.*

We thank the reviewer again for these helpful and valuable comments.